PREPARED FOR SUBMISSION TO JHEP                                      BRX-TH6725

# Entangled universes

**Divij Gupta,**$^{a,b}$ **Matthew Headrick,**$^b$ **and Martin Sasieta**$^b$

$^a$*Department of Physics, University of Illinois, Urbana IL 61801, USA*

$^b$*Martin Fisher School of Physics, Brandeis University, Waltham MA 02453, USA*

*E-mail:* divijg3@illinois.edu, headrick@brandeis.edu, martinsasieta@brandeis.edu

ABSTRACT: We propose a generalization of the RT and HRT holographic entanglement entropy formulas to spacetimes with asymptotically Minkowski as well as asymptotically AdS regions. We postulate that such spacetimes represent entangled states in a tensor product of Hilbert spaces, each corresponding to one asymptotic region. We show that our conjectured formula has the same general properties and passes the same general tests as the standard HRT formula. We provide further evidence for it by showing that in many cases the Minkowski asymptotic regions can be replaced by AdS ones using a domain wall. We illustrate the use of our formula by calculating entanglement entropies between asymptotic regions in Brill-Lindquist spacetimes, finding phase transitions similar to those known to occur in AdS. We construct networks of universes by gluing together Brill-Lindquist spaces along minimal surfaces. Finally, we discuss a variety of possible extensions and generalizations, including to universes with asymptotically de Sitter regions; in the latter case, we identify an ambiguity in the homology condition, leading to two different versions of the HRT formula which we call "orthodox" and "heterodox", with different physical interpretations.

A video abstract is available at https://youtu.be/2g0pyr6ugrc.

# Contents

---

# 1  Introduction

The Ryu-Takayanagi (RT) formula [1] is one of the foundational results of holography in AdS spaces. It formalizes the idea that the classical connected geometry of the AdS bulk emerges collectively from quantum correlations of a large number of microscopic degrees of freedom of the CFT, in the form of quantum entanglement [2]. The formula reads

$$S(A) = \frac{|\gamma_{\mathrm{RT}}(A)|}{4G_{\mathrm{N}}}\,. \tag{1.1}$$

On the left-hand side, $S(A) = -\mathrm{Tr}(\rho_A \log \rho_A)$ is the entanglement entropy of the CFT state $\rho_A$ restricted to the spatial subregion $A$. On the right-hand side, $\gamma_{\mathrm{RT}}(A)$ is the RT surface, defined as the minimal-area surface homologous to the boundary subregion $A$, and $|\cdot|$ denotes its area.

The RT formula (1.1) has many important consequences. Notably, it makes more transparent the mechanism for the emergence of space in holography. The observation is that versions of (1.1) hold in tensor network models parametrizing the entanglement structure of critical many-body ground states [3]. Despite being very crude models of gravity, the "emergent AdS space" is simply the tensor network in these models.

Additionally, the RT surface or its generalizations [4, 5] delimit the *entanglement wedge* of $A$: the bulk region whose quantum information is fully determined by $A$ [6–10]. In this sense, RT surfaces characterize the holographic encoding of the local structure of the bulk. The properties of RT surfaces in AdS space ensure that such an encoding is robust against partial erasures of the physical CFT space, a defining property of a quantum error-correcting code [7, 11, 12]. Entanglement wedge reconstruction has also provided a new perspective on the unitary evaporation of black holes [13, 14].

In many respects, the RT formula can be viewed as a particular manifestation of ER = EPR [15]. On the other hand, ER = EPR is expected to be far more general, since it only relies on quantum entanglement, a defining property of quantum systems. Therefore, within our limited understanding of holography beyond AdS, it is reasonable to expect that some lessons from the RT formula can be extrapolated to other spacetimes, assuming that quantum mechanics is what ultimately defines gravity non-perturbatively in those spacetimes. In particular, the lessons which should readily

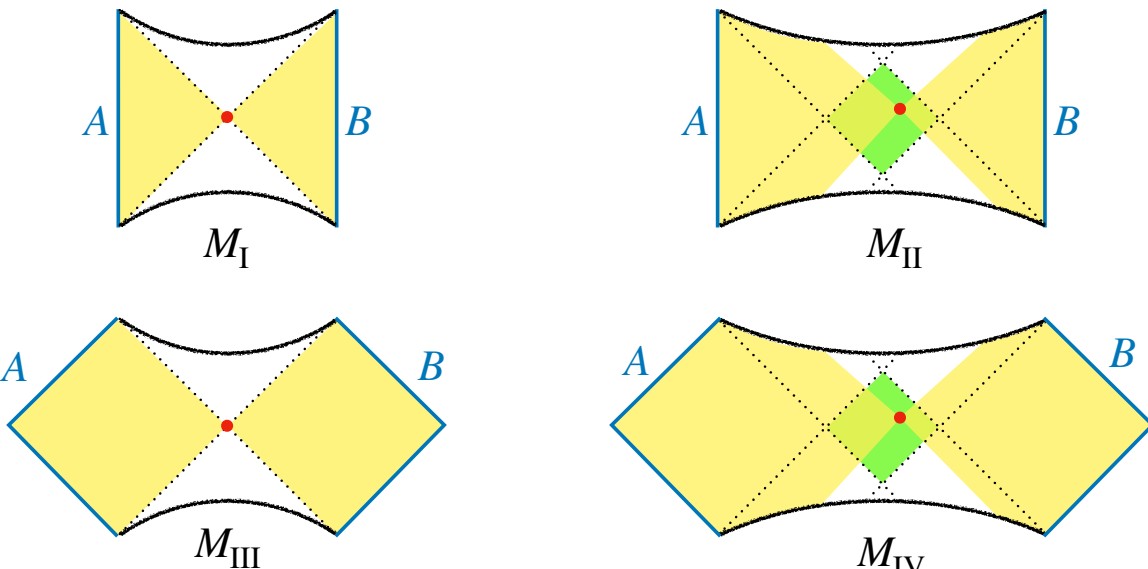

**Figure 1**. Penrose diagrams for the spacetimes $M_\mathrm{I}, M_\mathrm{II}, M_\mathrm{III}, M_\mathrm{IV}$ discussed in the main text. $M_\mathrm{I}$ is an AdS-Schwarzschild spacetime; the conformal boundaries $A, B$ are shown in blue, the exterior regions $W(A), W(B)$ in yellow, the bifurcation surface $\gamma_\mathrm{bif}$ in red, the future and past singularities in black, and the event horizons as dotted lines. $M_\mathrm{II}$ is a generic two-boundary asymptotically AdS wormhole; the causal shadow is shown in green, the entanglement wedges in yellow, and the HRT surface $\gamma_\mathrm{HRT}$ in red. (Of course, a truly generic spacetime does not have the spherical symmetry required to draw a Penrose diagram. Our purpose here is simply to illustrate the important qualitative features of the spacetime.) $M_\mathrm{III}$ is a Schwarzschild spacetime and $M_\mathrm{IV}$ is a generic two-sides asymptotically flat spacetime, with the same features shown as for $M_\mathrm{I}$ and $M_\mathrm{II}$.

generalize are those that do not rely on the spatial locality of the holographic system, given that, unlike holographic CFTs, there is no reason for the putative quantum systems describing these spacetimes to be spatially local.

## 1.1 Main claim

The main purpose of this paper is to propose and study a generalization of the RT formula, and its covariant Hubeny-Rangamani-Takayanagi (HRT) version, to multiboundary wormhole spacetimes with *asymptotically flat boundaries*. (Later in the paper we will also consider de Sitter and other asymptotics.) To introduce and motivate this generalization, we will step through a series of four examples, before proceeding to the general case. The corresponding spacetimes are illustrated in Fig. 1.

## I. AdS-Schwarzschild

To begin, suppose we have a (UV-complete, non-perturbatively defined) quantum theory of gravity that admits a semi-classical $\mathrm{AdS}_D$ $(D \geq 3)$ vacuum $V$. The definition of $V$ includes specifying the compactification manifold (if any), the values of any moduli, etc. By semi-classical, we mean that $V$ is not in the strict classical limit but is close enough to be well described by classical Einstein gravity. Quantized with $\mathrm{AdS}_D$ boundary conditions corresponding to $V$, which we'll call $\mathrm{AdS}^V$ boundary conditions for short, this theory has a Hilbert space $\mathcal{H}^V$. (Such boundary conditions typically entail fixing the leading fall-off of the fields near the conformal boundary, while allowing subleading fall-offs to fluctuate.) $\mathcal{H}^V$ equals the Hilbert space of the CFT dual to $V$, quantized on $\mathbf{R} \times S^{D-2}$, and there is a matching of observables between the two descriptions.

This theory necessarily admits (maximally extended) AdS-Schwarzschild solutions, which have two exterior regions, each obeying $\mathrm{AdS}^V$ boundary conditions; see fig. 1 (top left). Let $M_\mathrm{I}$ be such a solution, and label the two conformal boundaries $A, B$, and corresponding exterior regions $W(A), W(B)$. An observer in either exterior region sees a static, eternal black hole, suggesting that their universe is in a mixed state, with entropy given by the Bekenstein-Hawking entropy:

$$S_{\mathrm{BH}} := \frac{|\gamma_{\mathrm{bif}}|}{4G_{\mathrm{N}}}, \tag{1.2}$$

where $\gamma_{\mathrm{bif}}$ is the bifurcation surface. We know that this spacetime indeed represents an entangled state in two copies of $\mathcal{H}^V$, with entanglement entropy given by (1.2). More precisely, there is a class of states,

$$\mathcal{H}_{M_\mathrm{I}} \subset \mathcal{H}_A^V \otimes \mathcal{H}_B^V, \tag{1.3}$$

that are all described by the same classical spacetime $M_\mathrm{I}$. The states in $\mathcal{H}_{M_\mathrm{I}}$ differ by the state of the bulk quantum fields (with negligible backreaction), as well as by non-perturbatively small parts of the wave function. For example, if $M_\mathrm{I}$ is above the Hawking-Page transition, then $\mathcal{H}_{M_\mathrm{I}}$ includes a thermofield-double state, in which the bulk fields are themselves in a thermofield-double state, and the wave function also has non-perturbatively small parts representing other spacetimes. $\mathcal{H}_{M_\mathrm{I}}$ is sometimes called the "code subspace". Its precise definition requires specifying various tolerances, such as how much backreaction is allowed; these details will not concern us. The important thing is that, for any (pure or mixed) state in $\mathcal{H}_{M_\mathrm{I}}$, the entropies of both $A$ and $B$ are given, at leading order in $G_{\mathrm{N}}$, by the Bekenstein-Hawking entropy:

$$S(A) = S_{\mathrm{BH}} + O(G_{\mathrm{N}}^0), \qquad S(B) = S_{\mathrm{BH}} + O(G_{\mathrm{N}}^0). \tag{1.4}$$

Throughout the rest of this paper (except in subsection 1.3, where we briefly discuss quantum corrections), we will be concerned only with the $O(G_N^{-1})$ part of the entropy, and will suppress the "$+O(G_N^0)$".

We also know that any operator on $\mathcal{H}_{M_I}$ representing a bulk observable localized in $W(A)$, can be lifted to an operator on $\mathcal{H}_A^V \otimes \mathcal{H}_B^V$ of the form $\mathcal{O} \otimes I$. We will abbreviate this statement by saying, "Observables in $W(A)$ lift to operators in the $A$-subalgebra." And similarly for $B$.

## II. General two-sided AdS black hole

Now suppose we deform the AdS-Schwarzschild spacetime by adding matter, gravitational waves, etc., without changing the $\text{AdS}^V$ boundary conditions. We'll call the deformed spacetime $M_{II}$. Assuming the matter obeys the null energy condition, the event horizons separate: the past horizon of $B$ now lies to the future of the future horizon of $A$, rather than coinciding as they do in AdS-Schwarzschild. As a result, the two conformal boundaries remain out of causal contact, and the causal shadow (the set of bulk points out of causal contact with both boundaries) becomes a codimension-0 spacetime region; see Fig. 1 (bottom left). An observer in one of the external regions now sees a black hole with time-dependent past and future event horizons; while they will still suspect that their universe is in a mixed state, they will not necessarily know its fine-grained entropy, since the time-dependent horizon areas are coarse-grained entropies.

As omniscient observers, we know that $M_{II}$ does indeed represent a set of entangled states $\mathcal{H}_{M_{II}} \subset \mathcal{H}_A^V \otimes \mathcal{H}_B^V$, and we also know the fine-grained entropy: it is given by the area of the HRT surface $\gamma_{\text{HRT}}$, a distinguished extremal surface lying in the causal shadow:

$$S(A) = S(B) = \frac{|\gamma_{\text{HRT}}|}{4G_N}. \tag{1.5}$$

The HRT surface also defines entanglement wedges $W(A), W(B)$ that play the role of the external regions of AdS-Schwarzschild, in the sense that observables in $W(A)$ lift to operators in the $A$-subalgebra, and similarly for $B$.

So far everything we've said is well known. We would like to point out, however, something that is not often said in this context: to make the statements in the previous paragraph, it is not necessary to invoke the existence of a CFT description of $\mathcal{H}_A^V$ and $\mathcal{H}_B^V$. This is because the entanglement in question is between two copies of the entire CFT, and the latter equals the quantum gravity quantized with $\text{AdS}^V$ boundary conditions. (In contrast, applications of the HRT formula involving boundary-anchored extremal surfaces refer to spatial factorizations of the CFT Hilbert space, a specifically field-theoretic concept with no straightforward analogue in quantum gravity.) We can

therefore choose to ignore the CFT and simply say that $M_{\mathrm{I}}$ and $M_{\mathrm{II}}$ represent entangled states of two AdS universes.

## III. Schwarzschild

Our claim is now that the entire story told above carries over to the asymptotically flat setting, essentially without change, except for the part about the CFT dual theory. We will again start with a Schwarzschild spacetime, then deform it.

Suppose we now have a quantum gravity theory that admits a $D$-dimensional Minkowski (Mink$_D$) vacuum $V'$. There is a Hilbert space $\mathcal{H}^{V'}$ for the theory quantized with Mink$^{V'}$ boundary conditions. A Schwarzschild spacetime $M_{\mathrm{III}}$ has two asymptotic regions obeying those boundary conditions.[1] We again label the conformal boundaries $A, B$ and exterior regions $W(A), W(B)$; see Fig. 1 (top right). An observer in either exterior region observes a static, eternal black hole, suggesting their universe is in a mixed state with entropy given again by the Bekenstein-Hawking entropy (1.2). On the other hand, the *entire* spacetime is not bounded by an event horizon, suggesting that it is in a *pure* state, or at least that its entropy is less than $O(G_{\mathrm{N}}^{-1})$. In other words, the two external regions purify each other. We know this is true for the quantum fields on either side of the bifurcation surface, so it seems plausible that it should be true for the gravitational field at the non-perturbative level as well.

We therefore come to the tentative conclusion that, like its AdS counterpart, Schwarzschild represents a set of entangled states, with entanglement entropy given by the Bekenstein-Hawking formula:[2]

**Conjecture 1.** $M_{\mathrm{III}}$ *represents a set of entangled states in two copies of* $\mathcal{H}^{V'}$,

$$\mathcal{H}_{M_{\mathrm{III}}} \subset \mathcal{H}_A^{V'} \otimes \mathcal{H}_B^{V'}, \tag{1.6}$$

*with entanglement entropy given by*

$$S(A) = S(B) = \frac{|\gamma_{\mathrm{bif}}|}{4G_{\mathrm{N}}}. \tag{1.7}$$

---

[1]Taking into account quantum corrections, the Schwarzschild black hole will eventually evaporate. We can either continuously feed it energy from the past null boundaries to counteract the evaporation and keep it static, or simply accept that the description given here is approximate and valid only over some long timescale of order $\hbar^{-1}$. This issue also arises for the other spacetimes considered in this paper (except for a large AdS-Schwarzschild black hole, which may be in equilibrium with its Hawking radiation, thereby representing a truly static spacetime). However, it does not affect the considerations in this paper, as we are working in a semiclassical regime and not considering very long timescales.

[2]The idea that the (Minkowski) Schwarzschild solution represents an entangled state can be bolstered by showing that a coupling between the two sides can make the wormhole traversable, which is interpreted as teleportation from the viewpoint of the quantum systems [16].

*Furthermore, observables in $W(A)$ lift to operators in the A-subalgebra, and similarly for B.*

The existence of a dual description of $\mathcal{H}^{V'}$ — be it a quantum field theory or any other kind of quantum mechanical system — is neither needed nor claimed for this conjecture. Thus, we can simply say that the Schwarzschild spacetime represents an entangled state of two asymptotically flat universes.

## IV. General two-sided asymptotically flat black hole

As in case II, we now obtain a spacetime $M_{\mathrm{IV}}$ by deforming the Schwarzschild spacetime $M_{\mathrm{III}}$ by adding matter and/or gravitational waves while maintaining the Mink$^{V'}$ boundary conditions. (The matter and waves may enter the spacetime through the past singularity or $i^-$ or $\mathcal{I}^-$ of either conformal boundary, and leave through the future singularity or $i^+$ or $\mathcal{I}^+$ of either conformal boundary.) We call the resulting spacetime $M_{\mathrm{IV}}$. The story is very similar. Again, the event horizons separate, the conformal boundaries remain outside of causal contact, and the causal shadow grows to be codimension-0; see Fig. 1 (bottom right).

An observer in one of the exterior regions again sees a black hole with time-dependent past and future horizons, suggesting that their universe is in a mixed state. However, assuming conjecture 1 is correct, the entire spacetime presumably still represents some set of entangled states, with entanglement entropy of order $G_{\mathrm{N}}^{-1}$. But what is the value of the entanglement entropy? Is there, in the causal shadow, an extremal surface that can play the role that $\gamma_{\mathrm{HRT}}$ did for $M_{\mathrm{II}}$? In the next section, we will argue that such a surface, which we will continue to call $\gamma_{\mathrm{HRT}}$, does exist, and defines entanglement wedges $W(A), W(B)$ in the same way as in asymptotically AdS spacetimes. We therefore propose:

**Conjecture 2.** *$M_{\mathrm{IV}}$ represents a set of entangled states in two copies of $\mathcal{H}^{V'}$,*

$$\mathcal{H}_{M_{\mathrm{IV}}} \subset \mathcal{H}_A^{V'} \otimes \mathcal{H}_B^{V'} , \tag{1.8}$$

*with entanglement entropy given by*

$$S(A) = S(B) = \frac{|\gamma_{\mathrm{HRT}}|}{4G_{\mathrm{N}}} . \tag{1.9}$$

*Furthermore, observables localized in $W(A)$ lift to operators in the A subalgebra, and similarly for B.*

In Schwarzschild, $\gamma_{\mathrm{HRT}} = \gamma_{\mathrm{bif}}$, so conjecture 1 is a special case of conjecture 2.

**General multiboundary spacetime**

There are many well-established generalizations of case II. For example, the conformal boundary may have more than two connected components; the spacetime may have a more complicated topology; the conformal boundaries may be other than $\mathbf{R} \times S^{D-2}$, with the spacetime obeying asymptotically *locally* $\mathrm{AdS}_D$ boundary conditions; and the boundary conditions (compactification manifold, values of moduli, cosmological constant, etc.) may vary over the conformal boundary. In all such cases, the HRT formula gives the entanglement entropy between any subset of the boundary connected components and its complement.

Assuming conjecture 2 is correct, then by analogy it makes sense to extend it to general numbers of asymptotic regions and topology, and to varying boundary conditions. Given a quantum theory of gravity admitting a classical, Einstein-gravity limit, let $M$ be a connected classical solution with $n$ asymptotic regions $a_1, \ldots, a_n$, each of which is either asymptotically locally $\mathrm{AdS}_D$ or asymptotically $\mathrm{Mink}_D$. ($M$ may contain end-of-the-world branes, domain walls connecting different vacua, and other massive objects, as long as they don't carry entropy of order $G_{\mathrm{N}}^{-1}$.) As we will show in the next subsection, for any subset $A \subseteq \{a_1, \ldots, a_n\}$, there exists an extremal surface $\gamma_{\mathrm{HRT}}(A)$ and entanglement wedges $W(A), W(A^c)$ obeying all the properties we expect from these constructs in the AdS setting. We therefore propose:

**Conjecture 3.** *$M$ represents a set $\mathcal{H}_M$ of entangled states in the tensor product of $n$ Hilbert spaces $\mathcal{H}^{a_i}$, each of which is obtained by quantizing the theory with the same boundary conditions as $a_i$:*

$$\mathcal{H}_M \subset \mathcal{H}^{a_1} \otimes \cdots \otimes \mathcal{H}^{a_n} \,, \tag{1.10}$$

*with entanglement entropy between $A$ and $A^c$ given by*

$$S(A) = S(A^c) = \frac{|\gamma_{\mathrm{HRT}}(A)|}{4G_{\mathrm{N}}} \,. \tag{1.11}$$

*Furthermore, observables localized in $W(A)$ lift to operators in the $A$-subalgebra.*

A small extension of conjecture 3 would state that the HRT formula also computes entanglement entropies of regions that include *subregions* of the AdS boundaries (potentially along with entire AdS and/or Minkowski boundaries). To avoid complicating the exposition, in this paper we will focus on regions that only include entire boundaries.

Most of the rest of this paper is devoted to giving evidence for and studying examples of conjecture 3. Aside from the above argument by analogy, we will provide two lines of evidence. First, we will show that, like the standard HRT formula, it passes

a suite of non-trivial checks (section 2). Second, we will give an argument based on replacing the Minkowski asymptotic regions with AdS ones, where the standard HRT formula applies (section 3).

Theories of gravity admit other kinds of boundary conditions besides Minkowski and AdS. Our conjecture can in principle be generalized to any boundary condition that admits a non-perturbative quantization, as long as the construction and properties of the HRT surface presented in the next section go through. One can also ask about closed universes, which have no spatial boundary on which to impose boundary conditions. We will discuss the case of de Sitter asymptotics in section 5 and other asymptotics in section 6.

## 1.2 Comparison to other non-AdS entanglement entropy formulas

The above conjectures are by no means the first attempt to generalize the notion of holographic entanglement beyond asymptotically AdS spacetimes. Indeed, many such formulas have been put forward in the context of various putative holographic dualities such as flat, wedge, Carrollian, Lifshitz, Galilean, and celestial holography, dS/CFT, and so on [17–27]. However, such works typically focused on *boundary-anchored* HRT surfaces (or their analogues), which are claimed to compute the entropies of subregions of the boundary. For a subregion to have an entropy would seem to require the Hilbert space to admit a spatial tensor factorization in which the Hamiltonian is local (or, in the algebraic QFT setting, the split property), which is characteristic of a local quantum field theory and which quantum gravity theories with non-AdS asymptotics may not have. Which is not to say that applications of HRT-like formulas to non-AdS spacetimes are meaningless, just that it is not clear in every case that they are actually computing entanglement entropies. This issue is likely related to the fact that extremal surfaces generally do not gracefully meet the conformal boundaries of non-asymptotically AdS spacetimes, requiring somewhat complicated constructions to define the relevant surfaces.

The difference in this work is that we are focusing on entire boundary connected components, so that the HRT surfaces are not boundary-anchored. Our claim is that such components *do* represent tensor factors; furthermore, we can be specific that each factor is just the Hilbert space for the theory quantized with those boundary conditions. This claim does not require a dual quantum field theory, or indeed a dual theory at all; it is entirely internal to the given quantum gravity theory. In this sense, our conjecture may be considered as a more modest, or conservative, generalization of the HRT formula than the ones referenced above.

A different kind of generalization of the HRT formula beyond AdS is provided by very general formulas, such as Miyaji-Takayanagi's "surface-state" (SS) correspondence

[28] and Bousso-Penington's "generalized entanglement wedge" (GEW) proposal [29, 30] (see also [31]), which are supposed to apply to any quantum theory of gravity and do not need a boundary at all for their definition. According to SS, a Hilbert space and state are associated to any open or closed codimension-2 spacelike surface $\Sigma$ (subject to a certain convexity condition we won't enter into the details here). The entropy of the corresponding state $\rho(\Sigma)$ is then given by the area of the smallest extremal surface spacelike-homologous to $\Sigma$. The GEW proposal is most easily understood in the time-reflection symmetric case. A subset $R$ of the time-reflection symmetric slice has an entanglement wedge $W(R)$ given by the region containing $R$ with the smallest boundary area, and that area gives the entropy of the corresponding state.

Our conjecture 3 is a special case of SS, where we take the surface $\Sigma$ to be a spatial sphere near infinity in the relevant (AdS or Minkowski) asymptotic boundary (or union thereof in case $A$ includes multiple boundaries). In the time-reflection case, it is also a special case of GEW, where we take the spatial region $R$ to be the set of points on the reflection slice outside (i.e. closer to the boundaries than) $\Sigma$. In either case, the relevant surface is the HRT surface appearing in conjecture 3, so the SS and GEW conjectures reduce to the latter one. In principle, this could be viewed as evidence in favor of conjecture 3. However, conjecture 3 is a far more conservative extension of standard AdS/CFT facts than SS or GEW, and is therefore arguably on safer ground.

## 1.3 Extensions

The HRT formula sits at the center of a large and growing world of investigations into information-theoretic properties of holography and quantum gravity. Our generalization, conjecture 3, naturally incorporates many of these extensions. In fact, we have already included one of them in conjecture 3, namely entanglement wedge reconstruction (or subregion duality). Here we discuss a few others and their implications. The list presented here is far from exhaustive, and our treatment will be brief, assuming the reader is already somewhat familiar with the literature in this area. In particular, we will not repeat the arguments that have been made in favor of each extension.

**Quantum corrections:** Perturbative corrections in $G_{\mathrm{N}}$, of order $G_{\mathrm{N}}^0$ and higher, may be incorporated into the HRT formula by the quantum extremal surface formula [5], in which the quantity to be extremized is not the area of the surface $\gamma$ but its generalized entropy,

$$S_{\mathrm{gen}}(\gamma) = \frac{|\gamma|}{4G_{\mathrm{N}}} + S_{\mathrm{bulk}}(\gamma), \tag{1.12}$$

where the last term is the entropy of the bulk fields between the surface $\gamma$ and $A$. This formula is believed to hold when the state of the bulk fields is sufficiently generic, or

more precisely when its min- and max-entropies are of the same order in $G_{\mathrm{N}}$, and is replaced by a more complicated version involving those entropies in the general case [32]. We would expect all of these considerations to carry over directly to the asymptotically Minkowski setting.

**Quantum error correction:** It has been pointed out that in AdS/CFT, as a consequence of entanglement wedge reconstruction, the map from the bulk quantum fields to the boundary CFT for a given classical spacetime must have the properties of a quantum error correcting code [7, 9]. The same arguments will hold in the present context. For example, certain wormholes connecting three universes will contain a codimension-0 spacetime region lying in the entanglement wedge of any two of the three universes. (We will construct an explicit asymptotically Minkowski example in subsection 4.2.) The state of the fields within that region must be reconstructible given the reduced state on any two of the universes; or, to put it differently, it must be reconstructible given arbitrary errors introduced in any one universe.

**Outer entropy:** The paper [33] defined a "minimar surface" $\mu$ in a multiboundary holographic spacetime as a compact marginally trapped surface that is minimal on a partial Cauchy slice that intersects the boundary on a Cauchy slice for an entire boundary connected component. The authors argued that the area of such a surface equals the "outer entropy" of $\mu$, defined by maximizing the entropy of the CFT on that boundary while holding fixed the data on that partial Cauchy slice (or equivalently the data in its causal domain, which is the outer wedge of $\mu$). These definitions and arguments are equally valid in the present setting, simply replacing "CFT" by "universe".

**Entanglement wedge cross section:** Given disjoint boundary regions $A, B$, their joint entanglement wedge $W(AB)$ can be considered a spacetime in its own right, bounded spatially by the joint HRT surface $\gamma_{\mathrm{HRT}}(AB)$ as well as by $A$ and $B$ themselves. The HRT formula can then be applied within $W(AB)$, to $A$ or $B$, with the homology constraint applied relative to $\gamma_{\mathrm{HRT}}(AB)$. The resulting "HRT surface" is the same for $A$ and $B$, and is called the entanglement wedge cross section (EWCS) surface. Its area, the EWCS, has been conjectured to equal quite a few different information-theoretic quantities, such as the entanglement of purification [34, 35], the reflected entropy [36], the logarithmic negativity [37], and the odd entropy [38]. We refer to the respective papers for the definitions of those quantities. Due to the difficulty of computing these quantities from first principles in the dual field theory, it is not yet clear which one of these conjectures is correct; in fact, it is quite possible that more than one are correct, since, although the quantities are not in general equal, some of them may be equal for holographic states.

The definition of the EWCS surface goes through equally well with asymptotically Minkowski boundaries. The putative dual quantities simply require a state $\rho_{AB}$ on a product Hilbert space $\mathcal{H}_A \otimes \mathcal{H}_B$ and are therefore well-defined in the present context. Presumably, whichever one or ones of the above conjectures are correct in the AdS context are correct here as well.

**Pythons:** Suppose we have a time-reflection symmetric holographic spacetime, and that on the symmetric slice there exist two locally minimal surfaces homologous to a given boundary region $A$. If the smaller one, $\gamma_{\text{RT}}$, is further from $A$ than the larger one, called the "constriction" $\gamma_c$, then the region between them, called the "python", is contained in the entanglement wedge and therefore in principle reconstructible from $A$. However, the lack of a simple HKLL-type reconstruction [39], together with arguments related to the complexity of decoding Hawking radiation, have suggested that this reconstruction is much more complex than the reconstruction of the exterior region between $\gamma_c$ and $A$. This complexity can be quantified in terms of the number $\mathcal{C}$ of gates required to write a generic local observable in the python as a circuit, where the gates are taken from a set of simple boundary operators in $A$, and simple means dual to a bulk local operator near the boundary. Specifically, it was conjectured in [40] that the complexity $\mathcal{C}$ goes like

$$\mathcal{C} \sim \exp\left[\frac{|\gamma_b| - |\gamma_c|}{8G_{\text{N}}}\right], \tag{1.13}$$

where $\gamma_b$ is the "bulge" surface, a non-minimal extremal surface contained in the python, whose existence can be inferred by a minimax argument in the space of surfaces homologous to $A$. Eq. (1.13) is the "python's lunch conjecture". (See [40–43] for specifics and further discussion, including the covariant generalization.)

The phenomenon of competing candidate RT surfaces also occurs in asymptotically Minkowski spacetimes; we will see explicit examples in section 4. The same minimax argument then implies the existence of a bulge surface. And the notion of a "simple" operator can be maintained without reference to the CFT, as a local observable in the asymptotic region. So the python's lunch conjecture can be ported directly over to the present context.

**Tensor networks:** Tensor networks (TNs) were originally proposed as toy models of holographic states because they obey a form of the RT formula and because they mimic the hyperbolic geometry and relation between energy scale and position in the extra dimension that are characteristic of holography [3]. The latter property is specific to AdS/CFT, and while it might be possible to construct a TN that is geometrically flat (or asymptotically flat), it is not obvious that the states such a network produces would be in any sense holographic.

However, there is a coarser kind of TN that is specifically adapted to computing entanglement entropies, without necessarily reproducing the UV physics of a CFT. The paper [44] starts from a Cauchy slice $\Sigma$ of a holographic spacetime, along with a fixed set of boundary regions $A, B, \ldots$, and constructs a TN based on a decomposition of $\Sigma$ along RT surfaces of the regions $A, B, \ldots$, along with a set of composite regions. Thus, each node in the network corresponds to a region of $\Sigma$, and each link to a (part of or whole) RT surface, with bond dimension equal to the area over $4G_{\mathrm{N}}$. If the set of composite regions is chosen without crossings, then the RT surfaces do not intersect, and the network is a tree. In that case, the tensors can be chosen so that the TN description of the state is exact. (If the RT surfaces intersect, then the situation is more complicated, and it is not clear if an exact TN description of the state exists.)

This coarse TN description should exist for asymptotically Minkowski spacetimes, assuming (as always) that the boundary regions are taken to be made up of entire connected components. We will discuss this further in subsection 6.1.

## 1.4 Outline of paper

The rest of this paper is organized as follows. In section 2, we define the HRT surfaces for asymptotically Minkowski and mixed AdS/Mink spacetimes and study their properties. Specifically, subject to some mild assumptions about the spacetime, we argue that: extremal surfaces in the relevant homology classes exist; they are outside of causal contact with all conformal boundaries; the entanglement wedges obey complementarity and nesting; and their areas obey strong subadditivity and are smaller than those of the relevant causal horizons. These properties give support to conjecture 3. We also show that, if a spacetime admits a global time-reflection symmetry, then the HRT surface is globally minimal on the invariant slice (RT formula).

In section 3, we give further support to conjecture 3, by showing that, in any quantum gravity theory that admits both $\mathrm{AdS}_D$ and $\mathrm{Mink}_D$ vacua connected by domain walls, it can be obtained as a limit of the standard HRT formula. The argument involves replacing Minkowski asymptotic regions with AdS ones, following the procedure of [45]. Along the way, we make some new observations about the gluing procedure. (More details are provided in appendix A.)

Multiboundary AdS wormholes, particularly in $D = 3$, are widely studied in the holographic literature (see e.g. [46, 47] for some early work). On the other hand, asymptotically flat multiboundary wormholes, to which we can apply conjecture 3, are perhaps not as familiar, and the reader may wonder whether they actually exist. In fact, there is a known class of exact vacuum initial data in $D = 4$ with an arbitrary number of asymptotic regions, called Brill-Lindquist metrics [48]. We study these in section 4, numerically finding the relevant RT surfaces and studying the entanglement

entropies. Among other phenomena, we find a mutual information phase transition similar to the familiar ones in AdS. We also find non-minimal extremal (bulge) surfaces relevant to the python's lunch conjecture [40]. We show that Brill-Lindquist geometries can be glued together to form networks of entangled universes (including spacetimes with zero and one asymptotic regions) and discuss their entanglement properties.[3]

In section 5, we consider generalizing conjecture 3 to spacetimes with asymptotically de Sitter regions. We point out several distinct consistent generalizations of conjecture 3, and comment on their possible physical interpretations.

Finally, in section 6, we conclude by discussing some further points and future directions, including tensor networks for describing entangled universes and the creation of entangled universes in inflation and by the collapse of radiation in flat space.

## 2   General properties of entangled universes

In this section, we will define HRT surfaces and entanglement wedges for spacetimes with asymptotically Minkowski and/or AdS regions. We will show that they share all the general properties of HRT surfaces in the usual AdS setting, namely:

- existence;

- lying in the causal shadow (causal wedge inclusion);

- complementarity;

- nesting;

- strong subadditivity;

- MMI;

- having area less than or equal to that of the causal information surface;

- in the presence of a time reflection symmetry, being minimal on the time reflection invariant slice (RT formula).

Our main point is that the usual arguments for these properties, from the AdS setting, go through just as well with asymptotically Minkowski boundaries, so we will be very brief.

---

[3]A forthcoming paper by one of the present authors [49] will use the Brill-Lindquist geometries to construct a toy model of black hole evaporation in flat space, following the example of the AdS model developed in [50].

As we will discuss, some of these properties constitute checks on conjecture 3, while others are interesting or useful properties of the surface or entropy. We will also present several equivalent formulations of the HRT formula: maximin, minimax, U-flow, and V-flow.

## 2.1 Setup

The bulk spacetime $M$, with dimension $D_M$, is assumed to have asymptotic regions $a_1, \ldots, a_n$ ($2 \leq n < \infty$), where each $a_i$ is either asymptotically locally $\mathrm{AdS}_D \times X_i$ or asymptotically $\mathrm{Mink}_D \times X_i$, the dimensions satisfy $D_M \geq D \geq 3$, and $X_i$ is a compact, connected manifold of dimension $D_M - D$. Each asymptotically AdS region $a_i$ has a conformal boundary $b_i$ that is a connected $(D-1)$-dimensional manifold equipped with a causal structure. Each Minkowski asymptotic region $a_i$ has a conformal boundary $b_i$ consisting of a point $i_i^0$ at spacelike infinity and $(D-1)$-dimensional null boundaries $\mathcal{I}_i^{\pm}$.

We assume that $M$ is (AdS) globally hyperbolic (GH). A Cauchy slice $\sigma$ for $M$ meets each boundary $b_i$ on $\sigma_i$, where $\sigma_i$ is a Cauchy slice for $b_i$ in the AdS case and $\sigma_i = i_i^0$ in the Minkowski case. (Henceforth, we will abbreviate *Cauchy slice* by *slice*.) We assume that, with these boundaries adjoined, $\sigma$ is compact. We also assume that the (Einstein-frame) metric $g_{\mu\nu}$ on $M$ is smooth and obeys the null curvature condition (NCC, $R_{\mu\nu} k^\mu k^\nu \geq 0$ for all null vectors $k^\mu$, a consequence of the Einstein equation and null energy condition). Finally, we assume that $M$ is complete, in the sense that it cannot be contained in a larger globally hyperbolic spacetime.

A *surface* is a $(D_M - 2)$-dimensional spacelike submanifold of $M$. Areas of surfaces are measured with respect to $g_{\mu\nu}$. A surface is *extremal* if it extremizes the area, or equivalently, if the trace of its extrinsic curvature vanishes.

The following key facts, which underpin most properties of HRT surfaces, follow from the assumptions of GH and NCC:

1. Given a closed surface $\gamma$, each point $x$ on the boundary of $J^{\pm}(\gamma)$ is connected to $\gamma$ by a null geodesic that meets $\gamma$ orthogonally and does not contain a conjugate point between $\gamma$ and $x$.

2. Given a congruence of null geodesics, if its expansion at a point $x$ is non-positive, then the expansion at all points further along the geodesic from $x$ is non-positive (focusing).

3. A null congruence of which every geodesic reaches a boundary $b_i$ has infinite limiting area, and therefore everywhere positive expansion.

We can generalize the setup slightly to allow $g_{\mu\nu}$ to have timelike singularities, e.g. conical singularities or junctions (hypersurfaces on which the metric is continuous but not differentiable), as long as they can be smoothed out in a way that preserves GH and NCC, so that the above facts still hold. End-of-the-world branes are also allowed, again so long as the above facts are valid. This will hold as long as the spacetime can be doubled along the brane (i.e. glued to a reflected copy of itself) and then smoothed out to produce a spacetime that obeys GH and NCC. The extremality condition on a surface, in particular the requirement to be extremal with respect to changes in where it intersects the brane, implies that the intersection must be orthogonal. The homology condition on a surface (defined below) becomes homology relative to the brane.

From the causal future and past $J^{\pm}(b_i)$ of each boundary, we define:

- The past and future event horizons $CH_i^{\mp}$, which are the past and future boundaries of $J^{\pm}(b_i)$ respectively. These are null congruences, with everywhere non-positive expansion going away from $b_i$.

- The causal wedge $CW_i := J^+(b_i) \cap J^-(b_i)$.

- The causal information surface $\Xi_i := CH_i^+ \cap CH_i^-$.

A generator of the event horizon $CH_i^{\mp}$ cannot reach any conformal boundary. Therefore, the boundaries are mutually acausal, and the causal wedges do not intersect; in other words, the wormhole is not traversable.

The *causal shadow CS* is the set of points in $M$ out of causal contact with all the boundaries:

$$CS := M \setminus \bigcup_i \left( J^+(b_i) \cup J^-(b_i) \right). \tag{2.1}$$

Any closed extremal surface must lie entirely in $CS$; else a subset of the generators of the boundary of its past or future would reach $b_i$ for some $i$, which is impossible since the expansion is initially zero. (If we were considering subregions of the AdS boundaries, the causal shadow would depend on the subregions. Here, since we're considering entire boundaries, it does not; there is simply a single causal shadow for the entire spacetime.)

## 2.2 HRT surface: definition & properties

Given a subset $A \subseteq 1, \ldots, n$ of the boundaries, a surface $\gamma$ is *homologous* to $A$ if there exists a Cauchy slice $\sigma$ containing $\gamma$ and a region $r \subset \sigma$ such that

$$\partial r = \gamma \cup \bigcup_{i \in A} \sigma_i. \tag{2.2}$$

Such a surface is necessarily closed. It follows immediately, via the region $r^c := \sigma \setminus r$, that $\gamma$ is also homologous to $A^c$. It divides $M$ into four spacetime regions: its future and past $J^\pm(\gamma)$ and the two wedges $w(\gamma, A) := D(r)$, $w(\gamma, A^c) := D(r^c)$. In the cases $A = \emptyset$ and $A = \{1, \ldots, n\}$, the empty set is homologous to $A$.

Following [4], the *HRT surface* $\gamma_{\mathrm{HRT}}(A)$ is defined as the least-area extremal surface homologous to $A$. We immediately have $\gamma_{\mathrm{HRT}}(A) = \gamma_{\mathrm{HRT}}(A^c)$. To simplify the exposition, we will assume that the minimum is unique; this assumption can be relaxed at the expense of a few mild complications. The *entanglement wedge* is the wedge for $\gamma_{\mathrm{HRT}}(A)$, $W(A) := w(\gamma_{\mathrm{HRT}}(A), A)$, and similarly for $A^c$. Since $\gamma_{\mathrm{HRT}}(A)$ is in the causal shadow, $W(A) \supseteq CW_i$ for all $i \in A$, and $W(A^c) \supseteq CW_i$ for all $i \in A^c$. The future and past boundaries of $W(A)$ are the *entanglement horizons* $EH^\pm(A)$; we write $EH(A) := EH^+(A) \cup EH^-(A)$. For any slice $\sigma$, $EH(A) \cap \sigma$ is a surface homologous to $A$ with area less than or equal to $\gamma_{\mathrm{HRT}}(A)$.

**Lemma 1.** *If a surface $\gamma$ is homologous to $A$, extremal, and minimal on some slice $\sigma$, then $\gamma = \gamma_{\mathrm{HRT}}(A)$.*

*Proof.* Otherwise, $|\gamma| > |\gamma_{\mathrm{HRT}}(A)| \geq |EH(A) \cap \sigma|$, a contradiction with minimality of $\gamma$ on $\sigma$. $\qquad\square$

In particular, if $M$ has a time-reflection symmetry with invariant slice $\sigma$, and $\gamma$ is the minimal surface homologous to $A$ on $\sigma$, then it is necessarily extremal (since it is extremal with respect to variations both within $\sigma$ and, by symmetry, orthogonal to it), hence it is the HRT surface; this is the RT formula. All of this is the same as in the standard AdS case, as long $A$ is a union of entire boundaries.

A very useful fact that again carries over directly from the standard AdS case is that the HRT surface may be found by a maximin procedure, as follows [51]. On each slice, find the minimal surface homologous to $A$, then maximize the minimal surface area over the choice of slice. The arguments used to show that a minimum and a maximum exist, and that the resulting surface is indeed the minimal-area extremal one, are not much affected by changing the asymptotic geometry from AdS to Minkowski. For the existence of a minimal surface on a given slice, the key point is that the metric is "large" near the boundaries, forcing the existence of a minimum in the interior; this still holds if some of the boundaries are asymptotically flat rather than hyperbolic. The argument for the existence of a maximum is the hardest step, and as far as we know a fully general proof does not exist. Roughly speaking, the assumption of completeness implies that the spacetime is bounded in the past and future either by some kind of singularity, where the spatial metric (or some components of it) collapse, or by a null Cauchy horizon (as in Reissner-Nordstrom). In either case, the minimal surface area

decreases as the past and future boundaries are approached, implying the existence of a maximum somewhere between them. In this very crude discussion, we are of course eliding all kinds of technical issues, for example concerning the topology on the space of slices and the continuity of the area functional; our point is simply that the same arguments apply equally well for asymptotically Minkowski as for asymptotically AdS spacetimes. The fact that the maximin surface is extremal is local and has nothing to do with the asymptotics of the spacetime. Finally, since it is extremal and minimal on a slice, by lemma 1, it is the HRT surface.

The remaining properties mentioned above are proved using the maximin formulation, and in particular the existence of a slice $\sigma$ on which $\gamma_{\text{HRT}}(A)$ is minimal. For example, its area is no larger than the sum of the causal information surfaces [51]

$$|\gamma_{\text{HRT}}(A)| \leq \sum_{i \in A} |\Xi_i| \,. \tag{2.3}$$

This is proved as follows. If we let $I_i$ be the interior part of the event horizons,

$$I_i := \left( EH_i^+ \setminus J^+(b_i) \right) \cup \left( EH_i^- \setminus J^-(b_i) \right), \tag{2.4}$$

each point on $I_i$ is connected to $\Xi_i$ by a null geodesic, on which the expansion is non-positive moving away from $\Xi_i$. Therefore, for any slice $\sigma$, the area of its intersection with $I_i$ is bounded above by $|\Xi_i|$. Minimality of $\gamma_{\text{HRT}}(A)$ on $\sigma$ then gives (2.3). All of this goes through whether the boundaries are AdS or Minkowski.

For nested boundary sets $A$ and $AB$, by applying maximin to the sum of the areas of surfaces homologous to each boundary set, one can show that there exists a slice on which $\gamma_{\text{HRT}}(A)$ and $\gamma_{\text{HRT}}(AB)$ are both minimal [51]. This fact is then used to prove that the respective entanglement wedges are nested, $W(AB) \supset W(A)$, and also that the areas of HRT surfaces obey the strong subadditivity and MMI inequalities,

$$|\gamma_{\text{HRT}}(AB)| + |\gamma_{\text{HRT}}(BC)| \geq |\gamma_{\text{HRT}}(B)| + |\gamma_{\text{HRT}}(ABC)| \,, \tag{2.5}$$

$$|\gamma_{\text{HRT}}(AB)| + |\gamma_{\text{HRT}}(BC)| + |\gamma_{\text{HRT}}(AC)| \geq$$
$$|\gamma_{\text{HRT}}(A)| + |\gamma_{\text{HRT}}(B)| + |\gamma_{\text{HRT}}(C)| + |\gamma_{\text{HRT}}(ABC)| \,. \tag{2.6}$$

The proofs of these statements are identical to the AdS case [51], so we will not repeat there here.

## 2.3 Implications of properties

Many of the properties of HRT surfaces discussed in the previous section are consistency checks on conjecture 3. The existence of an HRT surface is the first check. The

strong subadditivity inequality is a check on the identification of its area with an entanglement entropy. The statement that the $A$-subalgebra contains the observables in $W(A)$ requires that, for disjoint boundary sets $A, B$, first, $W(AB) \supset W(A)$ and, second, $W(A)$ and $W(B)$ are spacelike-related, so that their observables commute. The first is the nesting property, and the second is implied by nesting and complementarity (i.e. the fact that $W(A^c)$ is the complementary wedge to $W(A)$). Complementarity is in fact a bit stronger than what is needed, which is merely that $W(A)$ and $W(A^c)$ are spacelike-related. Indeed, in the case of degenerate HRT surfaces, the correct entanglement wedge is presumably the smallest one; if this is true, then $W(A)$ and $W(A^c)$ are separated by a gap, but they are still spacelike related (since degenerate HRT surfaces lie on a common Cauchy slice [52]).

The fact that $\gamma_{\text{HRT}}(A)$ lies in the causal shadow also constitutes a check on conjecture 3, since it protects the HRT surface, and hence the entropy, from being changed by disturbances to the spacetime sent in from any of the AdS or past or future Minkowski boundaries and propagating causally. Such a disturbance, which is associated with changes to the subleading fall-off of the metric and other fields near one of the boundaries $b_i$ (in the AdS case) or $\mathcal{I}_i^{\pm}$ (in the Minkowski case), is a local unitary — local in the sense that it acts on one Hilbert space factor $\mathcal{H}^{a_i}$.

The inequality (2.3) reflects the fact that the entanglement entropy is a fine-grained entropy whereas the causal information surface areas are presumably some kind of coarse-grained entropy, although the precise form of the coarse-graining that gives rise to the latter is not known. A more precise statement can be made about the asymptotic area of the future event horizon, which is the Bekenstein-Hawking entropy of the black hole in its final equilibrium state; here the coarse-graining simply reflects all the information available to an observer who stays outside the horizon. This area is no larger than $|\Xi_i|$; so the sum over $i \in A$ is again bounded by $|\gamma_{\text{HRT}}(A)|$.

Just as, in the standard AdS setting, these properties constitute an extensive and stringent set of consistency checks on the HRT formula and entanglement-wedge reconstruction, they also constitute an extensive and stringent set of checks on our conjecture 3. All of the assumed properties of the spacetime come into play: completeness, global hyperbolicity, dynamics in the form of the null curvature condition (which reflects the classical equations of motion of both the metric and the matter), and boundary conditions. The fact that it passes such highly non-trivial tests lends strong support to the validity of conjecture 3.

The other two properties discussed in the last subsection are not per se consistency checks, but are important to know nonetheless. The fact that the HRT surface is the minimal surface on a time-reflection slice is certainly a very useful fact that we will make extensive use of in this paper. Finally, the MMI inequality (2.6) is a constraint on the

entanglement structure of semiclassical states in quantum gravity. It is interesting that, according to our conjecture, it is not related to AdS but holds also for asymptotically flat states. The interpretation, from a quantum-information viewpoint, of the MMI inequality is unknown; a proposal, called "bipartite dominance", was put forward in [53], but has been contested [54, 55].

MMI is the simplest in an infinite family of inequalities known to be obeyed by the RT formula [56]. These inequalities are statements about the areas of minimal surfaces on Riemannian manifolds, and their proofs are independent of the topology, geometry, and boundary conditions of the manifold. Therefore, they all hold for the RT version of conjecture 3. Furthermore, there is evidence that they hold for the HRT formula as well (see [52] for a summary of the state of the art); if this is true then they presumably hold with Minkowski boundaries as well.

## 2.4   Minimax & flow formulations

In addition to maximin, the HRT formula admits several other, equivalent, formulations, namely a minimax formula and so-called V-flow and U-flow formulas. These formulas, and proofs that they are equivalent to the HRT formula, can be found in [57]. (For more on minimax, see also [58].) Here we will simply indicate how they can be adapted in the presence of a Minkowski boundary. In each case, there is a choice of whether to group the null boundaries $\mathcal{I}_i^\pm$ together with the spacelike one $i_i^0$ or together with the singularities (or Cauchy horizons) that bound the spacetime in the future and past.

The minimax formula involves finding the maximal-area acausal surface on a time-sheet (a timelike hypersurface), and then minimizing over the choice of timesheet. The boundary region $A$ is encoded in a homology condition on the timesheet. This is a *spacetime* homology condition, as opposed to the spatial homology condition found in the definition of the HRT surface. Specifically, the time-sheet $\tau$ should be the bulk part of the boundary of a spacetime region $R \subseteq M$ whose conformal boundary includes $b_i$ for all $i \in A$, and does not include $b_i$ for $i \in A^c$. For Minkowski boundaries, one can alternatively relax the homology condition to the following: the conformal boundary of $R$ must contain $i_i^0$ for $i \in A$ and not contain $i_i^0$ for $i \in A^c$. This allows $\tau$ to include components that end on $\mathcal{I}_i^\pm$. The area of an acausal surface in $\tau$ will then be unbounded above, effectively removing it from consideration at the minimization step.

The U-flow formula involves a divergenceless future-directed timelike vector field $U^\mu$, with vanishing flux on all the boundaries $b_i$. $U^\mu$ is subject to the following bound, for every spacelike curve $\mathcal{C}$ connecting $A$ to $A^c$:

$$\int_{\mathcal{C}} ds\, |U_\perp| \geq \frac{1}{4G_{\mathrm{N}}}\,, \tag{2.7}$$

where $ds$ is the proper distance along $\mathcal{C}$ and $U_\perp$ is the projection of $U^\mu$ orthogonal to the tangent vector of $\mathcal{C}$. (This nonlocal constraint can be expressed as a local constraint at the expense of adding a scalar field.) Since $U^\mu$ is divergenceless and has vanishing flux on the AdS boundaries, its flux is the same through all slices. Subject to the constraint (2.7), the minimal flux equals the HRT surface area. Alternatively, the no-flux boundary condition on $b_i$ can be dropped for the Minkowski boundaries, allowing flux to enter through $\mathcal{I}_i^-$ and leave through $\mathcal{I}_i^+$; this does not affect the flux through a slice.

Finally, the V-flow formula involves a divergenceless vector field $V^\mu$ with flux that enters the spacetime through $b_i$ ($i \in A$) and leaves through $b_i$ ($i \in A^c$). $V^\mu$ is subject to the following bound, for every timelike curve $q$:

$$\int_q dt \, |V_\perp| \leq \frac{1}{4G_{\mathrm{N}}} \,, \tag{2.8}$$

where $dt$ is the proper time along $q$ and $V_\perp$ is the projection of $V^\mu$ orthogonal to the tangent vector of $q$. (This nonlocal constraint can be expressed as a local constraint at the expense of adding a scalar field.) Subject to the constraint (2.8), the maximal flux of $V^\mu$ equals the HRT surface area. Alternatively, for Minkowski boundaries, one can require the flux to enter and leave through just $i_i^0$; since the space is so large there, this will not affect the norm bound or the maximal flux.

## 3  Replacing Minkowski boundaries with AdS

We will now make a different argument for conjecture 3. This argument assumes that we have a quantum gravity theory that admits both $\mathrm{Mink}_D$ and $\mathrm{AdS}_D$ vacua as well as a domain wall connecting them, which may be treated as a classical source obeying the null energy condition. (See e.g. [59, 60] for discussions of the question of the existence of domain walls connecting different vacua in quantum gravity.) We can then take a spacetime $M$, satisfying the assumptions laid out in the previous section, remove the Minkowski asymptotic regions $a_i$, and, using a domain wall, glue in AdS asymptotic regions $a_i'$. The result is a spacetime $M'$ with the same topology as $M$, but with only AdS conformal boundaries. According to the standard HRT formula, $M'$ represents an entangled state in the Hilbert space of $n$ CFTs, or equivalently of $n$ AdS universes. If the gluing has not introduced any smaller extremal surfaces, then the entanglement entropies are computed by the original HRT surfaces, in the original portion of the spacetime. In other words, the entanglement is controlled by the original spacetime $M$. This suggests that the entanglement entropies, at least, are the same between $M$ and $M'$, even though they involve different Hilbert spaces. In other words, the entanglement

involves the "infrared" in the parlance of AdS/CFT — the part of the spacetime far from the boundary — while going from $M$ to $M'$ changes the "ultraviolet".

## 3.1  Gluing AdS onto Schwarzschild

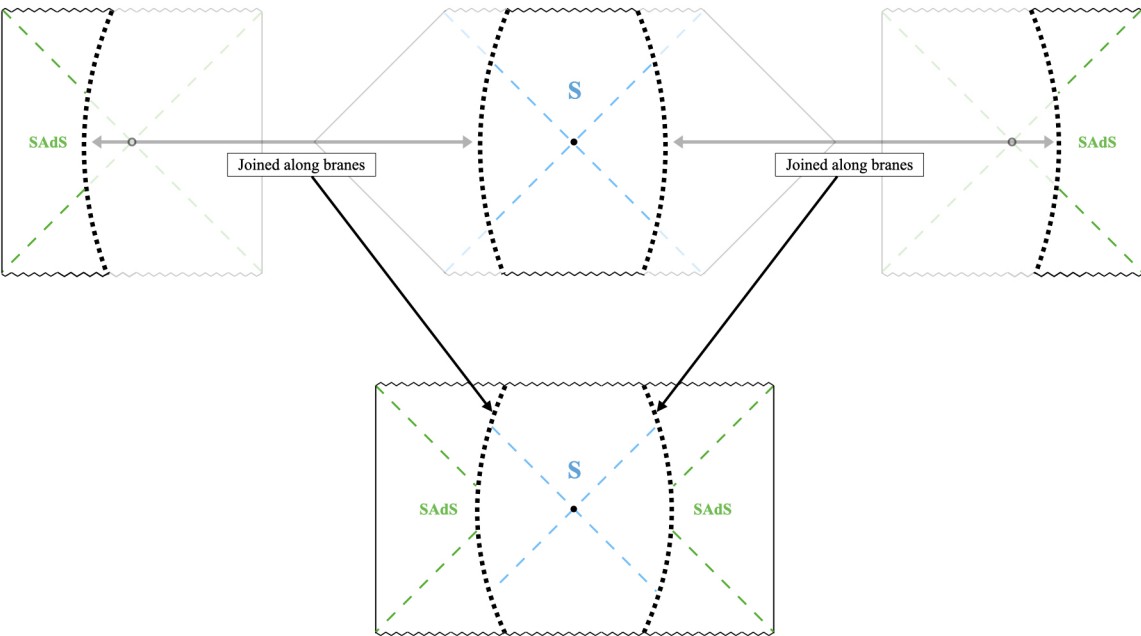

**Figure 2**. Gluing process to impose AdS boundary conditions on the Schwarzschild metric. On the top, the left and right Penrose diagrams are (identical) SAdS patches while the middle is Schwarzschild. The lighter regions are "cut out" and after gluing across the branes (dotted lines) we obtain the metric in the second line.

We will illustrate the above construction in the simplest setting, namely where $M$ is a Schwarschild solution with black hole mass $\mu_S$, and where in passing to $M'$ we preserve spherical symmetry, time reflection, and a spatial reflection that exchanges the asymptotic regions (see Fig. 2). The fact that the spacetime has a time reflection symmetry will allow us to use the RT formula. Gluings of spacetimes with zero (or positive) cosmological constant inside of asymptotically AdS ones were discussed in [45], in the thin-shell limit of the domain wall. For concreteness, following [45], we will focus on the case $D = 4$, although the construction is actually fairly independent of the dimension.

In the thin-shell approximation, the stress tensor of the wall is a delta function localized at the shell. The shell, which we also refer to as a matter brane, has an associated parameter $\kappa = 4\pi\sigma$, where $\sigma > 0$ is the domain wall tension. To have a

valid gluing, we also impose the Israel junction conditions [61], which (1) require the metric to be continuous across the domain wall and (2) relate the jump in extrinsic curvature across the junction to the brane tension. This yields the equation for the radial motion of the shell, which is constrained by various considerations, as described in Appendix A.

In the complete spacetime $M'$, the brane has a trajectory. For time-symmetric metrics, we can have 2 non-trivial trajectories: the brane starts infinitely small at $r = 0$, grows to a maximum and then shrinks back down to $r = 0$; or the brane starts infinitely large at $r = \infty$, shrinks to a finite size and then expands back to $r = \infty$. Since the brane must behave the same as viewed from either metric, the brane is attached to the same points in both metrics, i.e. either both at $r = 0$ or both at $r = \infty$. For simplicity, we assume the brane starts and ends on the singularity, i.e. $r = 0$.

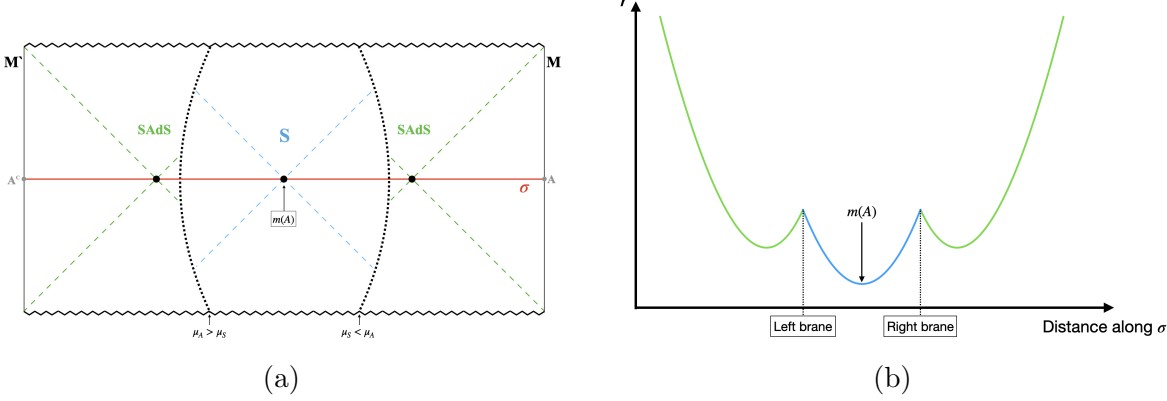

**Figure 3**. Case 1 for replacing asymptotic regions of Schwarzschild spacetime with patches of Schwarzschild-AdS spactime, requires $\mu_A > \mu_S$ for the gluing. The SAdS patches include the SAdS bifurcation surface. Including the original Schwarzschild bifurcation surface, the spacetime thus contains three candidate HRT surfaces, of which the smallest and hence the true HRT surface can be chosen to be the Schwarzschild one. The other two — the SAdS bifurcation surfaces — are constrictions for the two boundaries respectively, and the regions between them and the HRT surface is a python. Each python contains a bulge surface, an index-1 extremal surface whose area according to the python's lunch conjecture quantifies the complexity of the reconstructing observables in the python [40]. Naively, the bulge surface is located at the domain wall, since that is extremal and a local maximum of the area among spherically-symmetric surfaces. However, it was shown in [43] that this surface has index greater than 1, and the true bulge in such a circumstance lies in the vicinity of the domain wall but breaks the spherical symmetry.

We can now glue asymptotically AdS spacetimes to each boundary of a two-sided metric, such as the Schwarzschild metric, by applying the junction conditions near

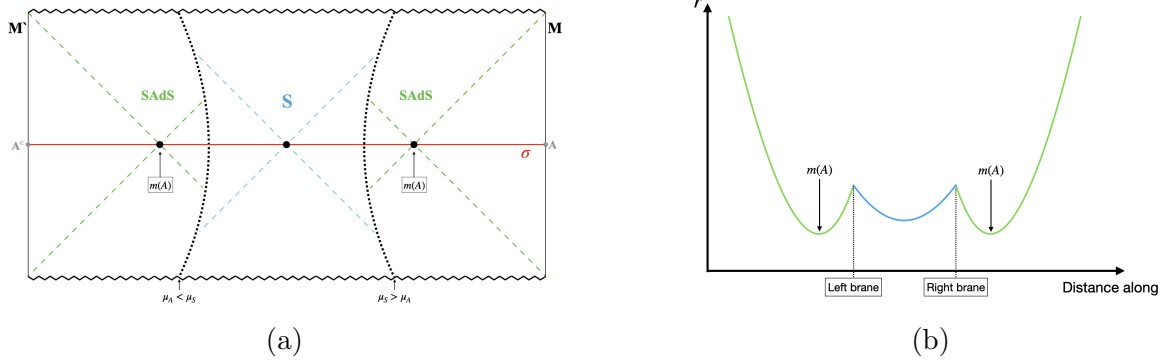

**Figure 4**. Case 2: Same as case 1 (Fig. 3), except here the gluing requires $\mu_A < \mu_S$ and so the SAdS bifurcation surfaces are smaller than the Schwarzschild one, making them the HRT surfaces.

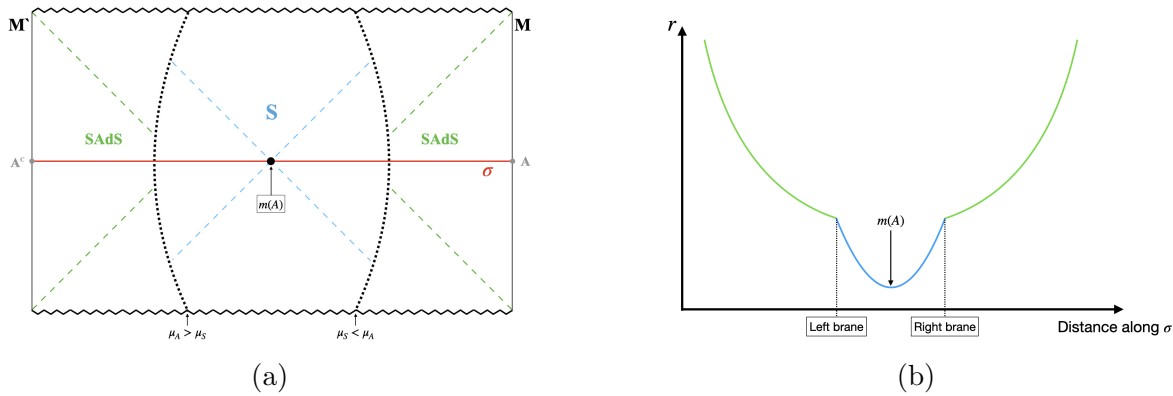

**Figure 5**. Case 3: SAdS patches do not include the bifurcation surface, so the only extremal surface is the Scharzschild bifurcation surface, which is therefore the HRT surface.

each boundary. Given the spherical symmetry, the spacetimes glued on either side are Schwarzschild-AdS (SAdS), defined by a (negative) cosmological constant $\lambda$ and a black hole mass $\mu_A$. This gives us a set of configurations for Schwarzschild in the middle with AdS boundary conditions on either side, to which we can then apply RT. There are three classes of solutions, shown in Figs. 3, 4, 5, which we'll call cases 1, 2, 3 respectively. In cases 1 and 2, the glued regions include the bifurcation surfaces of the SAdS spacetime, while in case 3 they do not. The SAdS bifurcation surfaces are extremal surfaces, homologous to each AdS boundary, and are therefore candidate HRT surfaces. The other candidate is the Schwarzschild bifurcation surface of the original spacetime. In case 2, since the gluings require $\mu_A < \mu_S$, the AdS bifurcation surface is smaller, and is therefore the HRT surface. Since case 1 requires $\mu_S < \mu_A$, the

Schwarzschild bifurcation surface can be chosen to be smaller by choosing a sufficiently large $\mu_A$. In case 3 there is only one candidate, the Schwarzschild bifurcation surface. Case 1 and 3 are therefore examples of the constructions used in the argument at the beginning of this section. Whether they are feasible depends on the values of the relevant parameters (see Appendix A); however, it is clear that there exists a large class of parameters for which gluings of case 1 or 3 are allowed and thus where the Schwarzschild bifurcation surface is the RT (minimal) surface.

Another consideration that will become relevant when extending this argument to the Brill-Lindquist wormholes in section 4 is that of performing the gluings arbitrarily far from the Schwarzschild bifurcation surface. Appendix A details how this can be done in the small tension regime ($\kappa^2 < \lambda$) while additionally ensuring $\mu_A > \mu_S$. Then, we may take $\mu_A \to \infty$ and glue the brane arbitrarily far away, while still ensuring the Schwarzschild bifurcation surface is minimal. As outlined in Appendix A, the geometry is of case 1.

## 3.2 Gluing AdS onto Schwarzschild-de Sitter

As a small aside, the same gluing procedure can be applied to the Schwarzschild-de Sitter (SdS) metric with black hole mass $\mu_S$ and cosmological constant $\lambda_{dS} > 0$. This results in configurations with a de Sitter patch inside AdS boundary conditions, to which we can apply the RT formula (see subsection 5.2 for a more in-depth discussion of RT in de Sitter). However, the presence of a cosmological horizon makes the gluings different from the previous section. The maximal extension of the solution involves an infinite tiling of consecutive patches of a black hole horizon (located at $r_h$) followed by a cosmological horizon (located at $r_c$). Gluing patches of SAdS on either side cuts off the tiling to include 2 types of symmetric inner regions: (1) including only the black hole patch (and thus only the black hole bifurcation surface) and (2) including cosmological horizon patches on either side of the black hole patch. The cosmological patches do not have singularities at $r = 0$, thus for the SAdS gluings on these patches we must have matter branes starting and ending at $r = \infty$, while for the de Sitter black hole patch we once again have branes glued at $r = 0$. The SdS solution must also be constrained by the Nairai limit, where requiring the SdS metric factor $f(r)$ has 2 positive zeroes imposes the constraint $\mu_S < \frac{2}{3\sqrt{3\lambda_{dS}}}$.

With these considerations, we begin with the allowed configurations for de Sitter black hole patches with SAdS glued on either side at $r = 0$ (see Appendix A.2 for details). The allowed configurations we obtain, as shown in Fig. 6, are identical to those for the $r = 0$ gluings in subsection 3.1. Thus the same considerations apply, and so for case 1 and 3 gluings, the RT surface can be chosen to be the SdS bifurcation surface for appropriate parameter values.

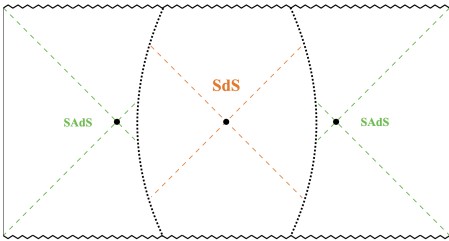

Case 1: All 3 bifurcation surfaces included, $\mu_A > \mu_S$

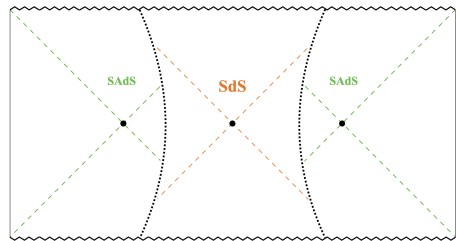

Case 2: All 3 bifurcation surfaces included, $\mu_S > \mu_A$

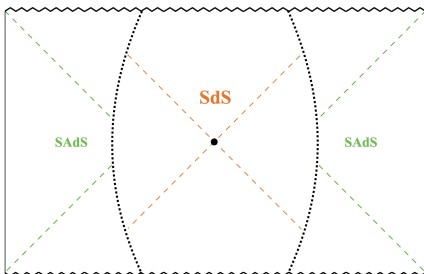

Case 3: Only S (Schwarzschild) bifurcation surface included, $\mu_A > \mu_S$

**Figure 6**. Configurations of bifurcation surfaces for $r = 0$ gluings with SdS in the middle

We additionally have gluings at $r = \infty$ in the dS vacuum patches, for which we have 2 possible configurations, which depend on whether the cosmological bifurcation surface is included or cut out. The gluing configurations again correspond to which bifurcation surfaces are included, with additional constraints now on the brane tension ($\kappa$) and de Sitter cosmological constant ($\lambda_{dS}$). Since the cosmological bifurcation surface is a local maximum along the time symmetric Cauchy slice $\sigma$, it is never the RT surface. This is also the reason we did not consider gluings with vacuum dS patches in the middle: since the dS horizon is a locally maximal surface, any gluing procedure will have geometries with the minimal surface either inside the SAdS patches or on the branes.

The allowed configurations are shown in Figs. 7 and 8. As shown by how $r$, the radial co-ordinate changes along $\sigma$, we see that we always have 3 candidate RT surfaces: the SdS black hole bifurcation surface or either one of the SAdS bifurcation surfaces. Numerical analysis indicates that we can select parameters within the Nariai limit such that the SdS black hole bifurcation surface is minimal, and thus the true RT surface. An application of the RT formula to the complete spacetime then selects the area of this surface that lies in the de Sitter patch to compute the entropy, as desired.

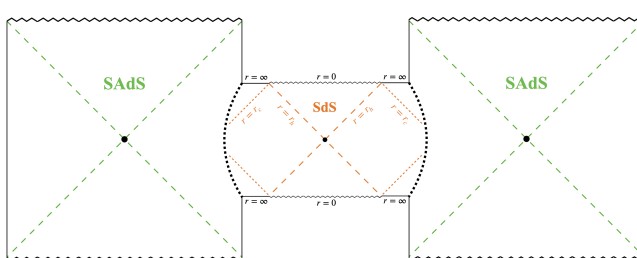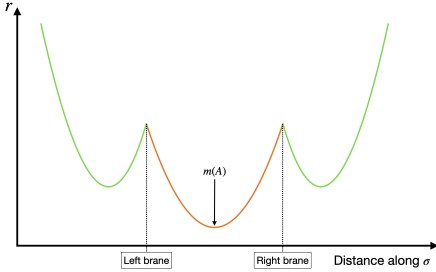

**Figure 7**. Case 4: SdS patch includes only the black hole bifurcation surface, not tyhe cosmological bifurcation surface. For small tension ($\kappa < \sqrt{\lambda_{dS} + \lambda}$), $\mu_A > \mu_S$ required for gluing. No additional constraints for large tension ($\kappa > \sqrt{\lambda_{dS} + \lambda}$).

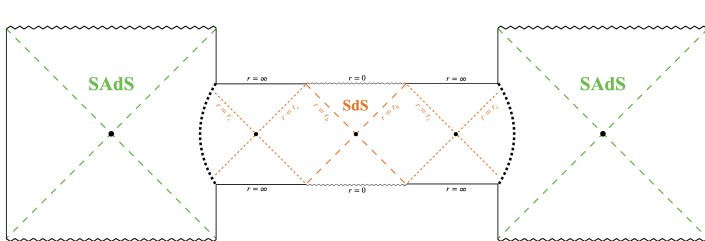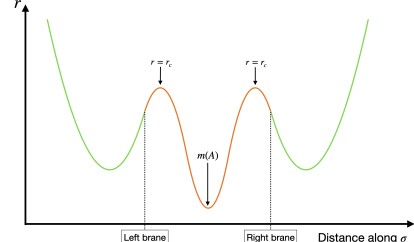

**Figure 8**. Case 5: SdS patch includes black hole and cosmological bifurcation surfaces. For large tension ($\kappa > \sqrt{\lambda_{dS} + \lambda}$), $\mu_S > \mu_A$ required for gluing. No additional constraints for small tension ($\kappa < \sqrt{\lambda_{dS} + \lambda}$). Note that the cosmological bifurcation surfaces are (locally) maximal and are thus not candidate HRT surfaces.

# 4 Entanglement entropies for Brill-Lindquist wormholes

So far we have mostly been talking theoretically about entanglement entropies in multi-boundary asymptotically Minkowski spacetimes. In this section, we will calculate entanglement entropies, using conjecture 3, in explicit examples of such spacetimes, namely four-dimensional Brill-Lindquist (BL) solutions [48]. The BL ansatz provides explicit time-symmetric initial data obeying the constraint equations for the source-free, $\Lambda = 0$ Einstein-Maxwell system, with an arbitrary number of asymptotically flat regions. Evolving this initial data to the past and future yields a time-symmetric vacuum solution connecting an arbitrary number of asymptotically Minkowski universes. These solutions are not known explicitly. Luckily, since we know that the HRT surface in a time-symmetric spacetime is the minimal surface on the symmetric slice (RT formula), we don't need the full solution to calculate entropies; the initial data is enough.

We will explore a few different BL configurations, with three and four asymptotic regions, including both neutral and charged black holes. We will also show that BL

geometries can be glued together along extremal surfaces to produce novel initial-data slices with more complicated topologies.

Unlike in the rest of the paper, throughout this section we set $G_N = 1$. The factors of $G_N$ can be recovered by multiplying all masses and entropies by $G_N$, for example by the replacements $m_i \to G_N m_i$, $S(A) \to G_N S(A)$.

## 4.1 Neutral BL ansatz

Let us start with the slightly simpler neutral BL ansatz; we will discuss the charged solutions in subsection 4.5 below. The assumption of time-reflection symmetry about the initial data slice implies that its extrinsic curvature vanishes, $K_{ab} = 0$. The constraint equation (for vacuum and $\Lambda = 0$) becomes simply $R^{(3)} = 0$ (where $R^{(3)}$ is the 3-dimensional Ricci scalar). This can be solved by a conformally flat metric,

$$ds_3^2 = \psi(\vec{r})^4 d\vec{r}^2 \,, \tag{4.1}$$

where $\vec{r}$ is a 3-dimensional vector, $d\vec{r}^2$ is the flat 3-dimensional Euclidean metric, and $\psi$ is a positive harmonic function on $\mathbf{R}^3$:

$$\nabla^2 \psi = 0 \,. \tag{4.2}$$

Asymptotic flatness for large $r$ implies that $\psi \to 1$ there. The unique solution to (4.2) is then $\psi = 1$. To get something more interesting, we remove $n - 1$ points at $\vec{r} = \vec{r}_i$ $(i = 1, \ldots, n - 1)$ from $\mathbf{R}^3$, allowing $\psi$ to blow up at those points:[4]

$$\psi = 1 + \sum_{i=1}^{n-1} \frac{\alpha_i}{\|\vec{r} - \vec{r}_i\|} \,, \tag{4.3}$$

where $\alpha_i$ are arbitrary positive coefficients (with units of length) and $\| \cdot \|$ denotes the 3-dimensional Euclidean norm. The change of coordinates

$$\vec{r}' = \alpha_i^2 \frac{\vec{r} - \vec{r}_i}{\|\vec{r} - \vec{r}_i\|^2} \tag{4.4}$$

shows that the region close to each puncture, $\|\vec{r} - \vec{r}_i\| \ll \alpha_i$, is also asymptotically flat, so this metric has $n$ asymptotically flat regions. In the case $n = 2$, the metric reduces to the initial data for the Schwarzschild metric of radius $\alpha_1$ (in isotropic coordinates, including both exterior regions); the bifurcation surface is the sphere $\|\vec{r} - \vec{r}_1\| = \alpha_1$.

---

[4]In the notation of [48], this corresponds to the case $\beta_i = \alpha_i$ and $\chi = \psi$, with $N = n - 1$.

The ADM mass $m_i$ measured in the asymptotic region $a_i$ (where $\vec{r} \to \vec{r_i}$) is computed by matching the first-order expansion of the metric near $\vec{r} = \vec{r_i}$ to the Schwarzschild mass term, giving:

$$m_i = 2\alpha_i \left( 1 + \sum_{\substack{j \neq i}}^{n-1} \frac{\alpha_j}{\|\vec{r_i} - \vec{r_j}\|} \right) \qquad (i = 1, \ldots, n-1). \tag{4.5}$$

For the ADM mass $m_n$ for the asymptotic region $a_n$ (where $\|\vec{r}\| \to \infty$), we similarly match the first-order expansion to the Schwarzschild mass term, giving:

$$m_n = 2 \sum_{i=1}^{n-1} \alpha_i. \tag{4.6}$$

With the following alternative ansatz,

$$ds^2 = \phi(\vec{s})^4 d\vec{s}^2, \qquad \phi(\vec{s}) = \sum_{i=1}^{n} \frac{\mu_i}{\|\vec{s} - \vec{s_i}\|}, \tag{4.7}$$

the $n$ asymptotic regions are treated democratically. The absence of a "1+" in the expression for $\phi$ means that the region $s \to \infty$ is not asymptotic; rather, the point $s = \infty$ is at a finite distance, so we should append it to $\mathbf{R}^3$, giving an $n$-punctured 3-sphere. The mass measured in the asymptotic region $a_i$ is given by

$$m_i = 2\mu_i \sum_{\substack{j \neq i}}^{n} \frac{\mu_j}{\|\vec{s_i} - \vec{s_j}\|} \qquad (i = 1, \ldots, n). \tag{4.8}$$

We call these "inverted coordinates", as they can be related to the $\vec{r}$ coordinates by the inversion $\vec{s} = \mu_n^2 \vec{r}/r^2$, where $\mu_n$ is an arbitrary length and the parameters in the two ansatzes are related by

$$\frac{\mu_i}{\mu_n} = \frac{\alpha_i}{r_i}, \qquad \frac{\vec{s_i}}{\mu_n^2} = \frac{\vec{r_i}}{r_i^2} \qquad (i = 1, \ldots, n-1) \tag{4.9}$$

and $\vec{s_n} = \vec{0}$.

## 4.2   Minimal surfaces for $n = 3$

Interestingly, already in their 1963 paper, Brill-Lindquist investigated minimal surfaces in their metrics, finding a transition in the topology of the minimal surfaces as the separation between the points $\vec{r_i}$ is varied. Specifically, they focused on the case $n = 3$ with equal parameters $\alpha_1 = \alpha_2 =: \alpha$. For each puncture $\vec{r}_{1,2}$, there is a minimal surface

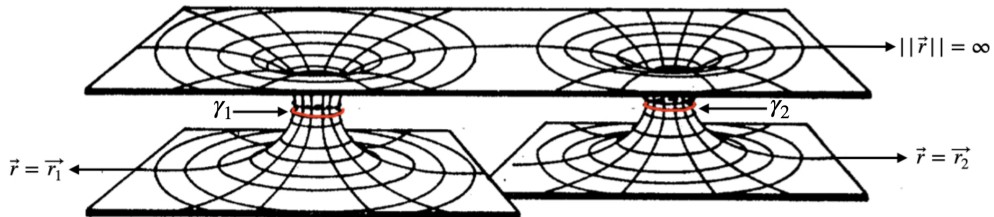

**Figure 9**. Embedding diagram of BL initial data geometry for $n = 3$, $\alpha_1 = \alpha_2 =: \alpha$, and large separation, $\|\vec{r}_1 - \vec{r}_2\| \gg \alpha$. There are two Einstein-Rosen bridges and two minimal surfaces $\gamma_1$, $\gamma_2$. (Figure adapted from [48].)

$\gamma_{1,2}$ of spherical topology enclosing that puncture; this is the throat of the corresponding Einstein-Rosen bridge.

For large separation $r_{12} := \|\vec{r}_2 - \vec{r}_1\| \gg \alpha$, $\gamma_{1,2}$ are the only minimal (or extremal) surfaces; as $r_{12}$ is increased, they approach the spheres $\|\vec{r} - \vec{r}_i\| = \alpha$ (see Fig. 9). The minimal surface homologous to the sphere at infinity in region $a_n$ is $\gamma_1 \cup \gamma_2$. An observer living in that region sees a pair of black holes, which under time evolution collide and coalesce. (Indeed, BL initial data is often used for simulating black hole collisions [62, 63].)

As we reduce the separation $r_{12}$, there is a critical separation $r_{12}^c \approx 3.1\alpha$ [64] at which a new (locally) minimal surface $\gamma_3$ appears, with spherical topology and surrounding both punctures. The surfaces $\gamma_{1,2}$ are hidden behind $\gamma_3$, so the observer in region $a_n$ sees only one black hole. The geometry on the slice now looks qualitatively like a three-boundary AdS wormhole, in the sense that each asymptotic region has a neck separating it from a central region (see Fig. 10). The three asymptotic regions are now in a sense on equal footing. In fact, for $r_{12} = \alpha$, we have $m_1 = m_2 = m_3 = \alpha$, and there is a $S_3$ group of isometries permuting the three asymptotic regions; this symmetry can be made manifest in the inverted coordinates, with the punctures lying at the vertices of an equilateral triangle and the parameters $\mu_i$ all equal (see Figs. 11, 12, and Appendix B.1 for details).

These observations by Brill-Lindquist already tell us a lot about the entanglement entropies for this system, given conjecture 3. The RT surface for universe 1 is $\gamma_1$, and similarly for universe 2. For universe 3, if $r_{12} > r_{12}^c$, then the RT surface is $\gamma_1 \cup \gamma_2$. This is also the RT surface for its complement, the union of universes 1 and 2, implying that the mutual information vanishes, $I(1 : 2) = 0$. On the other hand, if $r_{12} < r_{12}^c$, then there is another candidate surface $\gamma_3$.

To go further, we need to know the areas of these surfaces: if $|\gamma_3| < 2|\gamma_1|$ then the RT surface jumps to $\gamma_3$ and $I(1 : 2) > 0$. To compute the areas of these surfaces, we

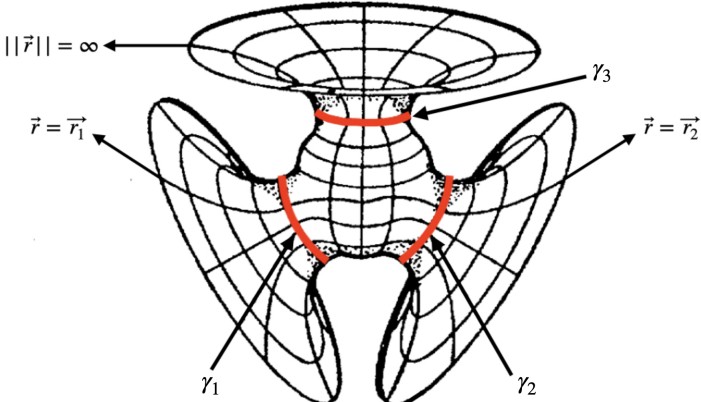

**Figure 10**. Embedding diagram of BL initial data geometry for equal masses and small separation, $r_{12} \ll \alpha$. There are three Einstein-Rosen bridges and three minimal surfaces $\gamma_{1,2,3}$. (Figure taken from [48].)

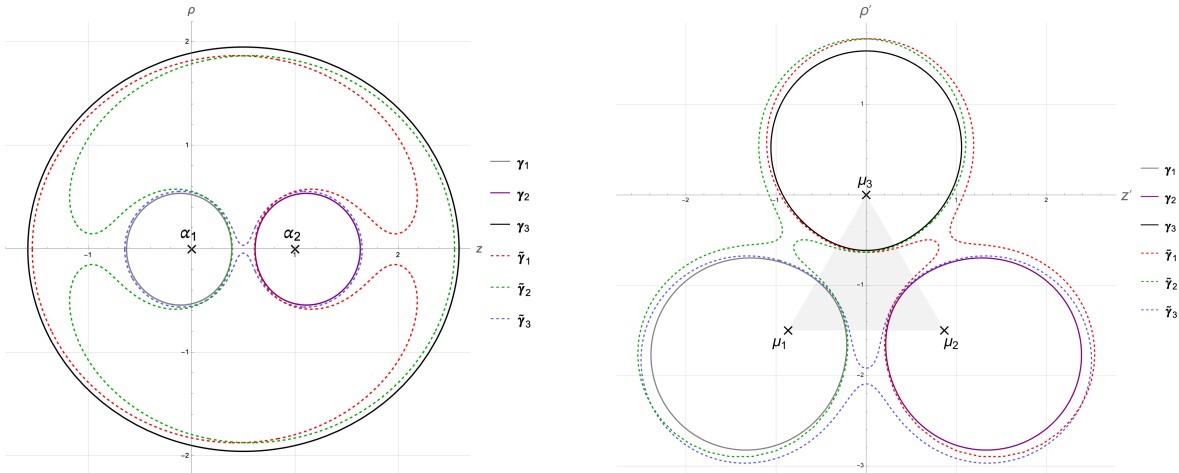

**Figure 11**. Planar sections of extremal surfaces in the $S_3$-symmetric BL metric, in the original $\vec{r}$ (left) and inverted $\vec{s}$ (right) coordinates. Solid curves are minimal surfaces $\gamma_{1,2,3}$. Dashed curves are index-1 extremal surfaces $\tilde{\gamma}_1, \tilde{\gamma}_2, \tilde{\gamma}_3$.

solved the minimal surface equation numerically using a shooting method; see Appendix B.2 for details. The areas of these surfaces are shown in Fig. 13. The surfaces themselves are shown for the $S_3$-symmetric case in Figs. 11 and 12; in this case, we find

$$I(1:2) = \frac{1}{2}|\gamma_1| - \frac{1}{4}|\gamma_3| = \frac{1}{4}|\gamma_1| \approx 64\pi r_{12}^2 \,. \tag{4.10}$$

General considerations from Morse theory suggest that, under smooth deformations of the metric, extremal surfaces in a given homology class should typically appear or

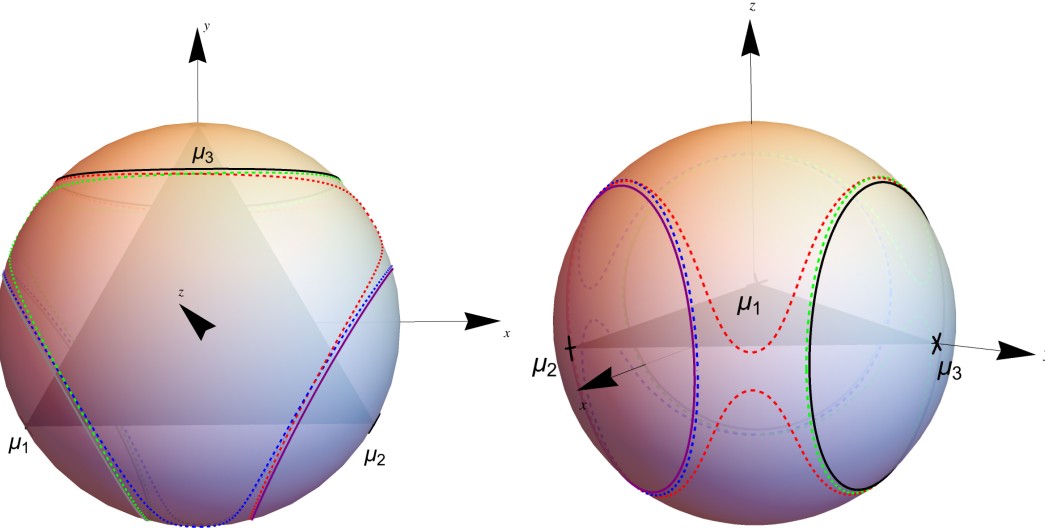

**Figure 12**. Two views of a stereographic projection to the sphere of the extremal surface sections shown in Fig. 11 in inverted coordinates. The masses are at the vertices of an equilateral triangle circumscribed in the sphere.

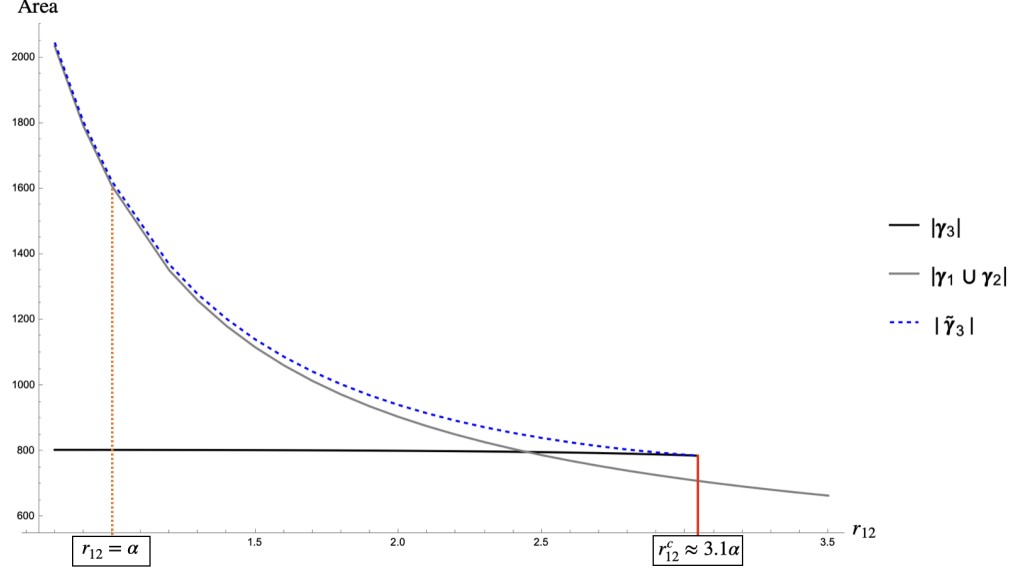

**Figure 13**. Areas of extremal surfaces as a function of $r_{12}$ for $\alpha = 1$. Note the swallowtail behavior at the critical separation $r_{12}^c$, where $\gamma_3$, $\tilde{\gamma}_3$ appear/disappear together. $S_3$ isometry is present for $r_{12} = \alpha$. While the area of $\gamma_3$ appears constant, it is actually slowly varying.

disappear in pairs, with Morse index differing by 1, the Morse index of an extremal surface being the number of independent small deformations of the surface that reduce its area at second order. We saw above that the locally minimal surface $\gamma_3$ appeared as the separation $r_{12}$ was decreased past the critical value $r_{12}^c$. We should therefore expect an index-1 surface to also appear at the same value of $r_{12}$. Indeed there is a second surface $\tilde{\gamma}_3$ appearing at $r_{12}^c$, as shown in Figs. 11, 12 and 13. While we have not checked directly that $\tilde{\gamma}_3$ has index 1, a narrow neck such as it exhibits is typically associated with a negative mode; in the vicinity of the neck it can be approximated by a catenoid in flat space, which indeed has one negative mode. This surface would play the role of the bulge surface in the python's lunch conjecture [40], controlling the complexity of reconstructing the region between the locally minimal surfaces $\gamma_3$ and $\gamma_1 \cup \gamma_2$.[5] By symmetry, there are also index-1 surfaces $\tilde{\gamma}_1, \tilde{\gamma}_2$ homologous to $\gamma_1, \gamma_2$ respectively (shown in Figs. 11, 12).

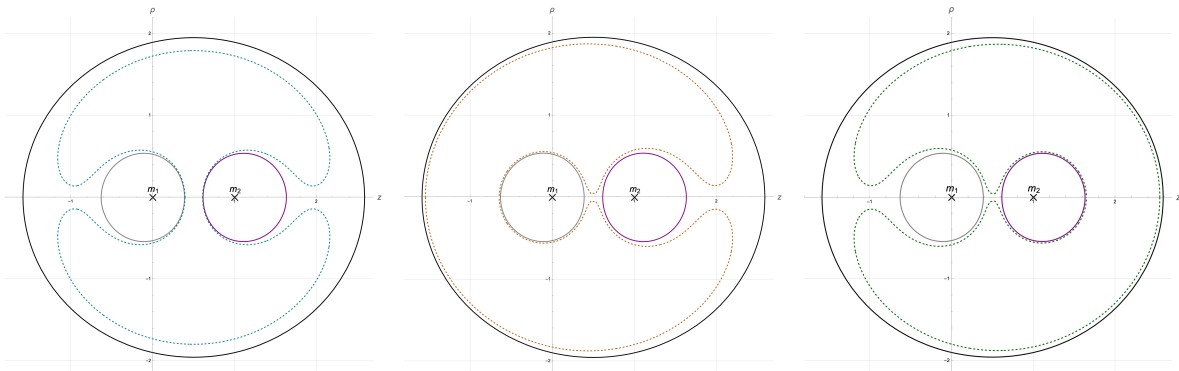

**Figure 14**. Index 2 extremal surfaces (dashed curves) in different homotopy classes

One can also find higher-index surfaces. For example, Fig. 14 shows null-homologous surfaces that possess two necks and therefore likely have index 2. The physical interpretation, if any, of such surfaces is not known. A complete classification of all extremal surfaces in BL geometries is also not known. Numerical searches to classify these surfaces have suggested that the number may even be infinite [65–68].

### 4.3   Minimal surfaces for $n = 4$

We will now briefly discuss extremal surfaces for the case $n = 4$. For concreteness, we will specialize to a highly symmetric configuration where, in inverted coordinates, the punctures lie at the vertices of a square and have equal parameters $\mu_i$ (see Appendix B.1 for details). This configuration has a dihedral $D_4$ isometry group. It can be described

---

[5]For a detailed discussion of the Morse index and bulge surfaces, see [43].

in the original BL coordinates with three collinear punctures with $\vec{r}_2$ centered between $\vec{r}_1$ and $\vec{r}_3$ and $\alpha_1 = \alpha_3 = 2^{1/2} r_{12}$, $\alpha_2 = r_{12}$.

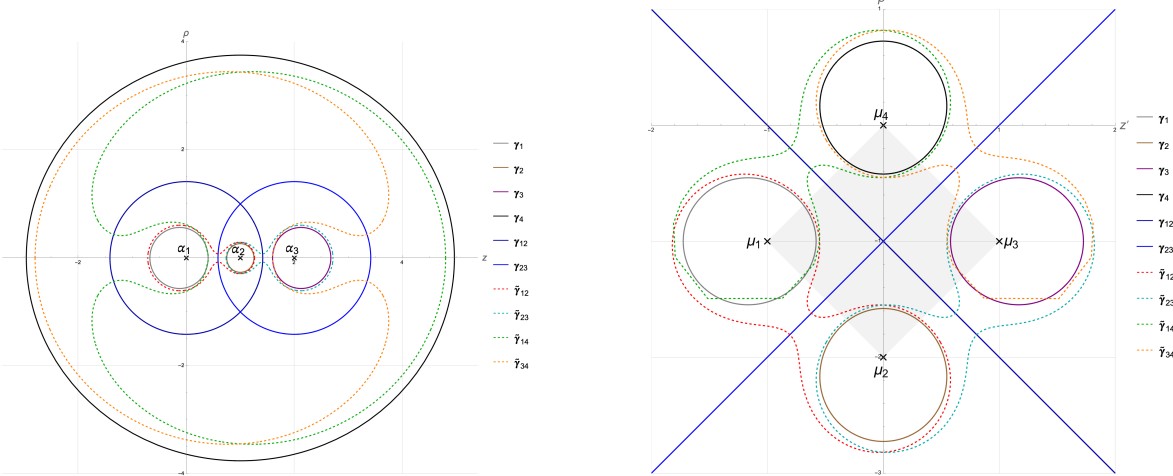

**Figure 15**. Planar sections of extremal surfaces in original (left) and inverted (right) coordinates. Solid curves are minimal surfaces; dashed curves are index-1 surfaces. The minimal surfaces $\gamma_{12}$ and $\gamma_{23}$ extend to infinity in the inverted coordinates. (The apparent intersections of $\tilde{\gamma}_{14}$ and $\tilde{\gamma}_{34}$ with $\gamma_1$ and $\gamma_3$ respectively are numerical artifacts.)

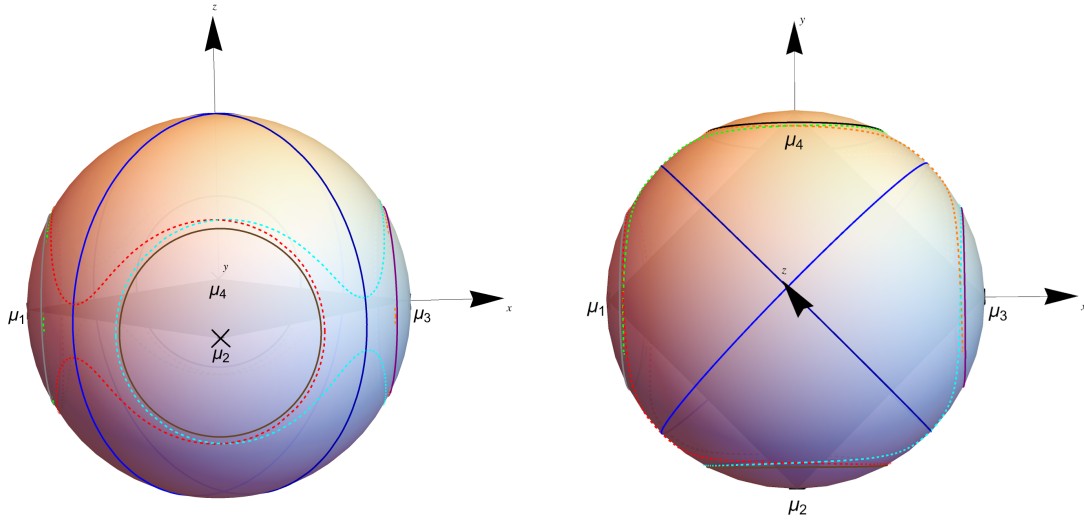

**Figure 16**. Two views of a stereographic projection to the sphere of the extremal surface sections shown in Fig. 15 in inverted coordinates. The masses are at the vertices of a square inscribed in the sphere.

We find six minimal surfaces: $\gamma_i$ $(i = 1, \ldots, 4)$ separates universe $i$ from the three others; $\gamma_{12}$ separates $1 \cup 2$ from $3 \cup 4$; and $\gamma_{23}$ separates $2 \cup 3$ from $1 \cup 4$ (see Figs. 15, 16). On the other hand, there is no minimal surface separating $1 \cup 3$ from $2 \cup 4$, or $1 \cup 3$ from $2 \cup 4$. Their areas are

$$|\gamma_i| \approx 938\pi r_{12}^2 \,, |\gamma_{12}| = |\gamma_{23}| \approx 1459\pi r_{12}^2 \,. \tag{4.11}$$

From these facts, we can deduce the RT surface for each set of universes. The RT surface for universe $i$ is $\gamma_i$. Since $|\gamma_{12}| < 2|\gamma_1|$, the RT surface for $1 \cup 2$ is $\gamma_{12}$; hence universes 1 and 2 have a non-zero mutual information:

$$I(1:2) = \frac{1}{2}|\gamma_1| - \frac{1}{4}|\gamma_{12}| \approx 104\pi r_{12}^2 \,. \tag{4.12}$$

(This also goes for $I(2:3)$, $I(3:4)$, and $I(4:1)$.) On the other hand, for $1 \cup 3$, the candidate RT surfaces are $\gamma_1 \cup \gamma_3$ and $\gamma_2 \cup \gamma_4$; these have equal area, so either one can be considered the RT surface. Either way, we find $I(1:3) = 0$. (The same goes for $2 \cup 4$.)

As we found for $n = 3$, there are also index-1 surfaces. In the region between $\gamma_1 \cup \gamma_2$ and $\gamma_{12}$, and in the same homology class as those surfaces, there is an index-1 surface $\tilde{\gamma}_{12}$ that, according to the python's lunch conjecture, serves as the bulge surface computing the complexity of reconstructing that region of the bulk (similarly for the other three consecutive pairs).[6] These surfaces are also shown in Figs. 15, 16.

## 4.4 Gluing BL spacetimes

We can create (initial data for) spacetimes with more complicated topologies by cutting and gluing BL metrics along minimal surfaces. As a first example, let $\Sigma$ be the $n = 3$ BL geometry with $\alpha_1 = \alpha_2$, as discussed in subsection 4.2. This geometry admits an isometric reflection that exchanges punctures 1 and 2, and surfaces $\gamma_1$ and $\gamma_2$. We can remove from $\Sigma$ the interior of these two surfaces (thereby removing universes 1 and 2) and identify them to make a surface $\gamma'$ (see Fig. 17). This results in a manifold $\Sigma'$ with just one asymptotic region, $a_3$, and a non-trivial fundamental group, $\pi_1(\Sigma') = \mathbf{Z}$.

The metric on $\Sigma'$ is continuous but not smooth. In particular, since $\gamma_1$ and $\gamma_2$ have non-zero extrinsic curvature, the extrinsic curvature of $\gamma'$, computed on its two sides, differs by a sign. By the Gauss-Codazzi equation, the (three-dimensional) Ricci tensor of $m'$ has a delta-function on $\gamma'$, proportional to the jump in the extrinsic curvature. However, because the surfaces $\gamma_{1,2}$ are minimal, their extrinsic curvatures are traceless, so the discontinuity in the extrinsic curvature of $\gamma'$ is also traceless, so the

---

[6]It should be noted this is not a complete classification of the index-1 surfaces. Some of these additional surfaces play a role in the black hole evaporation model proposed in [49]

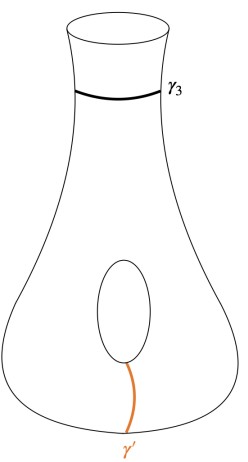

**Figure 17**. Gluing of $n = 3$ BL geometry along surfaces $\gamma_1$ and $\gamma_2$ to make surface $\gamma'$.

Ricci scalar of $\Sigma'$ vanishes on $\gamma'$ (and everywhere else of course). Hence $\Sigma'$ obeys the vacuum constraint equations, and is valid (albeit singular) initial data. Its time-evolved spacetime $M'$ includes a gravitational shock propagating away from $\gamma'$, to the past and future, in both spatial directions. This shock is hidden behind the event horizon(s), so is invisible to an observer in the remaining asymptotic region $a_3$, who would not notice any difference between $M'$ and the time evolution $M$ of $\Sigma$. However, observers willing to jump into the black holes would notice a difference, at least in the regime of large separation where there is no surface $\gamma_3$ and so there are initially two distinct black holes. Specifically, observers who jump into different black holes can meet in $M'$ (assuming they survive crossing the shock) but not in $M$.

According to conjecture 3, $M'$ represents a set of states in the Hilbert space $\mathcal{H}^{a_3}$, and these states have vanishing entropy (at order $G_N^{-1}$). They are therefore microstates of the black hole(s) seen in region $a_3$. In the case where $\vec{r}_1$ and $\vec{r}_2$ are close enough that a surface $\gamma_3$ exists, then this surface is a constriction, and the index-1 surface $\tilde{\gamma}_3$ is the bulge surface, indicating, according to the python's lunch conjecture, an exponential complexity for reconstructing observables behind $\gamma_3$. An intriguing question is whether the extremal surface $\gamma'$, which is homologically non-trivial, and indeed minimal in its homology class, but not homologous to any boundary region, represents entanglement in any sense. In the limit that $\gamma_3$ is much smaller than $\gamma'$, we can also think of $M'$ as representing a small black hole inside a closed universe with spatial topology $S^2 \times S^1$; in the spirit of [69], we can think of this black hole as modelling an observer in the closed universe, with Hilbert space dimension roughly $e^{|\gamma_3|/4G_N}$, and $\mathcal{H}^{a_3}$ as an auxiliary Hilbert space that purifies this observer.

This construction can be extended to any two minimal surfaces related by a reflection (i.e. an isometry that maps the excised parts of the space to each other). Thus, if a BL wormhole with $n$ asymptotic regions has a reflection symmetry that relates two of them, they can be excised and the respective minimal surfaces glued together to create a wormhole with $n-2$ asymptotic regions and fundamental group $\mathbf{Z}$. For example, if an $n=4$ wormhole $\Sigma$ has a reflection symmetry that exchanges $\gamma_1$ and $\gamma_2$, the result is a wormhole $\Sigma'$ connecting the universes 3 and 4. Their respective RT surfaces in $\Sigma$, $\gamma_3, \gamma_4$, find themselves in the same homology class in $\Sigma'$; whichever one is smaller is now the RT surface for both 3 and 4. Hence, in passing from $\Sigma$ to $\Sigma'$, the entropy of one of the remaining universes has decreased, while the other has stayed the same.

We can also glue copies of a BL geometry to itself. For example, let $\Sigma$ and $\tilde{\Sigma}$ be BL geometries whose punctures are related by a reflection through some plane (in either the original or inverted coordinates). They may be glued together along any pair of minimal surfaces that are images of each other under the reflection, giving a new wormhole $\Sigma'$. This is essentially the same construction used to construct canonical purifications holographically, for example, for the purpose of computing the reflected entropy [33, 36]. We may therefore conjecture that $\Sigma'$ represents the canonical purification of the reduced density matrix for the remaining (unexcised) universes of $\Sigma$.

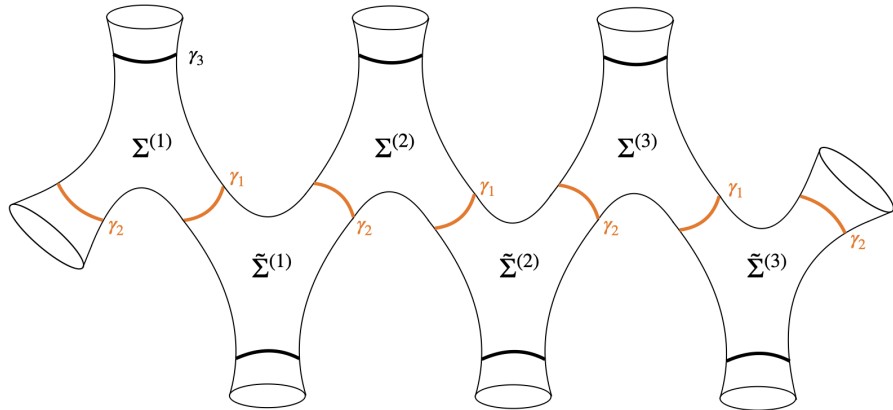

**Figure 18**. Gluing copies of $n=3$ BL geometries to form a chain; the surfaces $\gamma_1$ on each end can also be glued together to form a closed chain.

We may continue gluing more copies of $\Sigma$ and $\tilde{\Sigma}$ together to make arbitrarily complicated networks of universes. As a relatively simple example, suppose that $\Sigma$ is an $n=3$ wormhole. We can glue a copy $\Sigma^{(1)}$ of $\Sigma$ to a copy $\tilde{\Sigma}^{(1)}$ of $\tilde{\Sigma}$ along, say, $\gamma_1$. This gives a 4-boundary wormhole. Then we can glue a second copy $\Sigma^{(2)}$ of $\Sigma$ to $\tilde{\Sigma}^{(1)}$ along $\gamma_2$, giving a 5-boundary wormhole. Then we can glue a second copy $\tilde{\Sigma}^{(2)}$ of $\tilde{\Sigma}$ to

$\Sigma^{(2)}$ along $\gamma_1$, giving a 6-boundary wormhole. (See Fig. 18.) We can continue in this fashion for as long as we want, making a wormhole that is essentially a linear chain, with two universes at each end and one on each internal link. If the chain has an equal number of $\Sigma$ and $\tilde{\Sigma}$ links, then we can close it up to make a circle by gluing the last $\tilde{\Sigma}$ to the first $\Sigma$ along $\gamma_2$. Being very simple, the entanglement entropies are easy to calculate for either the open or closed chain, given the minimal surface areas in the original manifold $\Sigma$. We would like to point out an interesting feature that gives a clue about the entanglement structure of the state represented by this spacetime. Namely, as the reader may easily check, the mutual information $I(A : B)$ between any two sets of universes $A, B$ that are *not* adjacent in the chain vanishes. This means that the state lacks long-range entanglement and is a Markov chain.

A more extreme construction would remove all asymptotic regions. For example, starting from a symmetric $n = 4$ BL geometry, we can glue $\gamma_1$ to $\gamma_2$ and $\gamma_3$ to $\gamma_4$. This leaves a closed universe; under time evolution it evolves into a singularity in both the past and future, yielding a big bang-big crunch cosmology. Taking conjecture 3 down to $n = 0$, this spacetime represents a set of states in a zero-fold tensor product Hilbert space, in other words, in a 1-dimensional Hilbert space (so the "set of states" is just one state). This agrees with the common lore that closed universes have a 1-dimensional Hilbert space.

## 4.5 Charged solutions

The original Brill-Lindquist metric is written for charged black holes. Here, instead of the conformal factor $\psi^4$ we consider the factor $(FG)^2$, where:

$$F = 1 + \sum_{i=1}^{n-1} \frac{f_i}{\|\vec{r} - \vec{r_i}\|} \tag{4.13}$$

$$G = 1 + \sum_{i=1}^{n-1} \frac{g_i}{\|\vec{r} - \vec{r_i}\|} \tag{4.14}$$

The uncharged solution is recovered for $f_i = g_i = \alpha_i$. The expressions for the ADM masses are now:

$$m_i = f_i \left(1 + \sum_{j \neq i}^{n-1} \frac{g_j}{\|\vec{r_i} - \vec{r_j}\|}\right) + g_i \left(1 + \sum_{j \neq i}^{n-1} \frac{f_j}{\|\vec{r_i} - \vec{r_j}\|}\right) \tag{4.15}$$

$$m_n = \sum_{i=1}^{n-1} (f_i + g_i) \tag{4.16}$$

where Eq. 4.5 and 4.6 are recovered for $f_i = g_i = \alpha_i$.

The charge on each sheet can be found by evaluating the flux of the electric field $E_i \equiv [\ln (F/G)]_{,i}$ through a sphere:

$$q_i = \frac{1}{4\pi} \int E_i n^i dS \tag{4.17}$$

$$= g_i \left( 1 + \sum_{j \neq i}^{n-1} \frac{f_j}{\|\vec{r_i} - \vec{r_j}\|} \right) - f_i \left( 1 + \sum_{j \neq i}^{n-1} \frac{g_j}{\|\vec{r_i} - \vec{r_j}\|} \right) \tag{4.18}$$

where $dS = (FG)^2 r^2 \sin (\theta) d\theta d\phi$ is the spherical area element. By conservation of flux, the charge on the asymptotic infinity sheet is $q_n = \sum_{i=1}^{n-1} q_i$.

Inverting the formulae for $m_i, q_i$ to obtain formulae for $f_i, g_i$ is difficult to do exactly, thus [48] gives the inverse formulae to leading order in $1/\|\vec{r_i} - r_j\|$:

$$f_i \approx \frac{1}{2}(m_i - q_i) \left( 1 - \frac{1}{2} \sum_{i \neq j}^{n-1} \frac{m_j + q_j}{\|\vec{r_i} - \vec{r_j}\|} \right) \tag{4.19}$$

$$g_i \approx \frac{1}{2}(m_i + q_i) \left( 1 - \frac{1}{2} \sum_{i \neq j}^{n-1} \frac{m_j - q_j}{\|\vec{r_i} - \vec{r_j}\|} \right) \tag{4.20}$$

Here, it's clear to see that the regularity condition Brill and Lindquist impose ($f_i, g_i > 0$) ensures all black holes are sub-extremal. Moreover, near-extremal black holes can be considered by taking either $f_i$ or $g_i$ to be very small (depending on the sign of the charge $q_i$). We can understand how the metric behaves in this limit by starting with an uncharged solution ($f_i = g_i$) and decreasing $f_i$ to 0 (extremal limit). This limit is important in the context of holography, since the exact quantum mechanical descriptions of certain BPS black holes/branes are under more control.

As shown in Fig. 19, in the extremal limit there also exists a critical point where the outer minimal surface disappears. Approaching the extremal limit, the minimal surfaces are pushed closer to the singularities, which, as we recall, are simply images of infinity. Thus, the BL wormhole is stretching to infinite proper length, with the minimal surfaces being pushed infinitely far away, which is expected from extremally charged wormholes. It should also be noted that in this extremal limit, the BL metric actually approaches the initial time slice of the Majumdar-Papapetrou (MP) geometry, which is a static solution of spherically symmetric, extremal black holes. It is given by the metric [70],

$$ds^2 = -\frac{dt^2}{U(\vec{x})^2} + U(\vec{x})^2 \left( dr^2 + r^2 d\Omega^2 \right) \tag{4.21}$$

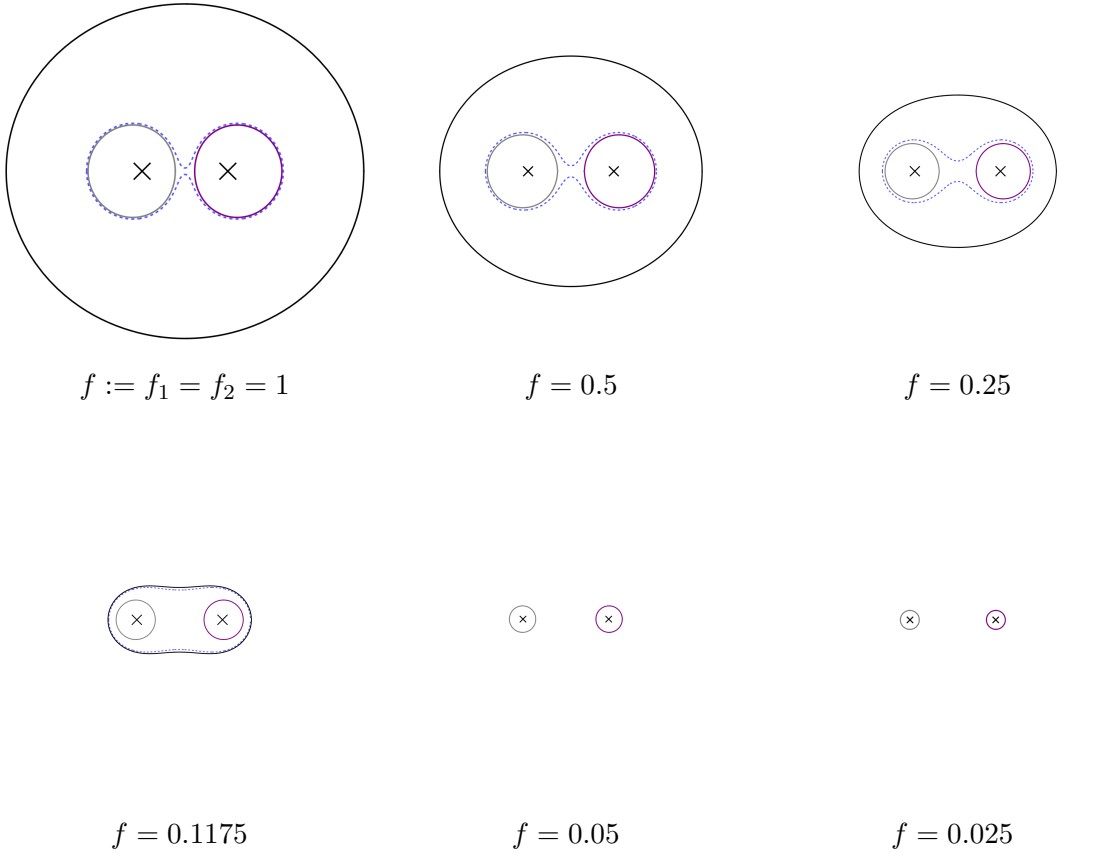

$$f := f_1 = f_2 = 1 \qquad\qquad f = 0.5 \qquad\qquad f = 0.25$$

$$f = 0.1175 \qquad\qquad f = 0.05 \qquad\qquad f = 0.025$$

**Figure 19**. Extremal surfaces for $n = 3$ charged Brill-Lindquist solution, with $g_1 = g_2 = 1$ and separation $r_{12} = 1$ fixed. As $f_1 = f_2 =: f \to 0$, the ADM mass and charge $m_i, q_i \to g_i$

where $U(\vec{x})$ is similar to the BL conformal factor and for $n - 1$ point masses is given by:

$$U(\vec{x}) = 1 + \sum_{i=1}^{n-1} \frac{m_i}{\|\vec{r} - \vec{r_i}\|} \,. \tag{4.22}$$

Computing the electric flux through a sphere around each mass, we also obtain $q_i = m_i$ i.e. all black holes are extremal (and positively charged). The initial time slice of this metric then has a wormhole geometry, with throats of area $\mathcal{A}_i = 4\pi m_i^2$ around each singularity. Taking the BL limit as above ($f_i \to 0$) we have the conformal factor $F \to 1$; thus the initial time slice of MP is equivalent to the extremal BL metric, with the conformal factor $G$ identified with $U$. To see that this holds, we can also consider the areas of the BL minimal surfaces in the extremal limit as shown in Fig. 20. The ADM masses $m_1, m_2 \to 1$ and so the area asymptotes to the finite value $4\pi m_1^2$, as expected for the MP geometry.

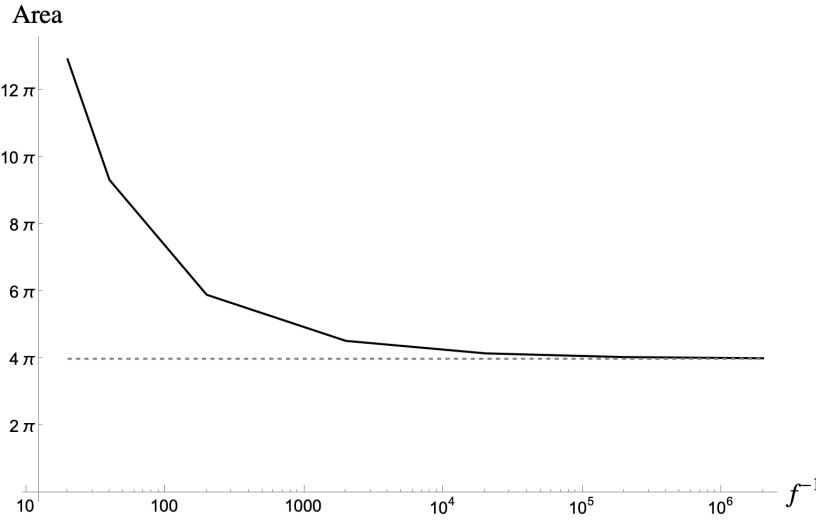

**Figure 20**. Area of black hole of mass $m_1$ in the extremal limit shown in Fig. 19

It should be noted that taking the $n = 3$ BL metric in this extremal limit ($f \to 0$), there are three extremal black holes but only two minimal surfaces. Since the wormhole is becoming longer as we approach the extremal limit, the outer minimal surface disappears at some point as it is pushed "below" the two inner minimal surfaces, since, as shown in Fig. 19, the outer minimal surface seems to approach and "collide" with the inner surfaces. Thus there are only two minimal surfaces that can be seen in the BL geometry due to the lengthening of the wormhole; nevertheless, there exist three extremal black holes.

In this limit, the charged BL metric can thus be thought of as a generalization of MP on the initial-time slice, allowing for non-extremal charged solutions. As mentioned before, these extremal/near-extremal black holes have known duals in certain supergravity theories, thus possibly allowing for a direct computation of the entanglement entropy and thus a direct verification of the RT formula in flat spacetimes. However, this is beyond the scope of this paper.

Another interesting configuration that can be obtained from the charged BL metric is shown in Fig. 21, where we consider a dipole geometry. The two singularities have equal and opposite charge, and thus the 3rd black hole (corresponding to the outer minimal surface) is uncharged. For $n = 3$, we set $q_1 = -q_2 =: q$ and $m_1 = m_2 =: m$ to

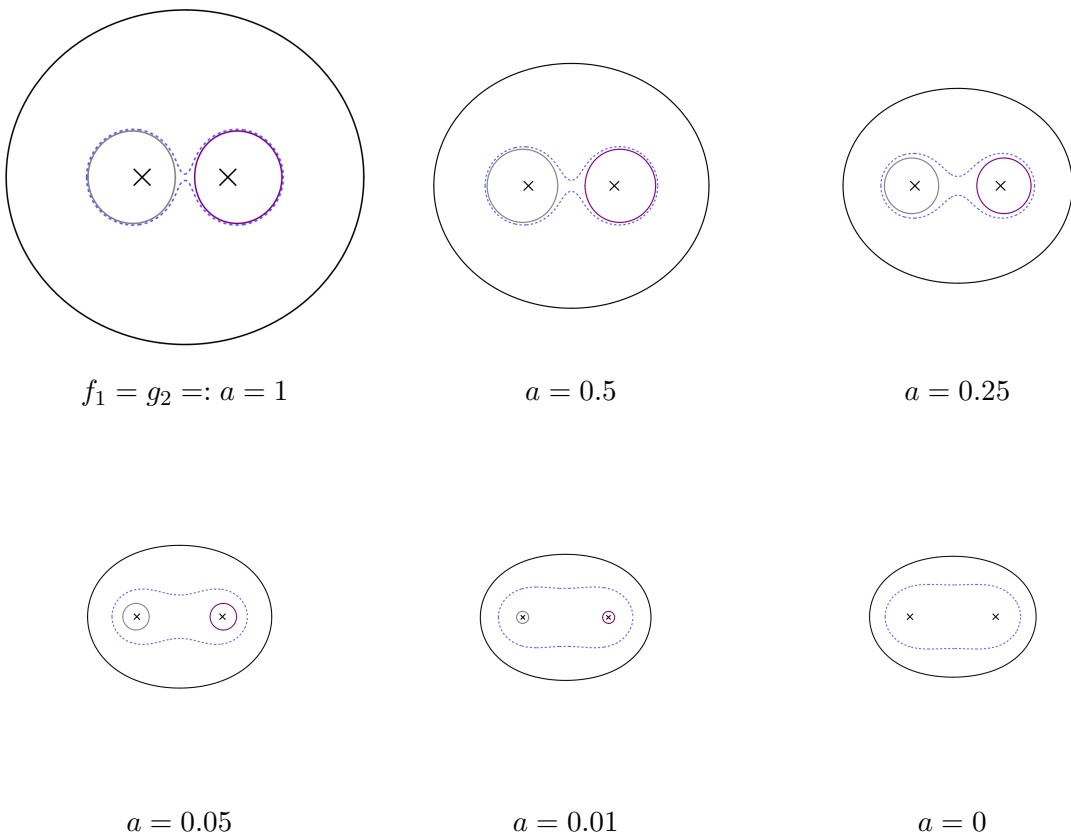

**Figure 21**. Extremal surfaces for dipole Brill-Lindquist solution, with $g_1 = g_2 =: b = 1$ and separation $r_{12} = 1$ fixed. As $a \to 0$, the ADM mass and charge $m, q \to b + \frac{b^2}{r_{12}} = 2$

obtain,

$$f_1 = g_2 =: a \tag{4.23}$$

$$f_2 = g_1 =: b \tag{4.24}$$

$$\implies q = b - a + \frac{b^2 - a^2}{r_{12}} \tag{4.25}$$

$$m = a + b + \frac{a^2 + b^2}{r_{12}} \ . \tag{4.26}$$

As seen in Fig. 21, the outer minimal surface no longer disappears in the extremal limit. In fact, along with the extremal index 1 surface (dotted blue line) it asymptotically approaches a finite area (the geometry is shown as the last snapshot) while the inner minimal surfaces are pushed to infinity. Similar to the previous limit, the areas of the charged black holes asymptotically approach finite values in the extremal limit,

shown in Fig. 22.

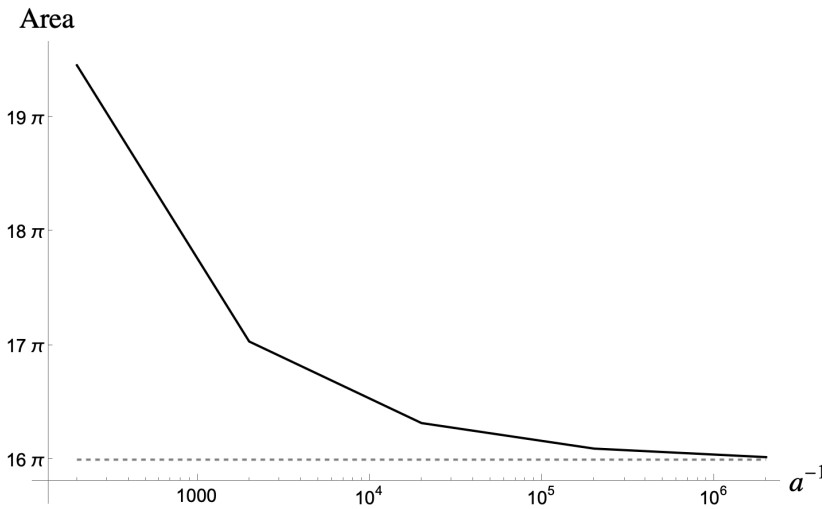

**Figure 22**. Area of black hole of mass $m_1$ in the extremal limit shown in Fig. 21 $((a \to 0))$

The mass $m \to 2$, which once again implies the relation $|\gamma| \to 4\pi m^2$ for both (extremal) black holes. This area is again consistent with the horizon area in MP, even though MP does not allow for negatively charged black holes. Such dipole configurations thus offer a novel near-extremal geometry, which may similarly admit exact CFT duals.

## 5    De Sitter asymptotics

De Sitter space is famously confusing in quantum gravity. It is not even known with any certainty whether there exist quantum gravity theories with dS vacua [71–75]. Even if, as has been conjectured, there do not exist absolutely stable dS vacua, there likely exist long-lived ones (such as the ones our universe seems to have been in during inflation and seems to be in now). Either way, we can consider spacetimes with dS asymptotic regions, although in the unstable case these would not be literally asymptotic but would decay at some very large time. The conformal boundary of such an asymptotic region is spacelike, forming either a past or future boundary of the spacetime. Our purpose in this section will be to explore possible consistent generalizations of the HRT formula and conjecture 3 to such spacetimes, and see what insight can be gained from them. We will find that multiple distinct generalizations are possible, with different interpretations.

In the first subsection below, we focus on a particularly simple class of spacetimes, before turning in the next subsection to the general case.

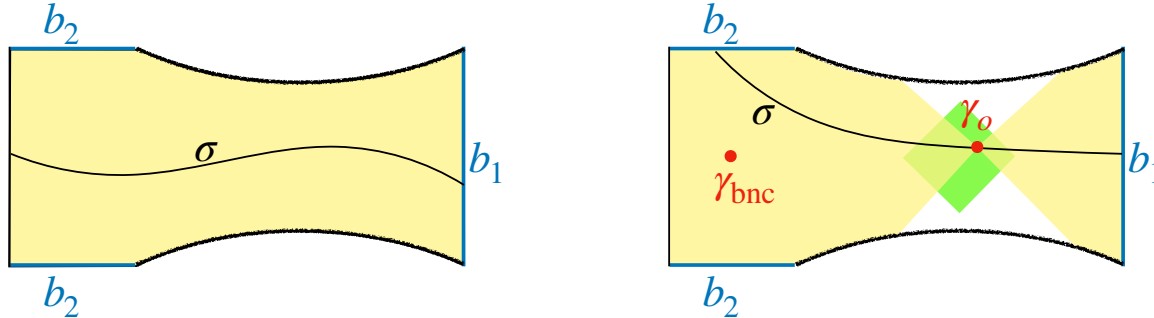

**Figure 23**. Penrose diagram for a wormhole connecting asymptotically AdS and dS regions. An observer in either region sees a black hole. Left: Orthodox application of HRT formula to AdS boundary $b_1$: $b_1$ is null-spacelike-homologous, since its intersection with any Cauchy slice $\sigma$ is null-homologous on $\sigma$. The entire spacetime is the entanglement wedge of $b_1$. Right: Heterodox application of HRT formula: instead of restricting to Cauchy slices, we allow achronal hypersurfaces $\sigma$ that end on the dS boundary $b_2$, and furthermore treat the intersection $\sigma \cap b_2$ as non-trivial in homology; we then find an extremal surface $\gamma_o$ lying in the causal shadow (green shading), which defines entanglement wedges (yellow). In time-symmetric spacetimes, there is an index-1 extremal surface $\gamma_{\text{bnc}}$ on the time-symmetric slice, lying in the dS region; typically, however, $\gamma_{\text{bnc}}$ is a bounce rather than a bulge, meaning that it has a positive temporal deformation mode.

## 5.1 Example: AdS-dS wormhole

Let's begin with a spacetime containing a wormhole connecting an asymptotically dS region to another asymptotic region which, for concreteness, we choose to be asymptotically AdS. We call the AdS region $a_1$ and the dS region $a_2$ (see Fig. 23). Because of the black hole in dS, there will necessarily be some matter on the other side of the dS region. (We could instead put another black hole there; we will discuss this below.) We call the AdS boundary $b_1$ and the union of the future and past dS boundaries $b_2$. An observer in either region sees a black hole. An explicit construction of such a spacetime can be seen in Fig. 40 (A1), involving part of a Schwarzschild-dS solution glued to part of a Schwarzschild-AdS solution along a domain wall.

An orthodox application of the HRT formula to the AdS boundary $b_1$ returns $S(b_1) = 0$: $b_1$ is spacelike-homologous to the empty set, which is therefore the HRT surface. (See left side of Fig. 23.) Furthermore, its entanglement wedge is the entire spacetime. The same result is obtained using the maximin formula, since on every Cauchy slice its intersection with $b_1$ is null-homologous. It is also obtained using the minimax formula, provided the spacetime homology constraint is enforced relative to $b_2$ as well as the singularities, so that $b_1$ is again null homologous. From this viewpoint, the

spacetime represents a microstate of the AdS black hole (or a set of $O(G_N^0)$ microstates).

Nonetheless, there exists a closed extremal surface $\gamma_o$ lodged in the throat of this wormhole. (See right side of Fig. 23.) For example, in the glued solution of Fig. 40 (A1), $\gamma_o$ is the bifurcation surface of the Schwarzschild-dS spacetime. In a general spacetime, the existence of a closed extremal surface can be proven using either maximin or minimax, but to do so we must change the rules of the game. In maximin, we should choose to include in the maximization not just Cauchy slices but also achronal hypersurfaces that end on $b_2$. If we consider the intersection of such a hypersurface $\sigma$ with $b_2$ to be non-trivial in homology (i.e. if we do *not* work relative to $b_2$), then its intersection with $b_1$ is also non-trivial. The induced metric on $\sigma$ is large near both the AdS and dS boundaries, so there exists a minimal surface somewhere in the middle. Now, if we push $\sigma$ forward or backward in time, close to either singularity, the minimal surface area will go to zero. Therefore, somewhere in the middle, there exists a maximin surface $\gamma_o$, which is extremal by the usual maximin arguments.[7] With minimax, we again strengthen the spacetime homology constraint so that $b_2$ is *not* considered trivial (i.e. we do not work relative to it). A time-sheet $\tau$ homologous to $b_1$ then ends on the past and future singularities, where the metric collapses, implying the existence of a maximal surface. If we push $\tau$ toward $b_1$ or toward the center of the dS region, the maximal area increases, so there is a minimax surface $\gamma_o$ somewhere in the middle, which again is extremal by the usual arguments.

The surface $\gamma_o$ is HRT-like in that it is a local minimum of the area against spatial deformations and a local maximum against temporal ones. By focusing, $\gamma_o$ lies in the causal shadow, i.e. it is out of causal contact with both $b_1$ and $b_2$. From the orthodox viewpoint espoused above, in which this spacetime represents a microstate of the AdS black hole, $\gamma_o$ would be the outermost extremal surface, whose area gives a coarse-grained entropy of the AdS black hole microstate [33].

We can try to go further, and appeal to the python's lunch conjecture (PLC) [40], according to which $\gamma_o$ would be interpreted as a constriction, and the dS region as a python. But there is a fly in the ointment: there may not exist a bulge surface, a necessary ingredient in the PLC. A bulge surface is an extremal surface in the given homology class with exactly one spatial negative mode and no temporal positive modes. In the time reflection-symmetric case, there does exist an extremal surface $\gamma_{\text{bnc}}$ with one spatial negative mode on the symmetric slice in the dS region. However, at least in the simplest solutions (such as the glued solution of Fig. 40 (A1), where $\gamma_{\text{bnc}}$ is the dS cosmological bifurcation surface), $\gamma_{\text{bnc}}$ also has a temporal positive mode; in

---

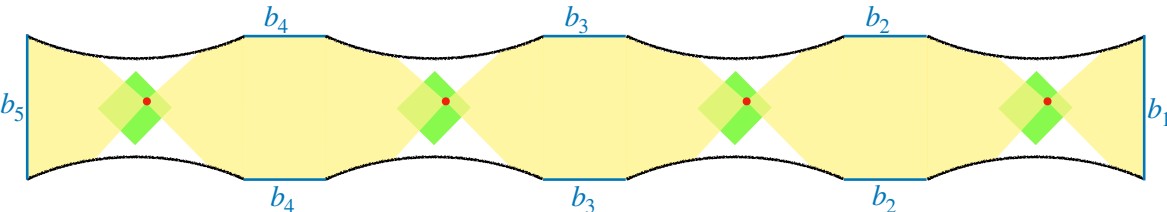

[7]As usual, there is a large freedom in choosing the maximin hypersurface $\sigma$, including where it ends on $b_2$. Therefore, $\sigma$ should not be viewed as dividing the dS boundary into subregions in any meaningful way.

**Figure 24**. Chain of dS regions connected by wormholes and capped at either end by an AdS region. The green regions are the the causal shadow and the red dots are HRT-like extremal surface. The yellow regions are entanglement wedges, according to the interpretation of the spacetime as an entangled state in the tensor product $\mathcal{H}^{b_5} \otimes \mathcal{H}^{b_4} \otimes \mathcal{H}^{b_3} \otimes \mathcal{H}^{b_2} \otimes \mathcal{H}^{b_1}$.

the classification of extremal surfaces of [76], this makes it a so-called *bounce* surface.[8] Thus, the status of the PLC in such spacetimes is unclear.

A different interpretation of this spacetime (which we could call "heterodox" to distinguish it from the above orthodox one) is that there is a Hilbert space $\mathcal{H}^{b_2}$ associated with the dS boundaries, presumably equal to the Hilbert space obtained by quantizing the quantum gravity theory with just dS boundary conditions, and the spacetime represents an entangled state in $\mathcal{H}^{b_1} \otimes \mathcal{H}^{b_2}$ with entanglement entropy equal to $|\gamma_o|/4G_N$. Since $\gamma_o$ lies in the causal shadow, one can define an entanglement wedge associated with each boundary. (See Fig. 23, right side.) Note that there is just one surface associated with both the future and past dS boundaries, so presumably one Hilbert space $\mathcal{H}^{b_2}$ for both of them.

Yet another interpretation of this spacetime, in the spirit of [69, 77], is that the black hole inside the dS region can be thought of as modeling an observer whose Hilbert space has dimension roughly $e^{|\gamma_o|/4G_N}$, with the AdS Hilbert space $\mathcal{H}^{b_1}$ being an auxiliary Hilbert space that purifies the observer.

We have presented three different plausible interpretations of this spacetime and of the surface $\gamma_o$. We emphasize that these interpretations are not necessarily mutually exclusive; they may simply be answering different questions or be valid in different approximations.

Although we have been considering a dS region that is asymptotically dS in both the past and future, the same considerations go through if the spacetime is asymptotically dS in just the past, with a singularity in the future, or vice versa.

## 5.2 De Sitter networks

We can also consider more complicated networks of universes, including dS regions, connected by wormholes. An example is shown in Fig. 24, involving a chain of dS regions capped at either end by an AdS region. Each wormhole has a causal shadow; by the same argument as in the previous subsection, each causal shadow hosts an HRT-like extremal surface. The orthodox interpretation of HRT would say that this spacetime represents an entangled state in the Hilbert spaces of the two AdS boundaries $\mathcal{H}^{b_5} \otimes \mathcal{H}^{b_1}$, with entanglement entropy given by the area of the smallest HRT-like surface. On the other hand, the heterodox interpretation would say that there is a Hilbert space $\mathcal{H}^{b_i}$ ($i = 2, 3, 4$) associated with each pair of dS asymptotic regions, and this spacetime represents an entangled state in the tensor product of all of the Hilbert spaces $\mathcal{H}^{b_5} \otimes \mathcal{H}^{b_4} \otimes \mathcal{H}^{b_3} \otimes \mathcal{H}^{b_2} \otimes \mathcal{H}^{b_1}$, with an entanglement wedge associated with each factor. As in the state represented by the Brill-Lindquist chain shown in Fig. 18, this state lacks long-range entanglement and is a Markov state.

Of course, much more complicated networks of universes can be imagined, involving arbitrary combinations of AdS, dS, and Minkowski asymptotic regions. For any such spacetime, there are two classes of HRT-like surfaces: those found by imposing the homology constraint relative to the dS boundaries (treating them the same as singularities and end-of-the-world branes), corresponding to the orthodox interpretation in which there are *not* Hilbert spaces associated with those boundaries; and those found by considering the dS boundaries to be non-trivial in homology (treating them the same as AdS or Minkowski boundaries), corresponding to the heterodox interpretation in which there is a Hilbert space associated with each dS boundary (or to each future/past pair of boundaries). The first class of surfaces obey all of the properties of HRT surfaces in asymptotically AdS and Minkowski spacetimes enumerated in section 2, as the proofs of those properties go through in the presence of dS boundaries. Whether or not this is also true for the second class of surfaces is not obvious (for example, the projection step in many of the proofs makes use of Cauchy slices, whereas the maximin algorithm that finds those surfaces does not involve Cauchy slices). This question constitutes a non-trivial test of the heterodox interpretation, which we leave to future work.

## 6 Discussion

In this final section, we consider a few last issues and loose ends, namely tensor networks representing asymptotically flat spacetimes, how to create entangled universes, the S-

---

[8]It is an interesting question whether there *always* exists a bounce surface in a spacetime with these asymptotics.

matrix interpretation, and the formation of wormholes by collapse. Finally, we close with a set of open questions and future directions.

## 6.1 Tensor networks

As discussed in subsection 1.3, a coarse TN description should exist for asymptotically Minkowski spacetimes. Since there is no CFT dual, this does not necessarily reproduce UV physics on the boundary, but following [44] there should exist a decomposition of the Cauchy slice $\Sigma$ along RT surfaces $\gamma_i$ of the (entire) boundary regions $A_i$.

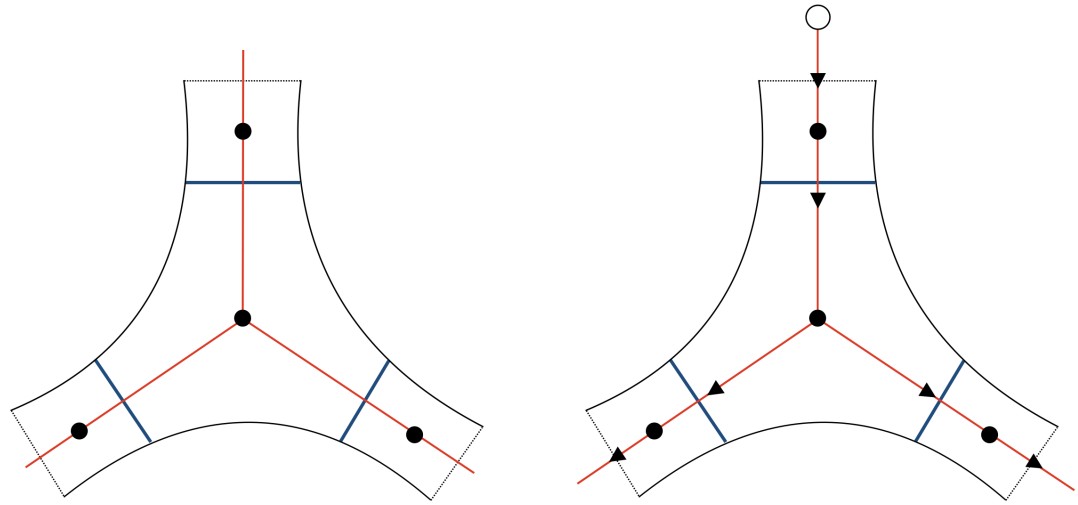

**Figure 25**. (Left) Target discretization of the $n = 3$ Brill-Lindquist wormhole, given a root-leaf orientation (right) by picking a root node (denoted by white circle)

Following the procedure outlined in [44], we can construct a tree tensor network for our "target" geometrization of the $n = 3$ BL metric by RT (minimal) surfaces. Here we consider a parameter regime where each of $\gamma_1, \gamma_2, \gamma_3$ are present and minimal, homologous to the entire boundaries $A_1, A_2, A_3$. As shown in Fig. 25, we then have a target discretization of the wormhole along the (non-intersecting) RT surfaces, which we must provide with a root-leaf orientation to ensure isometry properties of the tensor network. One may consider this "state" to correspond to a dual CFT living on the boundary by gluing AdS patches sufficiently far away from the minimal surfaces; however, this is not necessary for the discussion. Regardless of the interpretation, we consider the tensor network to represent a state $\psi^{A_1 A_2 A_3}$ in the tensor product Hilbert space $\mathcal{H} = \mathcal{H}_{A_1} \otimes \mathcal{H}_{A_2} \otimes \mathcal{H}_{A_3}$. The state is constructed by contracting bulk tensors over the bond Hilbert spaces $\mathcal{H}_{\gamma_i}$, where the bond dimensions are $|\gamma_i|/4G_N$. On the tensor

network diagrams, upper (lower) indices are indicated by outward (inward) pointing arrows.

Then we follow the inductive procedure outlined in [44]: add RT surfaces from the leaves up to the root, adding all "child" surfaces before the "parent" surface. We also invoke the entanglement distillation procedure, wherein each bond can be distilled into EPR pairs $\phi_i$ that are then contracted over (instead of contracting tensors with each other) to form a PEP (projection of entangled pairs) network. The distillation results in the state being decomposed into $|\gamma_i|/4G_N$ Bell pairs and a leftover state $\sigma_i$, which exists in a Hilbert space of subleading dimension.

As outlined in Fig. 26, we add the RT surfaces inductively, forming a bipartite network at each stage and modifying the tensors to respect the existing structure. For example, when adding $\gamma_2$ after adding $\gamma_1$, to preserve the structure of the states $\phi_1, \sigma_1$, we must replace our bulk tensor $W_2$ by

$$W'' = \sigma_1^{-1}\phi_1^{-1}W_2 \tag{6.1}$$

Similarly, when adding the final surface $\gamma_3$, we replace the bulk tensor $W'$ by

$$W = \left(\sigma_1^{-1}\phi_1^{-1}\right)\left(\sigma_2^{-1}\phi_2^{-1}\right)W' \tag{6.2}$$

A similar procedure can be followed for $n = 4$ BL wormholes when there exist no intersecting minimal surfaces.

An interesting consideration here is what happens when we glue BL wormholes together. As outlined in subsection 4.4, BL geometries can be glued together along minimal surfaces to form valid initial data. A natural question then is how are tensor networks preserved (or not preserved) under such gluings? More precisely, can the TN of the glued geometry be formed by some kind of contraction of the 2 component TNs? One could conceive of such a procedure for the $n = 3$ BL geometry. Consider our TN $\psi^{A_1 A_2 A_3}$ and a copy TN $\tilde{\psi}^{\tilde{A}_1 \tilde{A}_2 \tilde{A}_3}$. Suppose we now consider identifying $A_1$ with $\tilde{A}_1$ and tracing over that index. It is unclear if the resulting state indeed has a well-defined geometric dual; one natural possibility would be the spacetime obtained by gluing together the above spacetimes along $\gamma_1$.

## 6.2 Creation of entangled universes

A natural question in this context is whether entangled universes could form dynamically from a single parent universe. It is interesting to note that this can happen via the Coleman-De Luccia decay of an inflating false vacuum [78]. The first possibility is that inflation eventually terminates for the parent universe by some other mechanism. If the true vacuum also inflates, one can end up with large numbers of "primordial ER

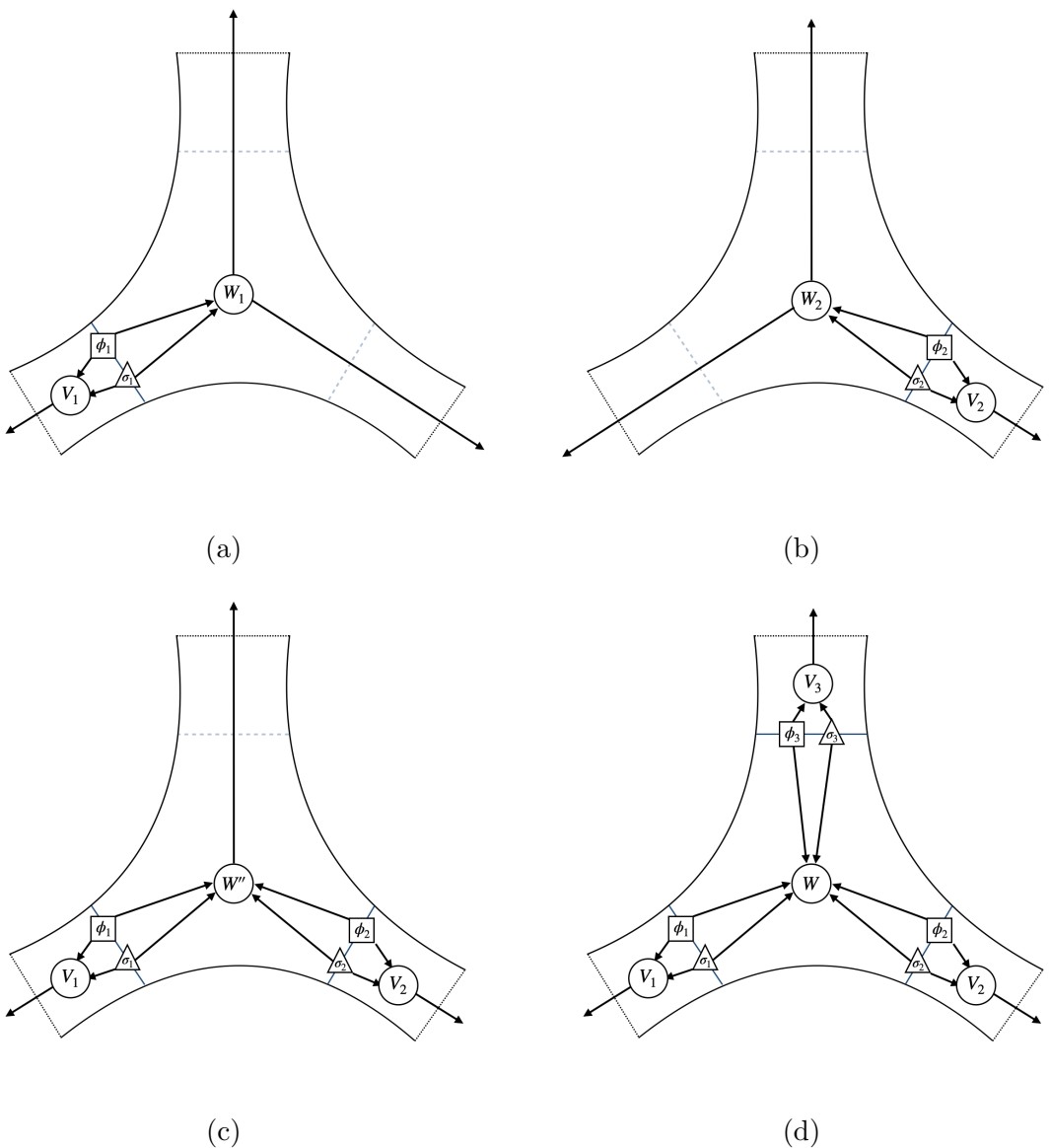

**Figure 26**. (a) We start by adding one of the 'child' surfaces, $\gamma_1$, which separates the bulk into the boundary tensors $V_1$ and $W_1$ contracted along $\phi_1, \sigma_1$. (b) The same procedure can be done for the second 'child' surface, $\gamma_2$, adding the tensors $V_1$, $W_2$. (c) To preserve the structure of the states $\phi_1, \sigma_1$ on the already constructed surface $\gamma_1$, we can replace $W_2$ by $W''$ given by (6.1) to accomodate both $\gamma_1, \gamma_2$ in the TN. (d) A similar procedure is done to add $\gamma_3$, where we first have our added boundary tensor $V_3$ contracted to some bulk tensor $W'$, which is then replaced by $W$ given by (6.2) to form the full network

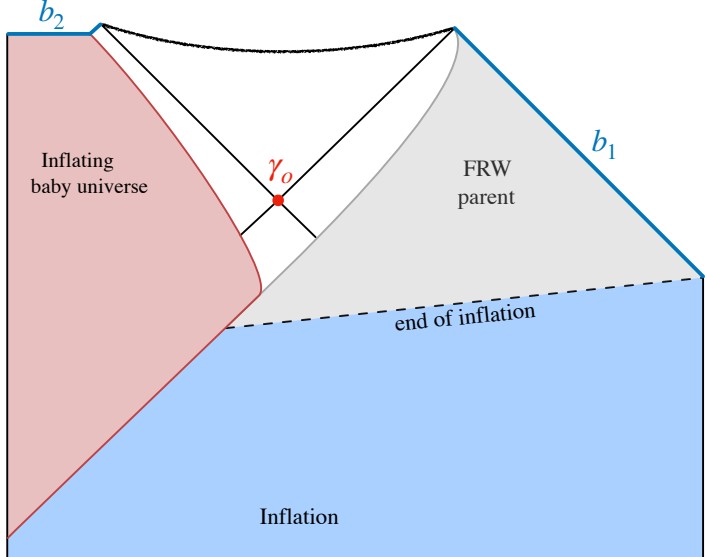

**Figure 27**. Primordial ER bridge from a false vacuum decay during inflation. The dashed line is the slice on which inflation terminates for the parent universe. If the bubble is large enough, it will inflate behind the horizon of a black hole as viewed from the parent universe. The boundary $b_1$ is future null infinity of the FRW universe with flat spatial slices, while $b_2$ includes both the future conformal boundary of the baby universe, which is a piece of de Sitter space, and the future null infinity of part of an asymptotically flat Schwarzschild spacetime (white region).

bridges" from a (e.g. asymptotically flat) parent FRW universe, each connected to an inflating baby universe. An example of a single nucleation event is shown in Fig. 27. For the details of this process, including the expected distribution of such primordial ER bridges, we refer the reader to [79]. In this case, the orthodox application of the HRT formula would give us zero entropy, in accord with the interpretation of a microstate in the hologram of the parent universe; however, the heterodox application of the HRT formula would give us the entropy of each baby universe.

Another possibility is to consider an eternally inflating meta-stable false vacuum. In this case, bubbles also nucleate, and asymptotically flat or AdS spatial boundaries form where the wall of the inflating bubble hits the future de Sitter boundary. This way, multiple universes can form. However, the nucleation of ER bridges connecting them is suppressed with respect to the nucleation of disconnected universes [80]. But even in the latter case, the state of the quantum fields can still get considerable entanglement between the different universes. An indication of this is that entanglement islands can form in related situations [81].

## 6.3 S-matrix and collapse

For asymptotically flat universes, the ER bridges that our conjectures apply to are unstable but long-lived configurations. The black holes will eventually evaporate, and the state of the Hawking radiation in each universe will remain entangled. These spacetimes can also be interpreted as S-matrix resonances of entangled in/out states of radiation between the universes. A basic observation is that the S-matrix factorizes between the different universes; and thus these spacetimes can be alternatively described as linear superpositions of products of single-universe S-matrix elements. This is in accord with the standard observation that spatial connectivity is a non-linear property in quantum gravity. In this case, this shows that the microscopic description of ER bridges cannot be intrinsic to the S-matrix formulation.

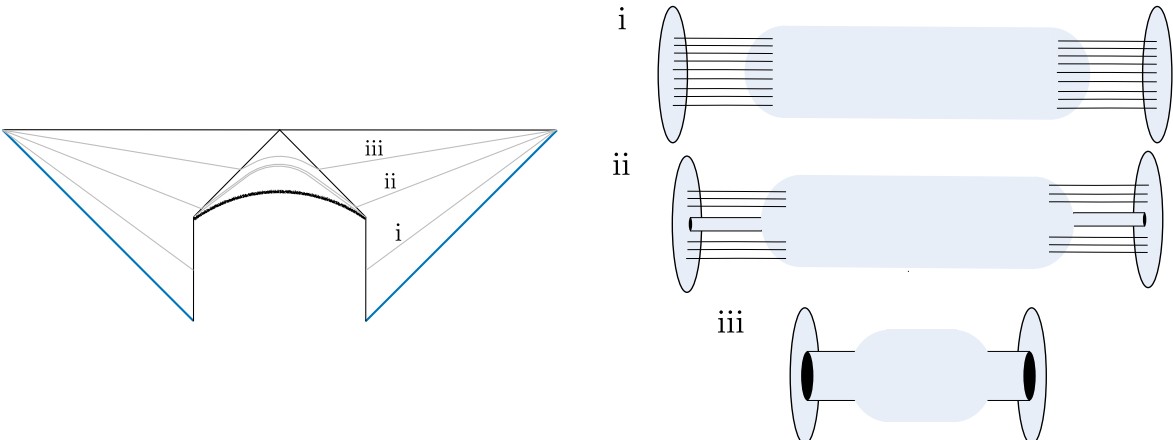

**Figure 28**. On the left, half of the Penrose diagram of the time-reversal of the Hawking evaporation of a two-sided Schwarzschild black hole. On the right, cartoon of the geometry and entanglement of the slices. This process can be interpreted as the semiclassical unitary needed to decode the baby universe from the radiation, as the latter becomes a causally connected white hole.

This also leads to a foundational question about ER=EPR: if we are given some entangled in-state of radiation in a collection of disconnected universes, and collapse this radiation to form black holes, do we create ER bridges? From the time-reversal of the Hawking evaporation in a two-sided Schwarzschild black hole, it seems that this is possible, as illustrated in Fig. 28. However, this corresponds to an extremely fine-tuned initial state, and the process violates the generalized second law of thermodynamics and the null energy condition. One interesting fact about this process, though, is that the black hole interior seems to already reside in the in-state of the radiation as

a closed universe (i).[9] Collapsing the very late Hawking quanta into a microscopic (perhaps Planck-scale) volume creates a geometric connection to this wormhole (ii). Furthermore, feeding this wormhole in a way which is fine-tuned with the state of the wormhole shortens the wormhole (iii). This process "decodes" the black hole interior, granting causal access to it as the white hole region. Failure to implement it correctly generates a singular stress-energy at the horizon.

What about more generic situations? A strategy to study this could be to first classify generic entangled states of multiple black holes which, in the vein of [83], are expected to contain inhomogeneous matter distributions supporting very long and fully connected wormholes. The unclear question is whether collapsing a generic, i.e., maximally complex, state of the radiation lands close to a generic entangled state of the black holes, given the irreversibility of the gravitational collapse.

## 6.4   Future directions

The generalizations of the HRT formula conjectured in this paper lead to many interesting questions, which we leave to future work.

A technical question raised in subsection 5.2 concerns the "heterodox" HRT-like surfaces that appear in throats of asymptotically dS spacetimes, namely: Do they obey all of the properties of "orthodox" HRT surfaces, discussed in section 2? This is an interesting general relativity question in itself, and is important for understanding whether the areas of these surfaces can sensibly be interpreted as entropies.

One can also ask about spacetimes with asymptotics other than AdS, dS, and Minkowski, such as pp-waves. One set of questions is whether we would still expect HRT-like surfaces in such spacetimes, and whether they would obey the properties that HRT surfaces do. Another puzzling class of spacetimes are closed universes that bang and crunch, and therefore have no asymptotic regions at all, either in time or space. A wormhole can certainly connect two such universes, and we would expect it to host an HRT-like surface. Does the area of that surface represent an entanglement entropy, and if so, of what state?

An obvious question is whether conjecture 3 can be supported by a path integral calculation of Rényi entropies, as was done for the RT formula by Lewkowycz-Maldacena [84]. Here there are a couple of issues to note. First, even in the time-reflection symmetric case, where the spacetime has a real Euclidean continuation, it is not clear that asymptotically flat wormholes are ever dominant saddles. Therefore, a more complicated path integral (perhaps along the lines of the fixed-area states [85] or the approach

---

[9]Simpler situations where baby universes appear in the state of the radiation can be constructed in conventional AdS/CFT [82].

of [86]) would presumably be required. Second, it's not obvious how to set up boundary conditions to calculate Rényi entropies for asymptotically Minkowski boundaries.

Finally, it would be interesting to test the conjectures in microscopic models of holography in flat space. Beyond the general arguments presented in section 3, there is direct evidence in this regard coming from the validity of the RT formula in AdS/CFT, given that the large-$N$ four-dimensional $\mathcal{N} = 4$ SYM theory describes a gravitational sector of ten-dimensional flat space in string theory. This sector includes small ten-dimensional Schwarzschild black holes, which are microcanonically stable states of the CFT at large energies. However, ideally, we would like to consider full-fledged flat space holograms, a candidate example being the D0-brane quantum mechanics conjectured to describe M-theory [87].

## Acknowledgments

We would like to thank Ning Bao, Sumit Das, Roberto Emparan, Netta Engelhardt, Guglielmo Grimaldi, Robie Hennigar, Gary Horowitz, Veronika Hubeny, Albion Lawrence, Hong Liu, Pratik Rath, and Brian Swingle for useful discussions. This work was performed in part at the Aspen Center for Physics, which is supported by the National Science Foundation grant PHY-2210452. M.H. and M.S. acknowledge support from the U.S. Department of Energy through DE-SC0009986 and QuantISED DE-SC0020360.

## A    Junction conditions for patching spacetimes

Here we provide a discussion of the procedure for patching the spacetimes together and how this is used to obtain the constraints on the geometries described in section 3. The procedure follows [45].

We make the simplifying assumption that the metrics we are patching are spherically symmetric. We also consider each junction one at a time; performing multiple gluings simply repeats the process at each junction.

It's also helpful to establish a system of Gaussian normal coordinates near the junction. The direction of the normal vector in these coordinates is then taken to be the 'outward' normal direction, thus providing a notion of "inner" and "outer" metrics.

$$ds_i^2 = -f_i(r)dt_i^2 + \frac{dr^2}{f_i(r)} + r^2 d\Omega^2 \tag{A.1}$$

$$ds_o^2 = -f_o(r)dt_o^2 + \frac{dr^2}{f_o(r)} + r^2 d\Omega^2 \tag{A.2}$$

The subscript $i$ refers to the inner metric while $o$ refers to the outer metric. Note that the time is different across the metrics ($t_i \neq t_o$ in general) but $r$ is a physically meaningful coordinate, and thus must be the same for both metrics. Our convention fixes the right to be the outward normal direction.

The induced 3-metric on the boundary,

$$ds^2_{\text{brane}} = -d\tau^2 + R^2(\tau)d\Omega^2 \tag{A.3}$$

represents the metric of the matter brane connecting the spacetimes, $\tau$ being the proper time. The brane is a matter sheet of positive tension, with its stress-energy tensor given by a delta function, and position within the spacetime given by $r = R(\tau)$. The tension $\sigma$ is related to the parameter $\kappa = 4\pi G_N \sigma$. We now apply the Israel junction conditions (see [88] for details) to relate the matter discontinuity (the brane) to the difference in extrinsic curvature across the metrics to obtain the equation of motion for the brane,

$$\dot{R}^2 + V_{\text{eff}}(R) = 0 \tag{A.4}$$

which is the equation of motion for a particle with 0 energy, moving in a potential $V_{\text{eff}}$. The potential is determined by the factors $f_i(r)$ and $f_o(r)$. For spherically symmetric configurations, the general form for the factors is $f(r) = 1 - \lambda r^2 - \frac{\mu}{r}$, where $\lambda$ is the cosmological constant and $\mu$ is the mass of a black hole. In terms of these parameters, [45] obtain an expression for $V_{\text{eff}}$:

$$V_{\text{eff}}(r) =$$
$$1 - \left(\lambda_o + \frac{(\lambda_o - \lambda_i - \kappa^2)^2}{4\kappa^2}\right)r^2 - \left(\mu_o + \frac{(\lambda_o - \lambda_i - \kappa^2)(\mu_o - \mu_i)}{2\kappa^2}\right)\frac{1}{r} - \frac{(\mu_o - \mu_i)^2}{4\kappa^2}\frac{1}{r^4} \tag{A.5}$$

This function has certain important properties; as $r \to 0$, $V_{\text{eff}} \to -\infty$. Furthermore, to have a configuration where the brane reaches the boundary we must have $V_{\text{eff}} \to -\infty$ for $r \to \infty$. There then exists a maximum $V_{\text{max}}$, and its sign determines the brane's trajectory.

The time symmetric trajectories from Fig. 29 are A, B and D, but in D the brane is stationary and so we only consider trajectories of type A and B, for which we require $V_{\text{max}} > 0$.

This is not sufficient to perform the gluing however, as in obtaining (A.5) we had to square the constraint on the extrinsic curvature and so we lost the sign information of the extrinsic curvature. Thus, we must also consider our expressions for the extrinsic

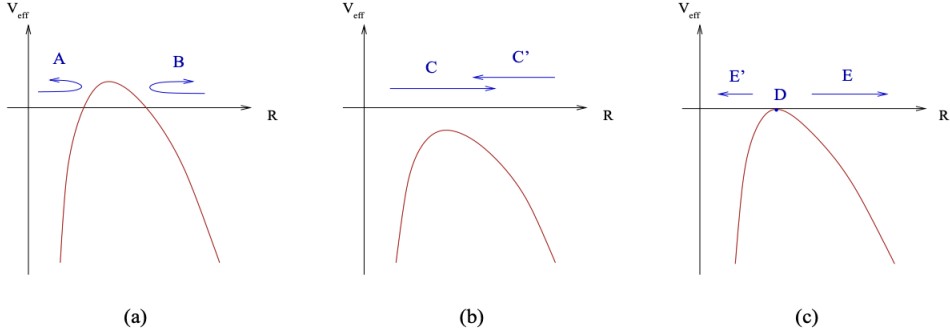

**Figure 29**. Shell trajectories for $V_{\text{eff}} \to -\infty$ as $r \to \infty$. **(a):** $V_{\max} > 0$, shell can expand from 0 and contract back (A) or contract from $\infty$ and expand back (B), **(b):** $V_{\max} < 0$, shell can expand/contract (C/C') without bound, **(c):** $V_{\max} = 0$, the shell can remain stationary (D) or contract/expand (E'/E) from finite size. *Reproduced from [45].*

curvature $\beta$, given for our ansatz by:

$$\beta_i(r) = \left(\frac{\lambda_o - \lambda_i + \kappa^2}{2\kappa}\right) r + \left(\frac{\mu_o - \mu_i}{2\kappa}\right) r^{-2} \tag{A.6}$$

$$\beta_o(r) = \left(\frac{\lambda_o - \lambda_i - \kappa^2}{2\kappa}\right) r + \left(\frac{\mu_o - \mu_i}{2\kappa}\right) r^{-2} \tag{A.7}$$

This is used to constrain the allowed worldlines for the brane. For branes attached at $r = 0$, we have to check that the corresponding $\beta \to \infty$ as $r \to 0$ if the outward normal near the attachment points in the direction of increasing $r$ (and vice versa, i.e. $\beta \to -\infty$ if the outward normal points to decreasing $r$). Similar conditions must be checked in the limit $r \to \infty$ for branes attached at $r = \infty$.

Now suppose we have the configuration above where we take SAdS to be the 'outer' metric. As shown above, both worldlines for the brane are indistinguishable by the previous considerations on the asymptotic behavior of the extrinsic curvature; however, one worldline passes through exterior region I and the other through region III. To determine which worldline is valid, we must look at the sign of $\beta_o$ at the point the brane is reflected back, i.e. along the Cauchy slice $\sigma$.

As we can see from Fig. 31, for case 1 $\beta_o$ is positive at the turning point and thus consistent with the brane through region I (where the outward normal points in the direction of increasing $r$), while in case 2, $\beta_o$ is negative at the turning point and is thus consistent with the brane through region III.

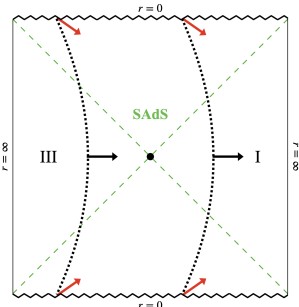

**Figure 30**. Two possible worldlines through SAdS for a brane attached at $r = 0$. Red arrows indicate direction of outward normal at point of attatchment; both point to increasing $r$ and are consistent with $\beta \to \infty$ at $r \to 0$. Black arrows indicate direction of outward normal on the Cauchy slice (turning point for the brane)

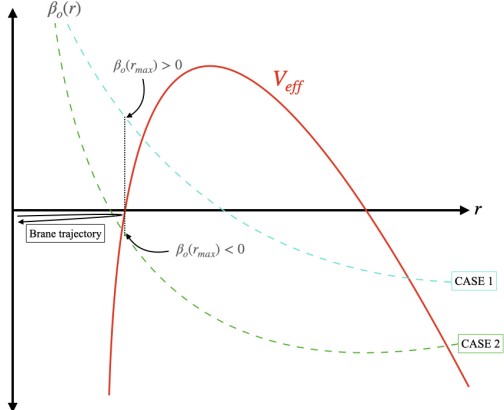

**Figure 31**. Two possibilities for extrinsic curvature with different signs at $r_{\max}$ ( turning point for the brane)

## A.1 Schwarzschild metric gluings

We have 2 gluings to perform, on a left and right junction, which are both at $r = 0$ when the spacetime in between is the flat space Schwarzschild metric. At the left junction, the left SAdS metric is "inner" while the Schwarzschild metric is "outer", and vice versa at the right junction. Assume the left/right SAdS metrics and branes are identical. Let the SAdS black hole mass be $\mu_A$ and the Schwarzschild black hole mass be $\mu_S$. $\lambda_o = \lambda_S = 0$ and set $\lambda_i = \lambda_{SAdS} = -\lambda < 0$. Now we obtain $V_{\text{eff}}$ and the

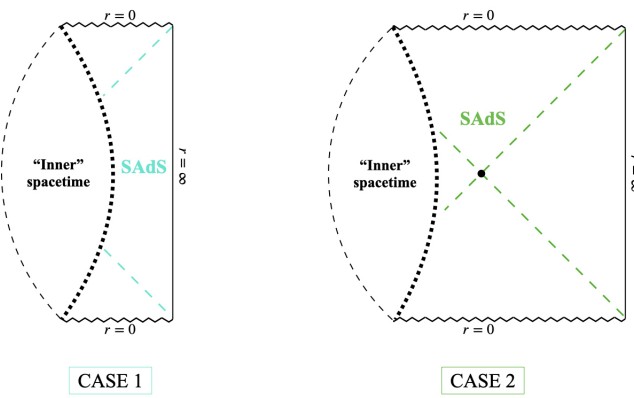

**Figure 32**. Spacetimes corresponding to extrinsic curvature values in Fig. 31

extrinsic curvatures:

$$V_{\text{eff}}(r) = 1 - \frac{(\lambda - \kappa^2)^2}{4\kappa^2}r^2 - \left(\mu_S + \frac{(\lambda - \kappa^2)(\mu_S - \mu_A)}{2\kappa^2}\right)\frac{1}{r} - \frac{(\mu_S - \mu_A)^2}{4\kappa^2}\frac{1}{r^4} \qquad (A.8)$$

$$\beta_i(r) = \left(\frac{\lambda + \kappa^2}{2\kappa}\right)r + \left(\frac{\mu_S - \mu_A}{2\kappa}\right)\frac{1}{r^2} \qquad (A.9)$$

$$\beta_o(r) = \left(\frac{\lambda - \kappa^2}{2\kappa}\right)r + \left(\frac{\mu_S - \mu_A}{2\kappa}\right)\frac{1}{r^2} \qquad (A.10)$$

Note that as $r \to \pm\infty$, $V_{\text{eff}} \to -\infty$ if $\kappa \neq 1$. Furthermore, $\beta_i > \beta_o \forall r \in [0, \infty)$.

Now consider branes at $r = 0$:

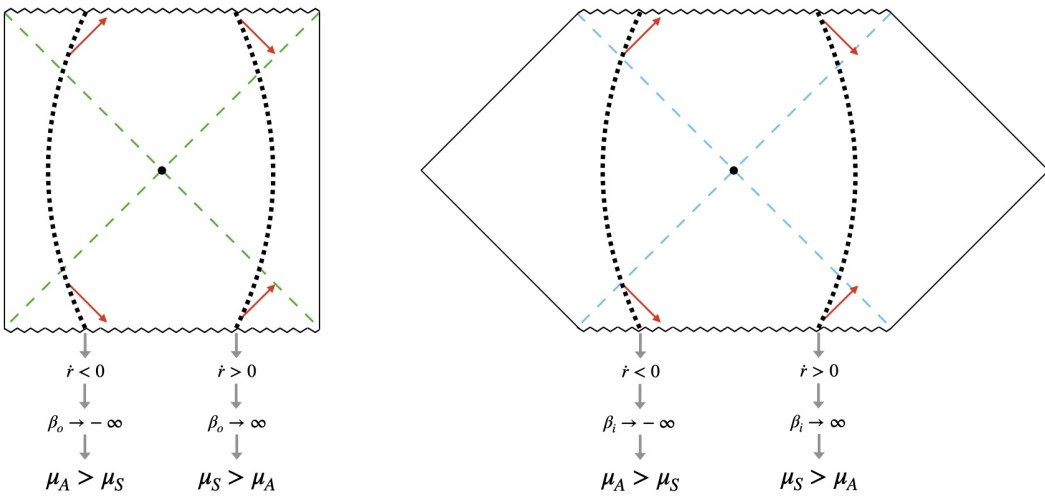

**Figure 33**. Conditions on branes attached at $r = 0$ with SAdS (green) "inside" and Schwarzschild (blue) "outside"

Here we consider 2 possible time symmetric worldlines for the branes attached at $r = 0$ in both metrics. The red arrows show the direction of the outward normal near the brane boundaries ($r = 0$). These indicate the behavior of $r$ near the boundary, i.e. whether it is increasing or decreasing, which determines the sign $\beta_{i/o} \to \pm\infty$ as $r \to 0$. The behavior of both $\beta_i$ and $\beta_o$ in this limit is governed by the 2nd term in (A.9)-(A.10) i.e. $\beta_{i/o} \to \infty$ if $\mu_S > \mu_A$ and vice versa. Note that throughout our analyses we take $\mu_A \neq \mu_S$.

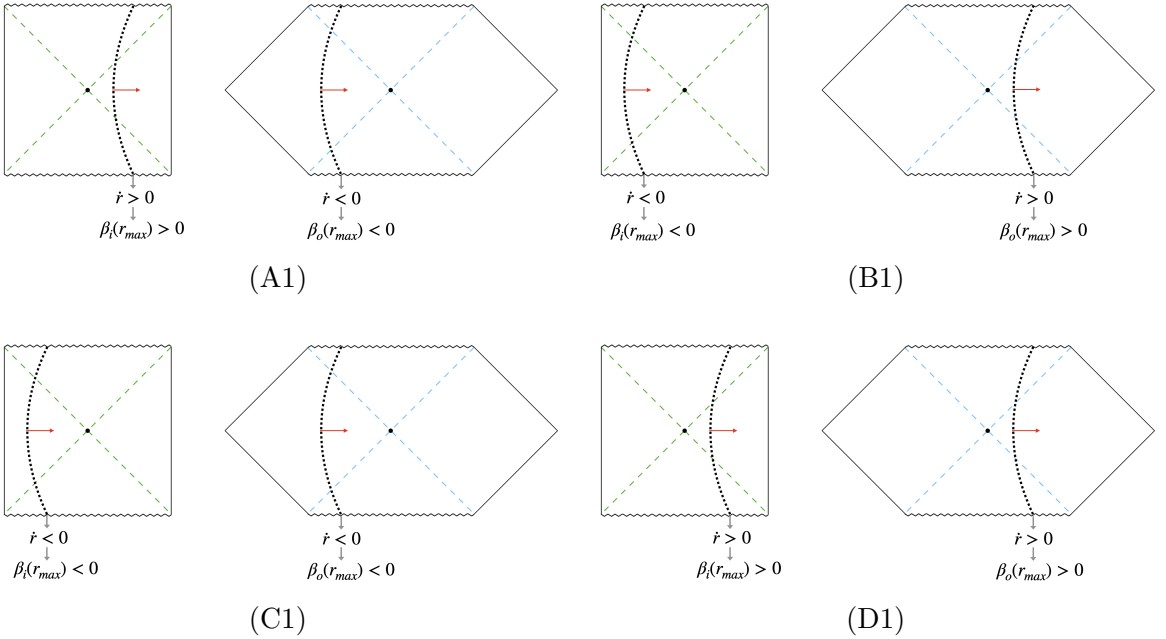

**Figure 34**. Configurations for $\mu_A > \mu_S$

Now we consider what exterior regions the worldlines can pass through for $\mu_A > \mu_S$. Fig. 34 shows the outward normal at the moment of time reflection symmetry. The direction of this normal must match the sign of the extrinsic curvature at this turning point; i.e. if the normal points to increasing (decreasing) $r$ in the relevant metric, then $\beta_{i/o}(r_{\max}) > 0$ ($\beta_{i/o}(r_{\max}) < 0$), where $r_{\max}$ is the radial coordinate at the moment of time reflection symmetry. Since $\beta_i > \beta_r \ \forall r$; (B1) is disallowed. From (A.9) we see the sign of $\beta_i$ has no determinate behavior for $\mu_A > \mu_S$; however $\beta_o < 0 \ \forall r$ if $\kappa^2 > \lambda$; thus (D1) is only possible with $\kappa^2 < \lambda$.

Similarly consider the configurations for $\mu_S > \mu_A$ shown in Fig. 35. By the same considerations as before, (B2) is disallowed. However, now there is a sign constraint on $\beta_i$; for $\mu_S > \mu_A$, $\beta_i > 0 \ \forall r$. Thus (C2) is also disallowed. Furthermore, $\beta_o > 0$ for all $r$ if $\kappa^2 < \lambda$; therefore (D2) is constrained to $\kappa^2 > \lambda$.

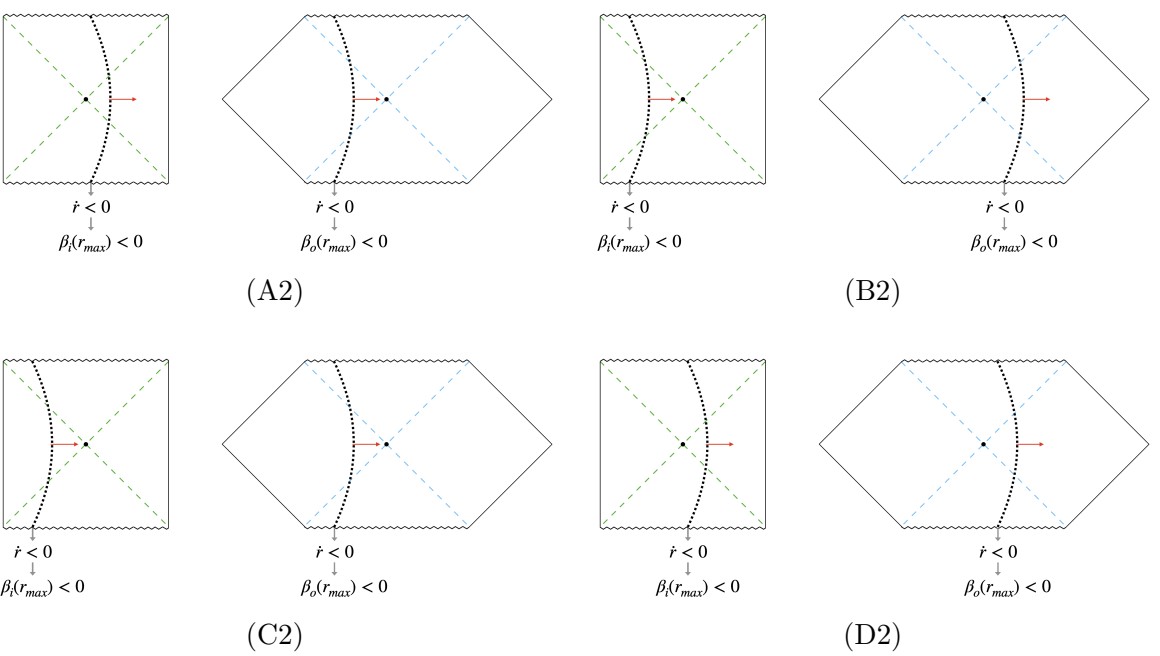

**Figure 35**. Configurations for $\mu_S > \mu_A$

All these configurations are summarised in Fig. 36.

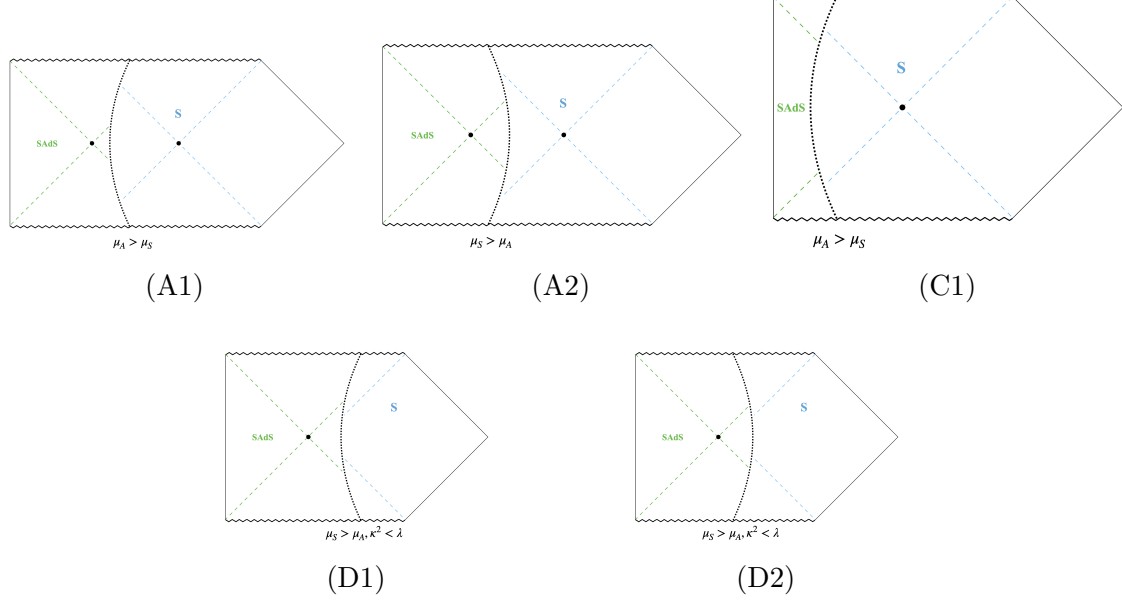

**Figure 36**. Configurations for branes at $r = 0$

The process is analogous for the right junction, where SAdS is now the outer metric and Schwarzschild is the inner metric. However we can skip the analysis by noting:

$$V_{\text{eff}}^{(L)}(r) = V_{\text{eff}}^{(R)}(r) \tag{A.11}$$

$$\beta_i^{(L)}(r) = -\beta_o^{(R)}(r) \tag{A.12}$$

$$\beta_o^{(L)}(r) = -\beta_i^{(R)}(r) \tag{A.13}$$

where we denote the functions corresponding to the left and right junctions by (L) and (R). Note this is only true for symmetric gluings, i.e. when the inner (outer) metric and brane parameters on the left are the same as the outer (inner) metric and brane parameters on the right. A consequence of this is that only configurations that are symmetric about the central bifurcation surface (which is the Schwarzschild horizon) are allowed. Thus the Schwarzschild bifurcation surface cannot be 'cut out', and the only possibilities for the gluings are the symmetric gluings (A1), (A1) and (C1), which correspond to case 1, 2, and 3 respectively from section 3.

Our goal was to consider fixed $\mu_S, \lambda, \kappa$ and show a gluing of case 1 or 3 can be performed for some tunable $\mu_A$. We also want the gluings to be arbitrarily far (i.e. the brane turning points $r_{\text{turn}} \to \infty$), as in section 4 we argue that the arguments from section 3 can be used to glue SAdS patches to BL wormholes arbitrarily far away. However, the constraint we have imposed on cases 1 and 3 (i.e. $\mu_A > \mu_S$) is only a necessary condition for the gluings to be performed. To find both necessary and sufficient conditions for the gluings is very difficult, as even solving for the condition that $V_{\text{eff}}(r_0) > 0$ for some $r_0$ involves solving a quintic.

Instead, we consider the small tension regime $\lambda > \kappa^2$. We also take $\mu_A > \mu_S$, as this is a necessary (but not sufficient) condition for the Schwarzschild bifurcation surface to be smaller than the SAdS bifurcation surface and thus to be the RT (minimal) surface. With these conditions, one finds $r_0 > 0$ such that

$$V_{\text{eff}}(r_0) = f_o(r_0) = 1 - \frac{\mu_S}{r_0} \tag{A.14}$$

where we are considering the left junction (the outer metric is Schwarzschild) and $r_0 = \left(\frac{\mu_A - \mu_S}{\lambda - \kappa^2}\right)^{1/3}$. We see that taking $\mu_A \to \infty$ sends $r_0 \to \infty$, which implies $V_{\text{eff}}(r_0) \to 1 > 0$, i.e. the potential has a positive maximum as required. Note that $r_{\text{turn}} < r_0$, but the effect of $\mu_A \to \infty$ is to shift $V_{\text{eff}}$ to the right, also implying $r_{\text{turn}} \to \infty$. Our gluings are thus arbitrarily far as required. Furthermore, $r_A > r_S$ in the limit $\mu_A \to \infty$, where $r_A$ and $r_S$ are the SAdS and Schwarzschild black hole radius, respectively. Thus we also

ensure the Schwarzschild bifurcation surface is smaller than the SAdS surfaces, and is thus the RT surface. The condition $\mu_A > \mu_S$ ensures $\beta_{i/o}$ have the correct asymptotic behavior for $r \to 0$; what we also require is that $\beta_o(r_{\text{turn}}) < 0$, as we must have the branes pass through the exterior regions of the Schwarzschild metric. For large $r$, $\beta_o$ is an increasing function and so $\beta_o(r_{\text{turn}}) < \beta_o(r_0) = 0$ as needed.

Thus, in the small tension limit ($\lambda > \kappa^2$) and for $\mu_A > \mu_S$, taking $\mu_A \to \infty$ guarantees an arbitrarily far gluing of case 1 or 3. Determining which of case 1 or 3 we have comes down to the sign of $\beta_i(r_{\text{turn}})$, which we cannot determine as $\forall r, \beta_i > \beta_o$. Numerical manipulation of the parameters suggests that in the limit $\mu_A \to \infty$ we obtain a case 1 gluing (including all 3 bifurcation surfaces, with Schwarzschild being minimal).

## A.2 Schwarzschild-de Sitter metric gluings

For SdS (Schwarzschild-de Sitter), we use the same method as for the Schwarzschild metric, and start with the $r = 0$ gluings. At the left junction, the left SAdS metric is "inner" while the SdS metric is "outer", and vice versa at the right junction. Assume the left/right SAdS metrics and branes are identical. Let the SAdS black hole mass be $\mu_A$, the SdS black hole mass be $\mu_S$, $\lambda_o = \lambda_{dS} > 0$ and $\lambda_i = \lambda_{SAdS} = -\lambda > 0$. Now we obtain $V_{\text{eff}}$ and the extrinsic curvatures:

$$V_{\text{eff}}(r) = 1 - \left( \lambda_{dS} + \frac{(\lambda_{dS} + \lambda - \kappa^2)^2}{4\kappa^2} \right) r^2 - \left( \mu_S + \frac{(\lambda_{dS} + \lambda - \kappa^2)(\mu_S - \mu_A)}{2\kappa^2} \right) \frac{1}{r} - \frac{(\mu_S - \mu_A)^2}{4\kappa^2} \frac{1}{r^4}$$
(A.15)

$$\beta_i(r) = \left( \frac{\lambda_{dS} + \lambda + \kappa^2}{2\kappa} \right) r + \left( \frac{\mu_S - \mu_A}{2\kappa} \right) \frac{1}{r^2}$$
(A.16)

$$\beta_o(r) = \left( \frac{\lambda_{dS} + \lambda - \kappa^2}{2\kappa} \right) r + \left( \frac{\mu_S - \mu_A}{2\kappa} \right) \frac{1}{r^2}$$
(A.17)

Note that as $r \to \pm\infty$, $V_{\text{eff}} \to -\infty$. Furthermore, $\beta_i > \beta_o \forall r \in [0, \infty)$.

Now consider the worldlines for branes at $r = 0$ as shown in Fig. 37.

Once again, the red arrows show the direction of the outward normal near the brane boundaries ($r = 0$). The behavior of both $\beta_i$ and $\beta_o$ in the limit $r \to 0$ is governed by the 2nd term in (A.16)-(A.17) i.e. $\beta_{i/o} \to \infty$ if $\mu_S > \mu_A$ and vice versa.

Now consider the possible worldlines for $\mu_A > \mu_S$ as shown in Fig. 38. By the relationships of $V_{\text{eff}}, \beta_i, \beta_o$ between the left and right junctions, we know that only a symmetric configuration may be allowed, and so the central SdS bifurcation surface may not be cut out.

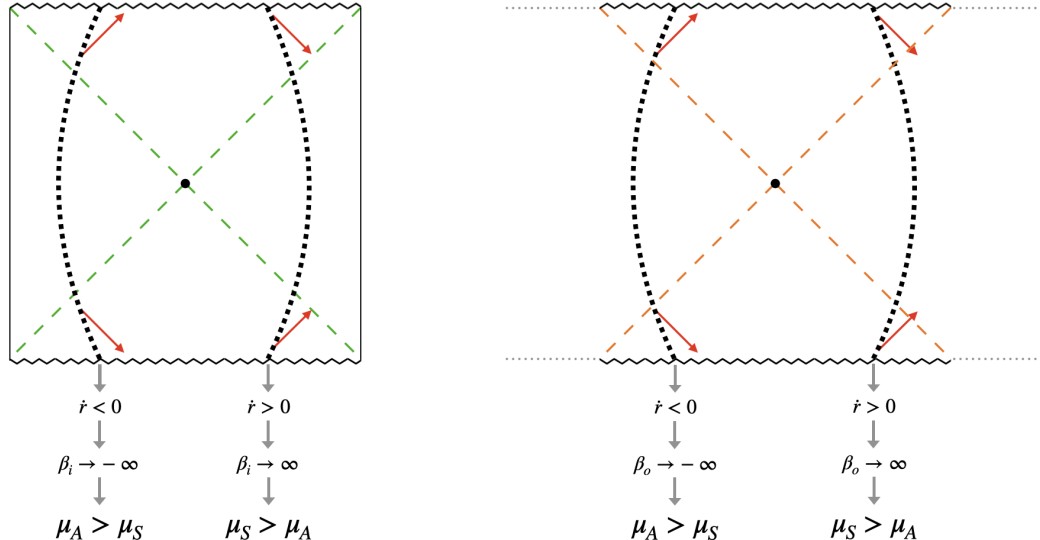

**Figure 37**. Conditions on branes attached at $r = 0$ with SAdS (green) "inside" and SdS (orange) "outside"

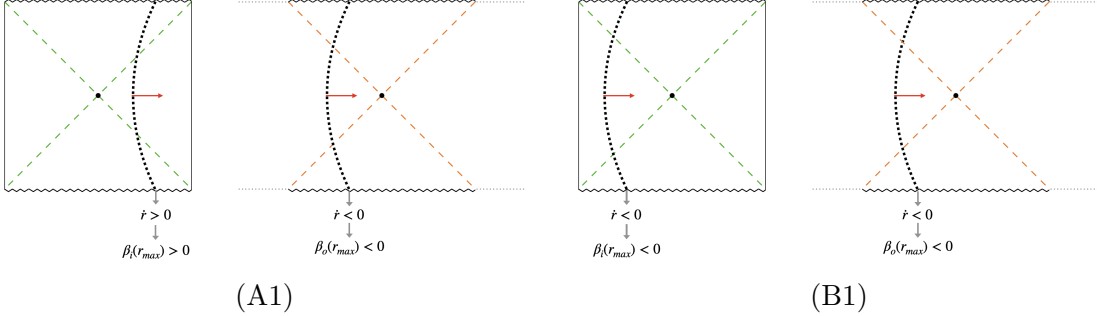

$$(A1) \qquad\qquad\qquad (B1)$$

**Figure 38**. Configurations for $\mu_A > \mu_S$

Similarly consider the configurations for $\mu_S > \mu_A$ shown in Fig. 39. Again we ignore geometries where the brane passes to the right of the SdS bifurcation surface as these would not yield symmetric gluings.

These are similar to the configurations found for $\mu_A > \mu_S$; however for $\mu_S > \mu_A$, $\beta_i > 0 \forall r$, so B2 is disallowed. Furthermore, $\beta_o < 0$ requires $\kappa > \sqrt{\lambda_{dS} + \lambda}$ for A2.

Thus we can summarize the (one-sided) gluings for $r = 0$ in Fig. 40, which correspond to cases 1, 2, and 3 in Fig. 6.

Now we consider gluings at $r = \infty$. Here $\beta_{i/o}$ are dominated by the 1st term ($\mathcal{O}(r)$) in (A.16)-(A.17) which leads to the brane trajectories shown in Fig. 41.

Since $\beta_i \to \infty$ when $r \to \infty$ for all $\lambda, \lambda_{dS}, \kappa$, the brane glued to the 'left' boundary

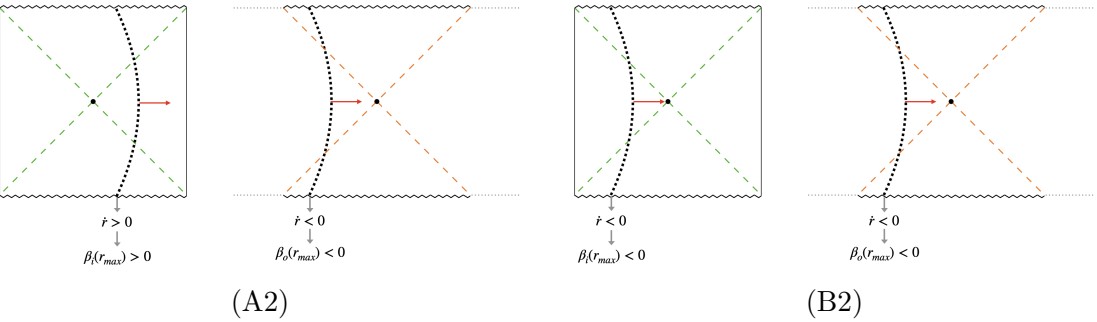

**Figure 39**. Configurations for $\mu_S > \mu_A$

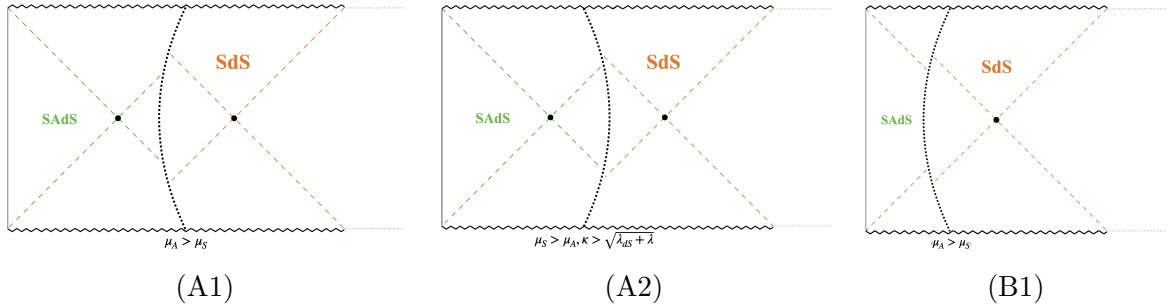

**Figure 40**. Configurations for branes at $r = 0$ with central metric SdS

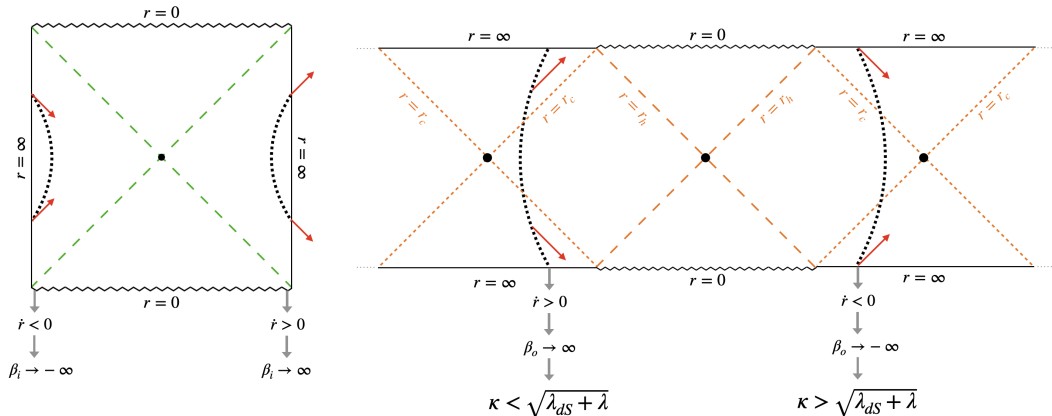

**Figure 41**. Conditions on branes attached at $r = \infty$. Note the limits are now for $r \to \infty$

of SAdS is disallowed; the brane can only be glued to the 'right' conformal boundary. Thus, we must only consider what bulk region the brane passes through in the SdS metric. There are 2 choices: to the left or to the right of the cosmological bifurcation surface, which have constraints as shown in Fig. 42.

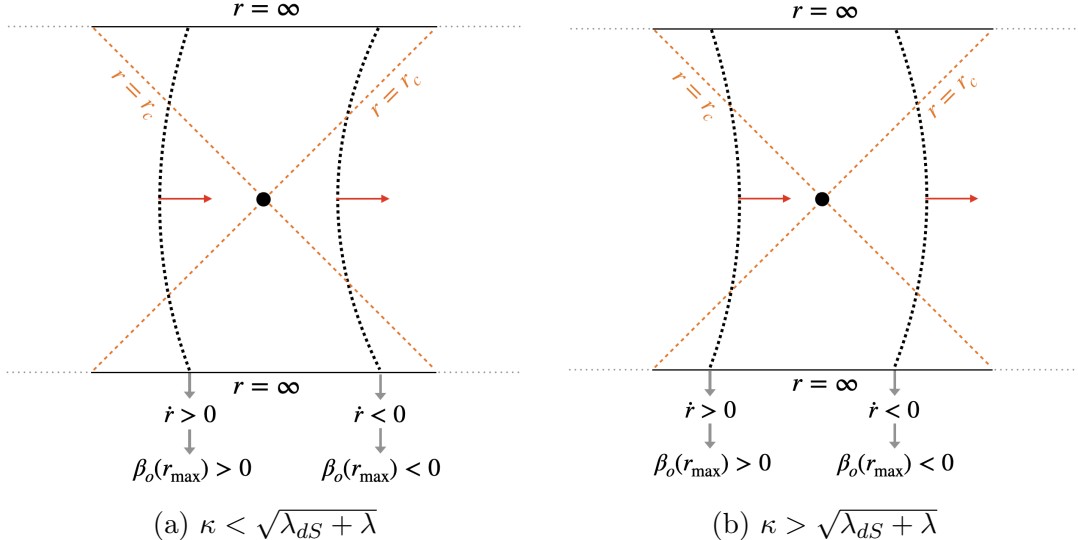

(a) $\kappa < \sqrt{\lambda_{dS} + \lambda}$  (b) $\kappa > \sqrt{\lambda_{dS} + \lambda}$

**Figure 42**. Conditions for $r = \infty$ gluings. Note that the cosmological bifurcation surface is (locally) maximal, not minimal like the black hole bifurcation surface

Starting with $\kappa < \sqrt{\lambda_{dS} + \lambda}$, this constrains $\beta_o$ to always be positive unless $\mu_A > \mu_S$. Thus for the brane on the right (i.e. "cutting out" the cosmological horizon) we have the additional constraint $\mu_A > \mu_S$, while there is no additional constraint for the brane on the left (i.e. including the cosmological horizon).

Similarly for $\kappa > \sqrt{\lambda_{dS} + \lambda}$, $\beta_o$ is always negative unless $\mu_S > \mu_A$. Thus for the brane on the left we have the additional constraint $\mu_S > \mu_A$ while there is no additional constraint for the brane on the right.

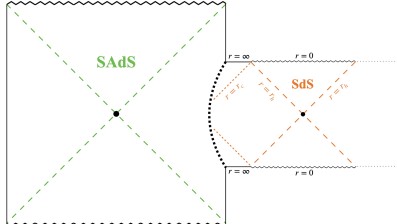
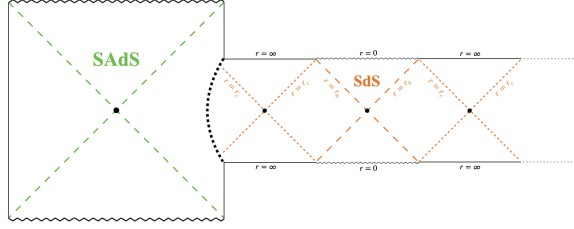

(C) For $\kappa < \sqrt{\lambda_{dS} + \lambda}$, additional constraint $\mu_A > \mu_S$ needed. No additional constraint for $\kappa > \sqrt{\lambda_{dS} + \lambda}$

(D) For $\kappa > \sqrt{\lambda_{dS} + \lambda}$, additional constraint $\mu_S > \mu_A$ needed. No additional constraint for $\kappa < \sqrt{\lambda_{dS} + \lambda}$

**Figure 43**. Conditions for $r = \infty$ gluings. Note that the SdS cosmological horizon is the maximal surface, not minimal like the black hole horizon

With this we have our allowed gluings for $r = \infty$ as shown in Fig. 43, the symmetric

2-sided gluings of which coincide with the case 4 and 5 geometries shown in Fig. 7 and 8. Furthermore, there is no analytic method to ensure all these cases can be formulated with parameters within the Nairai limit; this can be seen numerically where parameters obeying the Nairai limit can still be found to form all 5 types of gluings.

# B   Further details about Brill-Lindquist

## B.1   Symmetric configurations

The symmetries of the Brill-Lindquist (BL) metric for certain parameter choices are more clearly manifest when going to inverted coordinates, where we have the ansatz,

$$ds^2 = \phi(\vec{s})^4 d\vec{s}^2 \,, \qquad \phi(\vec{s}) = \sum_{i=1}^{n} \frac{\mu_i}{\|\vec{s} - \vec{s}_i\|} \,, \tag{B.1}$$

Taking $\vec{s}_n = 0$, we can perform the inversion $\vec{s} \to \mu_n^2 \vec{r}/r^2$ to relate the inverted coordinates to the original BL coordinates.[10] To obtain a cylindrically symmetric metric, we require the $n - 1$ punctures to be collinear (in BL co-ordinates) and only the separations between masses are physically meaningful, not the actual positions. Thus the positions $\{\vec{r}_i\}$ constitute $n - 2$ degrees of freedom, as we may fix say $\vec{r}_1 = 0$. Additionally, the BL parameters $\{\alpha_i\}$ constitute an additional $n - 1$ degrees of freedom, giving a total of $2n - 3$ degrees of freedom. However, in inverted coordinates we have the $n - 2$ positions $\{s_i\}$ ($\vec{s}_n = 0$ and $\vec{s}_1$ is fixed by $\vec{r}_1$) and $n$ parameters $\{\mu_i\}$, yielding $2n - 2$ degrees of freedom. This suggests a redundant parameter in inverted coordinates, and we make the choice to additionally fix $\vec{s}_2$. The question now remains on what is a convenient choice for $\vec{s}_2$.

Motivated by the symmetric configurations of subsections 4.2 and 4.3, we choose to set up the punctures in inverted coordinates along the circumference of a unit circle centered in the $z' - \rho'$ plane at $\rho' = -1$ (numerical analysis uses cylindrical coordinates as detailed in Appendix B.2). Now we take $\mu_n$ placed at the origin and $\mu_1, \mu_2$ placed on the vertices of a regular $n-$gon inscribed in the circle and the remaining punctures arbitrarily along the circle, as shown in Fig. 44. There are 2 reasons for doing so, the first being that performing the inversion $\vec{s} \to \mu_n^2 \vec{r}/r^2$ maps the circle to the line $\rho = -0.5$, thus ensuring the punctures are collinear in BL coordinates. The 2nd reason is that by placing $\mu_1, \mu_2$ (and $\mu_n$) on the vertices of a regular $n-$gon, to obtain a $D_n$ symmetric configuration one must simply place the remaining $n - 3$ punctures on the remaining vertices of the $n-$gon.

---

[10]Recall that the point $s = \infty$ must be appended to the inverted coordinates, thus making the inverted coordinates isomorphic to a 3-sphere (via steregraphic projection)

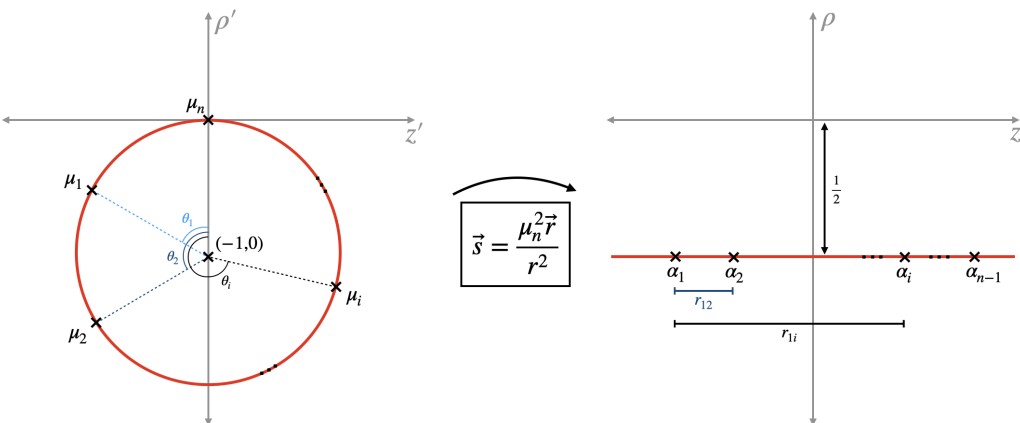

**Figure 44**. Setup to ensure the masses remain collinear. On the left are the masses placed on a circle in inverted co-ordinates (represented by the primed cylindrical co-ordinates) which are transformed by $\vec{s} = \mu_n^2 \vec{r}/r^2$ to the Brill-Lindquist co-ordinates on the right (unprimed cylindrical coordinates). Note that $\rho, \rho'$ must be non-negative; here we are using an abuse of notation where $\rho, \rho' \geq 0$ corresponds to $\phi, \phi' = 0$ and $\rho, \rho' < 0$ corresponds to $\phi, \phi' = \pi$, with $\phi, \phi'$ being the respective cylindrical angles

More explicitly, we can parameterise the positions of the punctures in inverted coordinates using the polar angle along the circle (with $\theta_n = 0$ being the origin), i.e. $\vec{s}_i = (z'_i, \rho'_i) = (-\sin\theta_i, \cos\theta_i - 1)$. Then we fix the angles $\theta_1 = \frac{2\pi}{n}$ and $\theta_2 = \frac{4\pi}{n}$. Now the other masses can be placed anywhere within $\theta_n \in (\theta_2, 2\pi)$ with arbitrary masses $\mu_1, \ldots, \mu_n$ and $\theta_n > \theta_m$ for $n > m$ to yield arbitrary configurations of $n-1$ collinear masses in the original co-ordinates. For our $D_n$ symmetric configurations, we simply set $\mu_1 = \mu_1 = \cdots = \mu_n$ and $\theta_i = \frac{2\pi i}{n}$. This yields the relations,

$$\alpha_i = \frac{\mu_i \mu_n}{\sqrt{2(1 - \cos\theta_i)}}, r_{1i} = \frac{\mu_n^2}{2}\left(\frac{\sin\theta_1}{1 - \cos\theta_1} - \frac{\sin\theta_i}{1 - \cos\theta_i}\right) \tag{B.2}$$

where $r_{1i}$ is the separation from puncture 1 to $i$, assuming 1 is at the origin in BL coordinates.

## B.2 Finding minimal surfaces

Standard techniques to find minimal surfaces fail to provide a general, analytic answer. As a result, numerical methods are used to locate Brill-Lindquist minimal surfaces. The standard approach is to write down an integral for the area and find the surface that minimizes this. We also impose cylindrical symmetry by setting all point masses to be collinear. It is then best to use cylindrical coordinates $(\rho, \theta, z)$ to exploit this

symmetry, as the problem of finding the minimal surface is now simplified to finding a curve in the upper half $\rho - z$ plane that can be rotated about the z-axis to obtain the complete surface.

Parameterising the curve by $(\rho(\lambda), z(\lambda))$ we can write down the area functional:

$$A = 2\pi \int_{\lambda=0}^{\lambda=\lambda_1} \rho\psi^2 \left(\psi^4 dz^2 + \psi^4 d\rho^2\right)^{1/2} \tag{B.3}$$

$$= 2\pi \int_{\lambda=0}^{\lambda=\lambda_1} \left[Q^2\dot{z}^2 + Q^2\dot{\rho}^2\right]^{1/2} d\lambda \tag{B.4}$$

where $Q = \rho\psi^4$ and $\cdot = d/d\lambda$ (unless specified, from now on a dot represents a derivative w.r.t $\lambda$).

Minimising yields an equation each for $\rho$ and $z$:

$$\frac{d\left(Q\dot{\rho}(\dot{z}^2 + \dot{\rho}^2)\right)}{d\lambda} = Q_{,\rho}\left(\dot{z}^2 + \dot{\rho}^2\right)^{1/2} \tag{B.5}$$

$$\frac{d\left(Q\dot{z}(\dot{z}^2 + \dot{\rho}^2)\right)}{d\lambda} = Q_{,z}\left(\dot{z}^2 + \dot{\rho}^2\right)^{1/2} \tag{B.6}$$

To solve these equations, there are several gauge choices we can make. We focused on the gauges proposed by Cadez [89] and Bishop [90].

Bishop proposed the gauge $\dot{z}^2 + \dot{\rho}^2 = Q^{-2}$. The equations then reduce to:

$$\ddot{\rho} = \frac{Q_{,\rho}}{Q^3} - \frac{2\dot{\rho}\dot{Q}}{Q} \tag{B.7}$$

$$\ddot{z} = \frac{Q_{,z}}{Q^3} - \frac{2\dot{z}\dot{Q}}{Q} \tag{B.8}$$

The advantage of this gauge is that substituting into B.4 reduces the expression for the area to $A = 2\pi\lambda_1$.

Cadez proposed the alternative gauge $\dot{z}^2 + \dot{\rho}^2 = \psi^8$ to aid with numerical computations, which yields the equations:

$$\ddot{\rho} = \frac{\dot{z}^2}{\rho} + \frac{1}{2}\left(\psi^8\right)_{,\rho} \tag{B.9}$$

$$\ddot{z} = -\frac{\dot{z}\dot{\rho}}{\rho} + \frac{1}{2}\left(\psi^8\right)_{,z} \tag{B.10}$$

Both gauges were used in our analysis, with the Cadez gauge providing greater numerical accuracy and the Bishop gauge providing an easy computation for the area.

Given the differential equations, we can now write down the boundary value problem (BVP) that corresponds to finding the minimal surfaces:

$$\rho(0) = \rho(\lambda_1) = 0 \tag{B.11}$$

$$\dot{z}(0) = \dot{z}(\lambda_1) = 0 \tag{B.12}$$

where B.11 ensures that we find a closed surface (upon rotating the curve about the $z$-axis), B.12 ensures the surface is continuous at $\rho = 0$ and $\lambda_1$ is the end point of the integration.

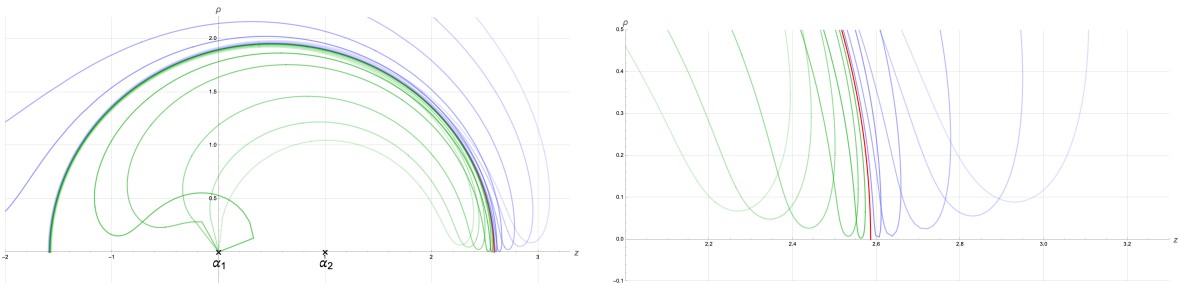

**Figure 45**. Visualization of the shooting method with a close up (right) of the integration between $z = -2$ and $z = 3$ (left). The red curve is the minimal surface $\gamma_1$, with blue curves originating to the left and green to the right. The blue curves are divergent while green curves converge to $\alpha_1$; this behaviour is interpolated to find $\gamma_1$

In lieu of an analytical solution, numerical solutions of BVPs can be given via shooting methods, where the BVP is turned into an IVP (initial value problem). Then a guess-and-check method is implemented, where various guesses for the initial conditions are checked until the endpoint conditions are satisfied for some value of $\lambda_1$. The initial value for $\dot{\rho}$ is constrained by the choice of gauge; for example, for the Cadez gauge, we have $\dot{\rho}(0) = \sqrt{(\psi(0))^8 - (\dot{z}(0))^2} = (\psi(0))^4$. Thus it is the initial $z$-value, $z(0) \equiv z_i$ that must be guessed and checked. Cadez [89] noted that the behavior of the integrated curve is also different immediately before and after $z_i$. For instance, assuming $z < z_i$ but close to $z_i$, the paths of integration converge to one of the masses. For $z > z_i$ but close to it, the paths of integration are divergent and vice versa if the behavior is reversed. Therefore, $z_i$ can be found via interpolation with a sufficiently fine grid. Similarly, the sign of $\dot{z}(\lambda_1)$ depends on whether $z_i \lessgtr z_0$, which provides an additional check on the value of $z_i$ [90]. Finally, we have been making the assumption that we can start and end our integration at $\rho = 0$; however (B.5), (B.6) are singular for $\rho = 0$, irrespective of gauge. Thus, instead of $\rho = 0$, the integration is started/ended at a tolerance $\tau > 0$ such that $\rho(0) = \rho(\lambda_1) = \tau$ for some $\lambda_1$.

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
