# Peer review of "Entangled universes"

_SciPost Physics_

## Round 1 · Referee Report · Anonymous (Referee 1) · 2025-8-20

Report

This paper argues that various classes of multiboundary asymptotically flat or AdS spacetimes represent entangled states in product Hilbert spaces each of which corresponds to one asymptotic region, based on certain generalizations of the RT/HRT holographic entanglement formulation. One of the main points the authors emphasize is that in discussions pertaining to the entire CFT, the existence of the dual CFT description is not necessary -- thus the entanglement discussed here is between two bulk universes, be they AdS-like or more general. This is somewhat different from various studies of holographic entanglement that discuss extremal surfaces anchored to the boundaries of the dual CFT boundary subregions. The extremal surfaces considered here are untethered bulk objects (e.g. appropriate bifurcation surfaces), so that data on the asymptotics disappears -- this allows the authors to generalize their arguments beyond AdS, where the dual non-gravitational (CFT) description is not as well-established. The authors argue that their candidate extremal surfaces satisfy various criteria known for standard RT/HRT surfaces. Concrete examples discussed nicely include classes of Brill-Lindquist spacetimes.

The paper is interesting and I recommend publication: but first I have the following questions [the references below are just for context]:

(1) perhaps most centrally, I'd like to understand the relation of the untethered extremal surfaces here to the IR limits of standard boundary-anchored RT/HRT surfaces admitting holographic entanglement entropy interpretations (in cases with known hologram).

(a) For instance, in the eternal AdS Schwarzschild black hole (dual to the CFT[L]xCFT[R] in the thermofield double state), standard boundary anchored extremal surfaces are the Hartman-Maldacena ones (1303.1080) connecting the left-right copies of the CFT for finite L+R subregions. In the IR limit (subregions going to entire space), do these end up effectively going over to the bifurcation surface discussed here in sec.1.1.I? I recall some discussions of this sort in Hartman-Maldacena (or maybe later work). Pictorially, for single boundary, this is reminiscent of the RT surfaces in the AdS black hole wrapping the horizon, thus appearing to localize there in the IR limit (entire boundary space), so I'm just wondering if the boundary-anchored boundary condition remains, esp. in the multiboundary case. In cases where there is a clear dual CFT description, it would appear that this has a precise answer.

(b) For eternal flat Schwarzschild, I suspect similar arguments may work if one introduces AdS-like boundaries on the left+right and finds the IR limit gives the bifurcation surfaces.

If in certain cases this IR limit does not end up with the bifurcation surface however, one might want to understand better the conditions under which one picks boundary-anchored vs untethered (closed) extremal surfaces (finite subregion vs entire boundary?).

(c) As a technical aside, since the discussion is about the entire space, perhaps an effective 2-dimensional description obtained from dimensional reduction over the transverse space (generalizing JT gravity in extremal black holes to generic 2-dim dilaton gravity) can be used efficiently on general grounds. The classical (dilaton) part of the generalized entropy then appears to map to the entropy here.

(2) With dS asymptotics, the authors discuss certain analogous surfaces in sec.5 such as in the AdS-dS wormhole. However I'm wondering about the status of the bifurcation surface in pure dS itself and whether (in line with this paper) it can be argued to encode bulk spatial entanglement between the Left+Right dS-Rindler universes (the area here is dS entropy).
An alternative interpretation, from a dS/CFT point of view, might be as bulk "timelike entanglement" between the Future+Past (dS-Milne) universes, which is roughly a space-time rotation. These sorts of admittedly novel (although consistent) arguments based on future-past extremal surfaces appear in arXiv:2210.12963 for CFT[F]xCFT[P], building on 1711.01107 and ref.[25]. In some ways, this appears more in line with the heterodox case, as in the paragraph after Fig.24, pg.45 (and the discussions in sec.6).

A minor typo: there appears to be a mismatch in the referencing of Fig.1 in some places -- e.g. sec.1.1. II (pg.4) quotes M_{II} in Fig.1 bottom left, whereas it is top right as per Fig.1. Likewise, sec.1.1.III pertains to Fig.1 bottom left I think.

Recommendation

Ask for minor revision

---

## Round 1 · Referee Report · Aron Wall (Referee 2) · 2025-8-21

Report

As I did not discover errors in the paper, I will focus this review on the question of originality and suitability for the journal.

For the most part, the holographic entanglement entropy conjecture has been studied in asymptotically AdS spacetimes.

The main point of this paper is to point out (with a variety of detailed calculations and arguments) that various aspects of the Ryu-Takayanagi and HRT proposals continue to work in asymptotically flat spacetimes (Lambda = 0), involving black holes and other wormhole configurations.

I found the arguments for this convincing, and novel. While I am tempted to say that the overall result was fairly obviously true--I certainly believed it before reading the paper--the fact is that nobody had seriously made the case in detail (to the best of my knowledge), and certainly not as carefully as these authors have done so. The authors therefore deserve the credit for this hypothesis.

As asymptotically flat spacetimes are a very important generalization of the holographic entropy dictionary, I therefore believe this paper adequately meets the criterion of "Open[ing] a new pathway in an existing or a new research direction, with clear potential for multi-pronged follow-up work."

The paper is also very clearly written and a pleasure to read.

(I am not totally convinced that the dS proposals are right, and indeed the authors propose multiple ways of dealing with such spacetimes. But this is not a criticism, as the section is written with an apporpriate degree of modesty and tentativeness, more as a guide to future possible directions.)

I therefore recommend publication in SciPost Physics.

Recommendation

Publish (meets expectations and criteria for this Journal)

---

## Round 1 · Referee Report · Anonymous (Referee 3) · 2025-9-4

Disclosure of Generative AI use

The referee discloses that the following generative AI tools have been used in the preparation of this report:

I used deepseek to polish the report.

Report

The Ryu-Takayanagi (RT) and Hubeny-Rangamani-Takayanagi (HRT) formulas provide a geometric description of entanglement entropy within the AdS/CFT correspondence, a holographic duality between asymptotically Anti-de Sitter spacetime and conformal field theory. However, our universe is not asymptotically AdS. A major open question has been whether and how these holographic concepts extend beyond the AdS/CFT correspondence. This paper proposes a generalization of the RT and HRT formulas to general spacetimes with multiple conformal boundaries. The entanglement entropy for a subset of the boundaries, denoted by $A$, is proposed to be proportional to the area of the HRT surface homologous to $A$.

Compared to previous proposals for holographic entanglement entropy in flat holography or dS/CFT, this work focuses on entire boundary connected components rather than subregions. As a result, the HRT surfaces are not boundary-anchored, unlike those in the aforementioned proposals. In special cases, the proposal reduces to the surface-state correspondence or the generalized entanglement wedge proposal, but it is more conservative.

As supporting evidence, the paper shows that the generalized HRT surfaces satisfy several expected properties of entanglement entropy. The authors also provide an alternative argument by gluing AdS spacetime onto a Schwarzschild black hole via a domain wall and argue that in some cases their proposal can be obtained from the standard HRT formula. Additionally, they carry out explicit computations in the Brill-Lindquist spacetime and uncover interesting phase transition structures. The paper also discusses possible extensions to de Sitter spacetimes.

The paper is well written, and I find most parts of the arguments convincing. I recommend the paper for publication in SciPost, with the following comments/questions:

  1. The proposal in this paper relies on the assumption that there are multiple boundaries. It would be interesting to see how the conjecture could be extended to subregions.

  2. When a Schwarzschild black hole is glued to asymptotic AdS regions via domain walls, is the holographic entanglement entropy still given by the standard HRT surface? In the Lewkowycz-Maldacena argument, the computation of entanglement entropy amounts to evaluating the variation of the action, whose contribution localizes to the boundary. Could the domain walls introduce additional contributions to the generalized gravitational entropy? The answer to this question will affect the validity of the argument in section 3.

Recommendation

Publish (easily meets expectations and criteria for this Journal; among top 50%)

---

## Editorial Decision

resubmitted